# Pan-RAF inhibitor exarafenib targets BRAF class II/III NSCLC and reveals ARAF-KSR1 resistance and combination strategies

Tadashi Manabe [1,2,3] ✉, Hannah C. Bergo [1,2,3], Qingtian Li[1,2,3], Tim Sen Wang[4], Paul Severson [4], Nichol Miller[4], Catherine Lee [4], Elifnur Yay Donderici[5], Nicole Zhang [5], Wei Wu [1,2,3], Yu-Ting Chou[1,2,3], Daniel L. Kerr[1,2,3], Paul Allegakoen[1,2,3], Kathryn B. Grandinetti [4], Liliana Soroceanu[4], Robert J. Pelham[4], Eric S. Martin[4], Eric A. Murphy[4], Vishesh Khanna[6], Joel W. Neal [6], Christopher T. Chen[6], Shumei Kato [7], Richard Williams[4] & Trever G. Bivona [1,2,3,8] ✉

Oncogenic BRAF mutations, including those in non-small cell lung cancer (NSCLC), are classified as Class I, II, or III. While approved therapies exist for BRAF Class I mutants, no approved therapies exist for Class II and III BRAF-mutated NSCLC. Analysis of a circulating tumor DNA database reveals Class II and III mutations comprise ~65% of BRAF-mutant NSCLC cases, with Class II patients showing worse outcomes than Class I. Exarafenib, a distinct pan-RAF inhibitor, demonstrates potent activity against BRAF Class II and III mutant preclinical models and initial clinical activity. Resistance studies reveal rewiring to an ARAF-mediated bypass pathway, characterized by RAS-mediated ARAF-KSR1 complexes maintaining MAPK signaling despite pan-RAF inhibitor treatment. RAS or MEK inhibition co-targeting is effective against this resistance mechanism. This study provides preclinical rationale for clinical testing of exarafenib in BRAF Class II/III cancers and unveils RAS-mediated ARAF-KSR1 complex formation as a resistance mechanism and rational co-therapy strategies.

The RAF family of serine/threonine kinases, consisting of ARAF, BRAF, and CRAF, plays a pivotal role in the mitogen-activated protein kinase (MAPK) signaling pathway by transducing signals from RAS GTPases to downstream effectors, including MEK and ERK, ultimately regulating cell proliferation, differentiation, and survival[1]. Under normal physiological conditions, this tightly regulated signaling cascade is essential for normal cell growth and development. However, in cancer, the MAPK pathway is frequently dysregulated, often through oncogenic mutations in RAS or RAF genes, leading to increased pathway activation and oncogenic cell growth[2].

BRAF mutations are present in a wide range of human cancers, with varying frequency across different tumor types[3]. These mutations represent a critical therapeutic target, as evidenced by the transformative impact of BRAF inhibitors in melanoma and other cancers. From recent studies, these mutations can be divided into three distinct classes based on their biochemical and signaling mechanisms: Class I mutations (e.g., V600E) are often considered RAS-independent and function as monomers; Class II mutations are independent of RAS and function as homodimers; and Class III mutations depend on RAS and act through heterodimer formation[4,5]. While selective BRAF inhibitors

[1]Department of Medicine, University of California, San Francisco, San Francisco, CA, USA. [2]Department of Cellular and Molecular Pharmacology, University of California, San Francisco, San Francisco, CA, USA. [3]Helen Diller Family Comprehensive Cancer Center, University of California, San Francisco, San Francisco, CA, USA. [4]Kinnate Biopharma, San Diego, CA, USA. [5]Guardant Health, Redwood City, CA, USA. [6]Division of Oncology, Department of Medicine, Stanford University School of Medicine, Stanford, CA, USA. [7]Division of Hematology and Oncology, University of California San Diego, Moores Cancer Center, La Jolla, CA, USA. [8]Chan-Zuckerberg Biohub, San Francisco, CA, USA. ✉e-mail: tadashi.manabe@ucsf.edu; trever.bivona@ucsf.edu

have transformed the treatment of Class I BRAF V600E-mutant cancers[6–8], there remains an unmet need for novel therapeutic strategies that can effectively target a broader spectrum of BRAF mutations that are classically refractory to current approved BRAF inhibitors, such as BRAF Class II and III mutants.

In non-small cell lung cancer (NSCLC), the leading cause of cancer mortality worldwide, BRAF mutations are present in approximately 2-4% of cases[9]. It is important to note that even a small percentage frequency of a specific genomic subset of NSCLC can equate to a substantial clinical cohort (e.g., tens of thousands of patients annually), given the high incidence of NSCLC in the US and worldwide[10,11]. Previous studies reporting the frequency and classification of BRAF mutations in NSCLC have been limited by relatively small sample sizes, likely due to the invasive nature of tissue biopsies and the associated risks for patients[12,13]. This has potentially led to an incompletely accurate estimation of the true prevalence and diversity of these mutations. Furthermore, despite the growing recognition that BRAF mutations are classified into three classes, the clinical characteristics and outcomes of each in patients with NSCLC are not yet fully established[12,14–16]. Recent advancements in genomic platforms and data repositories, such as the GuardantINFORM™ database[17], which is a comprehensive and validated resource of genomic data obtained through non-invasive liquid biopsies from patients with advanced solid tumors, now offer fresh opportunities to more thoroughly investigate the prevalence and distribution of BRAF mutations (and other genomic alterations) in cancers. The use of liquid biopsies, despite certain caveats, can allow for a comprehensive analysis of the tumor genomic landscape, as it can capture tumor heterogeneity and enable the detection of mutations that may be missed by traditional tissue biopsies[18,19]. By leveraging this valuable resource, we aimed to gain insights into the clinical landscape of BRAF-mutant tumors, with a focus on NSCLC, and identify potential differences in patient outcomes across mutation classes, addressing current knowledge gaps.

Given that there are limited effective therapeutic options for patients with Class II and III BRAF mutations, we also conducted a comprehensive investigation of the therapeutic potential of exarafenib, a novel type 2 pan-RAF inhibitor and ATP-competitive small molecule agent[20], in BRAF-mutant NSCLC. Exarafenib targets both monomeric and dimeric forms of mutant BRAF, potentially overcoming the limitations of selective V600X inhibitors, which target monomers[4,21]. Through systematic preclinical studies, we show that exarafenib achieves potent anti-tumor activity across all mutation classes, with robust efficacy in diverse cell lines and patient-derived xenograft models. To identify potential biomarkers of response and rational combination partners that could enhance therapeutic efficacy and maintain tolerability, we systematically profiled exarafenib-resistant derivatives. Through this approach, we uncovered an unexpected adaptive resistance pathway involving drug-induced ARAF-KSR1 scaffold complex formation that maintains MAPK signaling despite pan-RAF inhibitor treatment. This mechanistic discovery led us to identify MEK inhibition as a synergistic combination strategy that prevents resistance by targeting a key convergence point of both canonical and bypass signaling pathways.

Our study provides comprehensive insights into the prevalence, clinical characteristics, and outcomes in patients harboring different classes of BRAF mutations and also underscores exarafenib as a targeted therapeutic approach for this molecularly defined patient population. Results of this study support the ongoing clinical development of exarafenib and have the potential to transform the clinical management of BRAF-mutant NSCLC and improve outcomes for the substantial number of patients with this challenging and deadly disease.

## Results

### Genomics, features, and outcomes of BRAF class mutant subtypes in patients including those with NSCLC

To gain additional insight into the biology and clinical relevance of the different BRAF mutations present in human cancer patients, we investigated the detection rate of oncogenic BRAF mutations (excluding fusions, large structural variants, or copy number variants) across various malignancies using the GuardantINFORM™ database, which contains targeted genomic data from over 160,000 patients diagnosed with advanced or metastatic cancer. In our analysis, we ensured that each patient was counted only once within each BRAF class to provide an accurate assessment of BRAF mutation prevalence. The classification of BRAF alterations is shown in Supplementary Table 1. Notably, while BRAF Class I mutations detected in circulating tumor DNA (ctDNA) were observed at rates exceeding 10% in specific cancers such as Erdheim-Chester disease (ECD), Langerhans cell histiocytosis (LCH), thyroid cancers, and melanoma (Fig. 1a), these mutations were comparatively less frequent in lung cancer (1.67%). However, despite the lower frequency of Class I mutations in lung cancer compared to other malignancies, lung cancer represented the largest number of patients with BRAF alterations across all cancers surveyed, given the large number of patients diagnosed annually[10,11] (Fig. 1a and Supplementary Table 2). Since lung cancer had the highest absolute number of tumors with BRAF mutations, we focused our subsequent analyses on the clinical characteristics and outcomes of NSCLC patients with BRAF alterations. Specifically, among the 51,783 lung cancer patients with detectable ctDNA, 2398 patients (4.63%) had tumors with non-overlapping BRAF mutations belonging to one of the three functional classes (Class I, II, or III only) (Supplementary Table 2). An additional 23 NSCLC patients had multiple BRAF alterations across functional classes (e.g., Class I and II, II and III), and these patients were counted as BRAF mutation positive in the overall analysis but excluded from individual BRAF class counts to avoid duplication. In NSCLC, BRAF Classes II and III are present in 65% of the BRAF-positive patients (Fig. 1b), with distinct frequencies among different mutations within each class (Supplementary Tables 3, 4 and 5). Interestingly, this distribution of BRAF mutation classes in NSCLC, with a higher proportion of Class II and III mutations, differs from patterns typically observed in many other cancers where BRAF V600E (Class I) mutations predominate[3,22]. These findings, consistent with prior tissue-based genomic studies[3,15], highlight an important unmet need for effective therapies targeting this large subgroup of patients with BRAF Class II and III mutations, which we address below.

The clinical characteristics of NSCLC patients with non-overlapping BRAF Class I, II, or III alterations are summarized in Supplementary Table 6. Across all three BRAF classes, the median age of patients was 70 years, consistent with a previous study[15]. Class I mutations were more frequently observed in females compared to males (Class I vs II: *$P = 0.0451$; Class I vs III: *$P = 0.0373$). BRAF Class II and III mutations were more prevalent in patients with a history of tobacco use compared to Class I mutations (Class I vs II: ****$P = 8.18 \times 10^{-9}$; Class I vs III: ****$P = 4.52 \times 10^{-8}$), in line with a previous report[15]. Our study, leveraging liquid biopsy technology, successfully extended these prior findings by enabling the analysis of a larger patient cohort to provide a more comprehensive understanding of the genomic landscape of BRAF alterations in lung cancer.

Given the distinct clinical characteristics observed in patients with different BRAF mutation classes, we sought to investigate potential associations between these genomic alterations and patient outcomes. Real-world overall survival (rwOS), defined as the time from the first detection of BRAF mutation in circulating tumor DNA to death from any cause, was evaluated in 813 unique NSCLC patients with BRAF alterations. The Kaplan-Meier survival curves for each BRAF mutation class are presented in Fig. 1c. The median rwOS was significantly longer

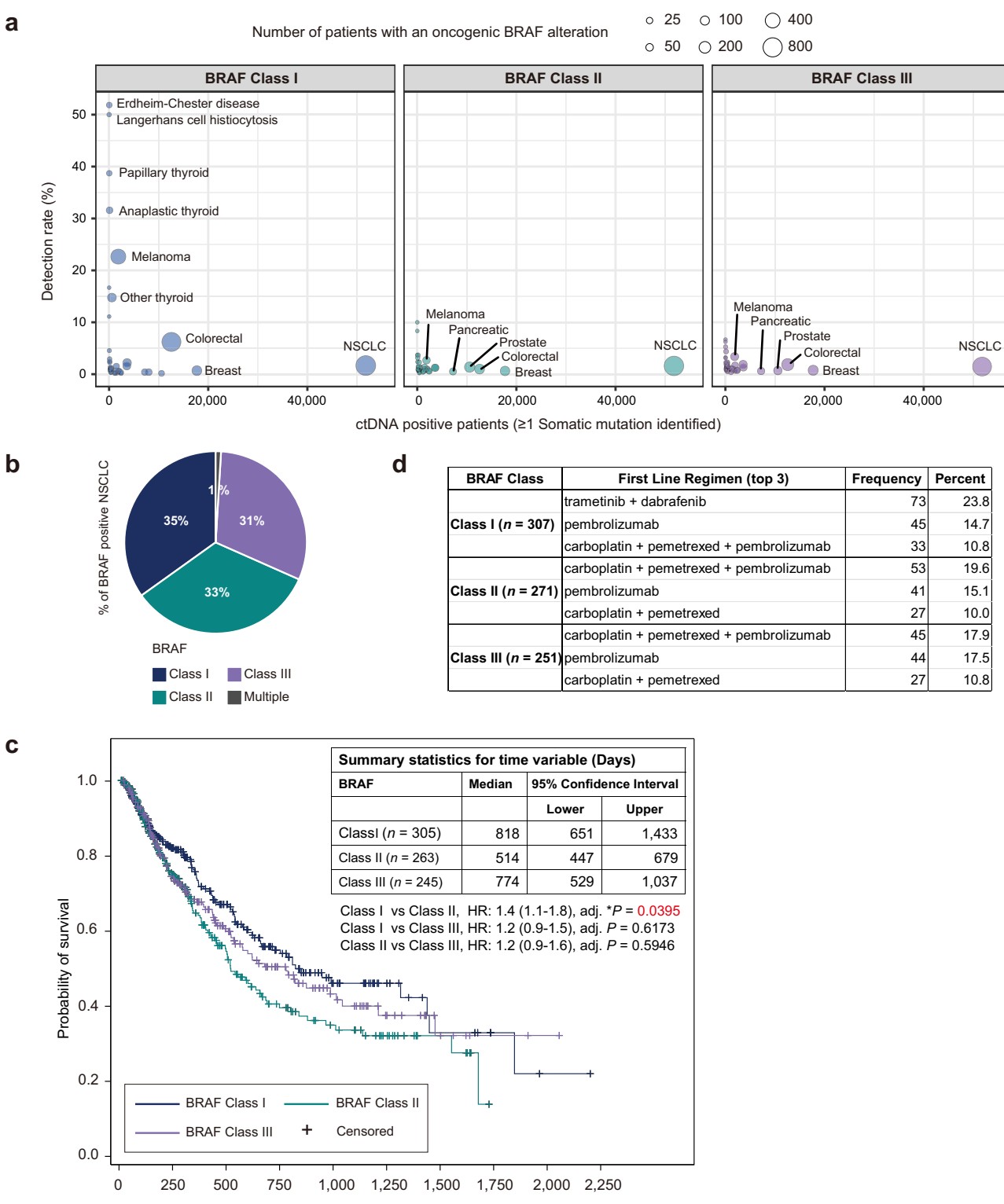

**Fig. 1 | Clinical genomic landscape and outcomes of NSCLC patients with oncogenic BRAF alterations. a** Detection rate of oncogenic BRAF Class I, II, and III mutations in ctDNA across various malignancies. Circle size represents the number of patients with an oncogenic BRAF alteration. **b** Distribution of BRAF mutation classes in NSCLC patients: Class I (35%), Class II (33%), and Class III (31%). **c** Kaplan−Meier curves depicting real-world overall survival (rwOS) for NSCLC patients with BRAF Class I, II, and III mutations. Median rwOS significantly longer in Class I versus Class II (818 vs 514 days, HR: 1.4 (1.1− 1.8), adjusted *P-value = 0.0395). Class III median rwOS (774 days) not significantly different from Class I (HR: 1.2 (0.9 − 1.5), adjusted P-value = 0.6173) or Class II (HR: 1.2 (0.9−1.6), adjusted P-value = 0.5946).

Statistical significance was determined using pairwise log-rank tests (two-sided) adjusted for multiple comparisons using the Bonferroni method. Test statistics were $\chi^2 = 6.15$ (Class I vs II), $\chi^2 = 1.60$ (Class I vs III), and $\chi^2 = 1.66$ (Class II vs III) with degrees of freedom (df) = 1. Hazard ratios (HR) and 95% confidence intervals (CI) were estimated using a Cox proportional hazards model. Table summarizes time variable statistics (days) for each BRAF class. *$P < 0.05$. **d** First-line treatment regimens for BRAF-mutant NSCLC patients from each mutation class. The table shows the top 3 most frequently used first-line therapies with corresponding patient numbers and percentages. The entire treatment information for 1st, 2nd, and 3rd-line therapies by BRAF mutation class has been provided in Supplementary Data 1.

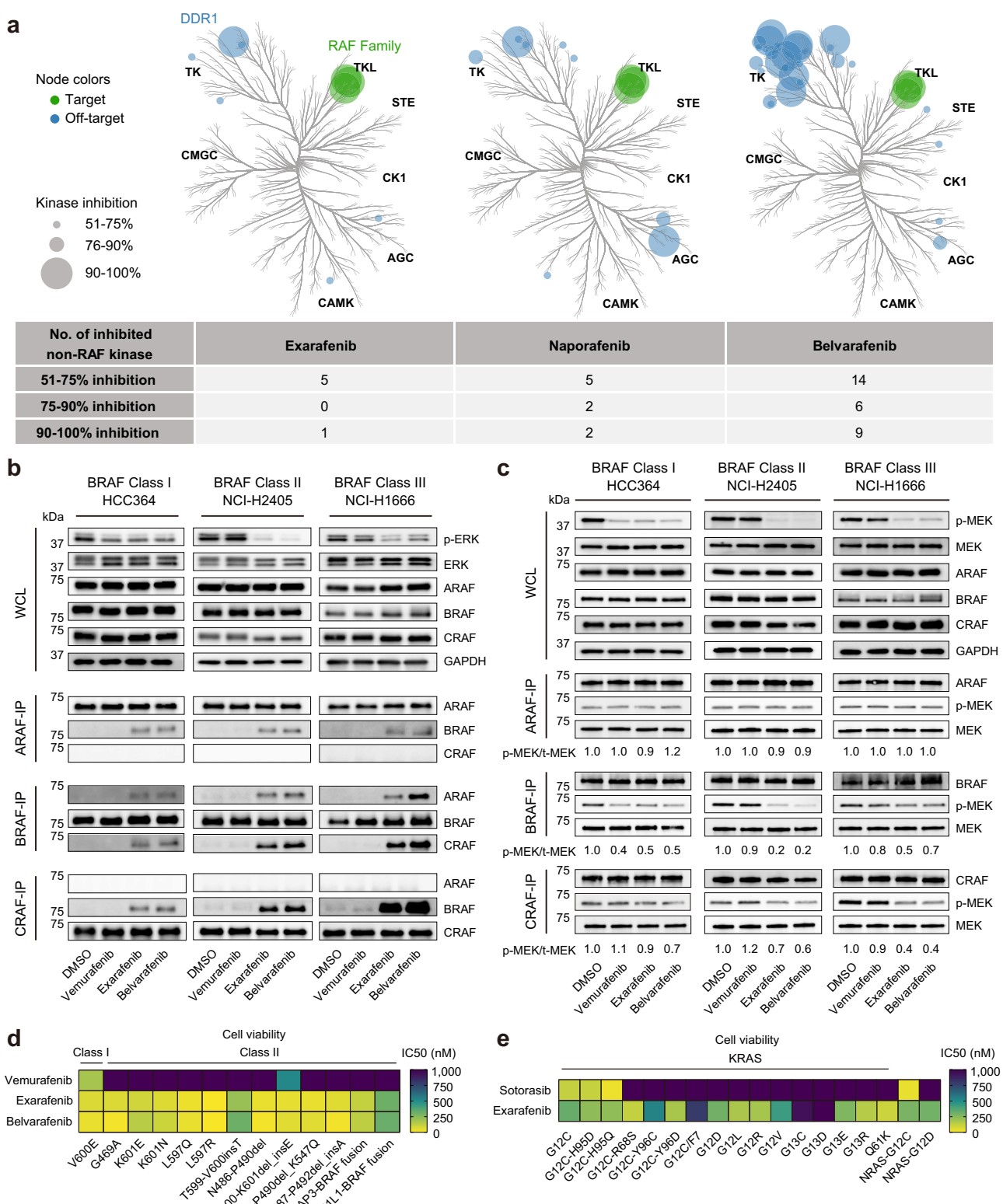

| No. of inhibited non-RAF kinase | Exarafenib | Naporafenib | Belvarafenib |
|---|---|---|---|
| 51-75% inhibition | 5 | 5 | 14 |
| 75-90% inhibition | 0 | 2 | 6 |
| 90-100% inhibition | 1 | 2 | 9 |

in BRAF Class I compared to BRAF Class II (818 vs 514 days, HR: 1.4 (1.1–1.8), adj. *P = 0.0395). The median rwOS for BRAF Class III was 774 days, which was not significantly different from Class I (HR: 1.2 (0.9–1.5), adj. P = 0.6173) or Class II (HR: 1.2 (0.9–1.6), adj. P = 0.5946). These survival differences could reflect differential access to effective targeted therapies (Fig. 1d). Class I patients had access to FDA-approved targeted therapy, with nearly a quarter (23.8%) receiving trametinib plus dabrafenib as first-line treatment. In contrast, Class II

and III patients, lacking effective targeted options, predominantly received immunotherapy-based regimens, including carboplatin plus pemetrexed plus pembrolizumab (19.6% and 17.9%, respectively) or pembrolizumab monotherapy (complete treatment information is provided in Supplementary Data 1).

Overall, our analysis confirms and extends prior observations, demonstrating that NSCLC patients with BRAF Class II mutations have worse prognosis compared to those with Class I mutations, potentially

**Fig. 2 | Exarafenib demonstrates potent RAF inhibition with minimal off-target effects and inhibits RAF monomers and dimers. a** Kinome trees showing the specificity profiles of exarafenib, naporafenib, and belvarafenib (1 μM) across 368 wild-type kinases. Green and blue circles represent on-target RAF and off-target kinases, respectively. Circle size indicates the level of inhibition. Exarafenib data are from ref. 20.; naporafenib and belvarafenib data were generated in this study. The table summarizes the number of inhibited non-RAF kinases at different thresholds. **b** IP-western blot analysis of ARAF, BRAF, CRAF, and MEK in NSCLC cell lines with different BRAF mutation classes, treated with 300 nM of indicated drugs for 4 h. Exarafenib and belvarafenib induce RAF heterodimerization. Data are representative of three independent experiments with similar results. **c** In vitro kinase assay evaluating the effects of vemurafenib, exarafenib, and belvarafenib on RAF isoform kinase activity in NSCLC cell lines. Exarafenib and belvarafenib exhibit potent inhibition of BRAF kinase activity in Class I and II BRAF mutations, while their

inhibitory effects are dominantly detected in CRAF kinase activity in Class III mutations. Data are representative of three independent experiments with similar results. **d** Cell viability assay showing the anti-proliferative effects of the drugs in Ba/F3 models expressing various Class I and II BRAF mutations. Heatmap presents $IC_{50}$ values, with lower values indicating higher potency. Vemurafenib shows high potency primarily against the V600E mutation (IC50 195 nM), while exarafenib (IC50 range: 7.4 nM−325 nM) and belvarafenib (IC50 range: 7.4 nM−332 nM) exhibit broad effects across all tested mutations. IC50 values were calculated from data of $n = 3$ biologically independent experiments. Cell viability curves are shown in Supplementary Fig. 1c. **e** Heatmap depicting the IC50 values of sotorasib and exarafenib against a comprehensive panel of Ba/F3 cell lines engineered to express different RAS mutations. IC50 values were calculated from data of $n = 3$ biologically independent experiments. Cell viability curves are shown in Supplementary Fig. 1d. Source data are provided as a Source data file.

due to the lack of effective targeted therapy options. These findings highlight BRAF Class II and III mutants as a substantial clinical patient cohort with aggressive disease that represents an important unmet therapeutic need.

## The novel pan-RAF inhibitor exarafenib potently and selectively inhibits RAF proteins

Current approved BRAF V600X inhibitors are less effective against other forms of mutant BRAF[4,21] and paradoxically activate wild-type isoforms, a phenomenon where mutant-selective BRAF inhibitors induce wild-type RAF heterodimerization (e.g., BRAF:CRAF) and transduce MEK-ERK pathway activity[23,24]. These limitations led to the development of pan-RAF inhibitors, which can target various RAF isoforms, including CRAF and ARAF, to overcome these limitations[25–27]. Exarafenib, a novel pan-RAF inhibitor, was discovered through a distinct structure-based drug design strategy to target the αC-helix-IN and DFG-out conformations of RAF kinases[20]. This strategic innovation allows exarafenib to inhibit both monomeric and dimeric forms of RAF[28], ensuring robust cellular potency across a broad range of RAF alterations and excellent pharmacokinetic properties while avoiding paradoxical activation of the pathway.

We evaluated the specificity of exarafenib in ex vivo biochemical assays across a panel of 368 wild-type kinases at 1 μM concentration, as visualized in kinome tree diagrams (Fig. 2a, left)[20]. Exarafenib exhibited potent inhibition of all three RAF isoforms while maintaining minimal off-target effects, with DDR1 being the only wild-type kinase showing >70% inhibition. In this study, we expanded our analysis by comparing exarafenib's specificity to that of other pan-RAF inhibitors, including naporafenib[26,29] and belvarafenib[27,30], which are also under development. In contrast to exarafenib, naporafenib, and belvarafenib displayed a higher number of off-target effects, with 4 and 15 wild-type kinases showing >70% inhibition, respectively, as is evident from the more numerous and larger off-target nodes in their kinome trees (Fig. 2a, middle and right). These findings highlight exarafenib's notable selectivity profile compared to the other clinical pan-RAF inhibitors evaluated, as well as its strong on-target potency.

## Mechanism of action: exarafenib inhibits RAF monomers and dimers

To establish the mechanism of action (MOA) by which exarafenib could exert anti-tumor effects in human cancer cells, we conducted a series of experiments using patient-derived lung cancer cell lines harboring BRAF mutations (Fig. 2b). Here, we focused on two key aspects: (1) the effects of exarafenib on RAF isoform protein interactions, and (2) the extent to which exarafenib inhibited each RAF isoform. Since recent studies have shown that RAF inhibitors can disrupt intramolecular interactions between the kinase domain and its N-terminal regulatory region and thereby induce RAF dimerization to varying degrees depending on their binding mode[23–25,31], we investigated RAF dimerization in the context of elucidating exarafenib's MOA.

To study these protein interactions under RAF inhibitor treatment, we performed immunoprecipitation (IP)-western blot analysis examining endogenous ARAF, BRAF, and CRAF protein complexes in three patient-derived lung cancer cell lines, each representing a different BRAF mutation class: HCC364 (Class I; BRAF V600E), NCI-H2405 (Class II; BRAF L485_P490delinsY), and NCI-H1666 (Class III; BRAF G466V). Upon analysis, despite the distinct biological features of each BRAF mutation class, exarafenib and belvarafenib induced significant RAF heterodimerization across all classes (Fig. 2b). In contrast, vemurafenib, an approved mutant BRAF V600X inhibitor[8], did not induce dimerization under the tested conditions, aligning with reports that it has a lower propensity for inducing RAF dimerization[25,31]. These observed dimerization patterns prompted us to further investigate how exarafenib affects the kinase activity of different RAF isoforms across various BRAF-mutant protein contexts.

Accordingly, we conducted two complementary in vitro kinase assays. First, we treated lung cancer cell lines in situ with the inhibitors for 4 h, followed by protein extraction and subsequent IP-kinase assay (Supplementary Fig. 1a, top). This approach allowed us to evaluate each inhibitor's ability to block RAF kinase activity in a cellular context. Second, we performed a BRAF-IP from untreated cell lysates and then added the inhibitors directly to the kinase reaction ex vivo (Supplementary Fig. 1a, bottom). This latter cell-free assay enabled us to assess the direct inhibitory effects of the inhibitors on BRAF kinase activity, largely independent of cellular uptake or off-target effects. Under both conditions, exarafenib efficiently inhibited BRAF kinase activity (Supplementary Fig. 1b), confirming its ability to enter cancer cells and bind to and selectively inhibit endogenous BRAF. Based on this result and to ensure the physiological relevance of our findings, we used the first experimental design for subsequent experiments. We extended the study to three lung cancer cell lines used in the co-IP experiments and assessed the effects of each RAF inhibitor on endogenous ARAF, BRAF, and CRAF kinase activity. In HCC364 cells, all three inhibitors effectively inhibited BRAF kinase activity (Fig. 2c, left). As expected, only exarafenib and belvarafenib inhibited BRAF and CRAF kinase activity in Class II NCI-H2405 cells (Fig. 2c, middle). In Class III NCI-H1666 cells, the effects on BRAF kinase activity were modest (Fig. 2c, right), likely due to the weak or absent kinase activity of the mutated BRAF typically found in Class III cell lines[5]. However, baseline CRAF kinase activity was strongly detected in NCI-H1666 cells and was significantly inhibited by exarafenib and belvarafenib (Fig. 2c, right), aligning with previous work showing that these mutants are CRAF-dependent in activating the downstream MAPK pathway[5]. Interestingly, endogenous ARAF kinase activity at an acute timepoint remained largely unaffected by drug exposure, contrasting with inhibition observed in the cell-free biochemical assay (Fig. 2a). This divergence likely reflects assay-context differences: the cell-free system maximizes active-site accessibility and catalytic output, whereas the endogenous IP-kinase assay may preserve native protein complexes (including scaffold-mediated interactions and dimer composition) and is further constrained by

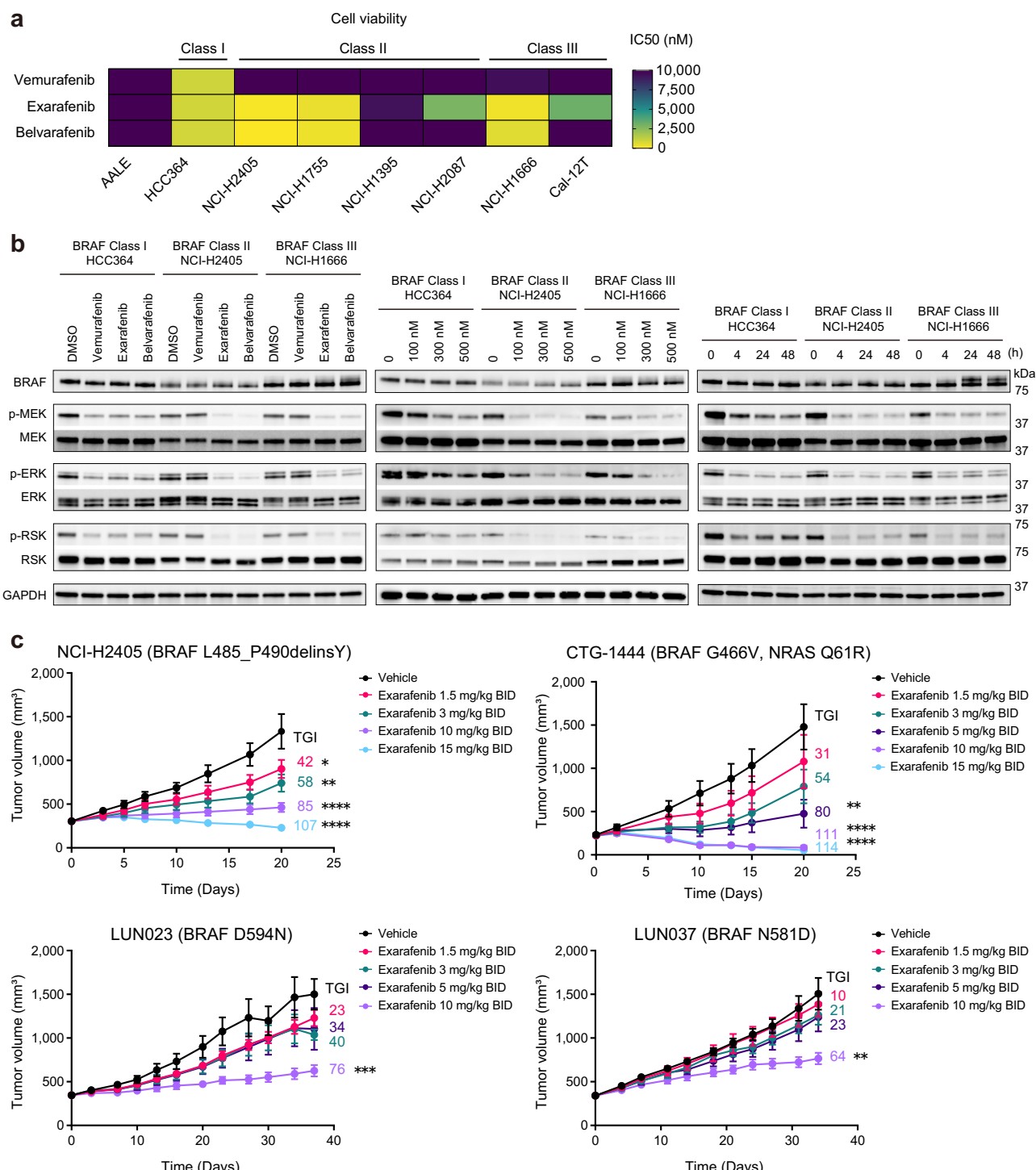

ARAF's low basal catalytic activity[32], limiting the dynamic range for detecting acute inhibition. These results demonstrate that although exarafenib induces RAF dimerization, it can directly target the dimerized RAF isoforms and effectively inhibit RAF kinase activity across the BRAF mutation classes.

## Exarafenib shows potent effects in BRAF-mutated isogenic Ba/F3 cells

To assess the activity of exarafenib on cell viability specifically driven by BRAF mutations in an isogenic background, we employed Ba/F3 cell models engineered to express Class I and Class II BRAF mutations that exhibit strong and readily detectable kinase activity. These models allow for a uniform cellular background, enforcing that the observed effects are due to the BRAF mutations and the inhibitors tested[33]. It is important to note that generating Ba/F3 models with BRAF Class III mutations is challenging due to the low or absent kinase activity[5], limiting our ability to assess exarafenib's effects on this class of mutations in this particular context. The results demonstrated that vemurafenib only inhibited cell proliferation in the V600E model, whereas exarafenib and belvarafenib displayed potent effects against all series of BRAF mutations tested (Fig. 2d and Supplementary Fig. 1c).

Given that oncogenic RAS mutations are upstream activators of RAF signaling and represent another major oncogenic driver in NSCLC, we sought to evaluate exarafenib's therapeutic potential beyond BRAF-

**Fig. 3 | Exarafenib demonstrates broad efficacy in BRAF-mutant lung cancer models. a** Cell viability assay comparing the effects of vemurafenib, exarafenib, and belvarafenib in patient-derived lung cancer cell lines with BRAF mutations from each functional class and the non-cancerous bronchial epithelial cell line AALE. Heatmap presents IC50 values (nM), with lower values (yellow) indicating higher potency and higher values (purple) indicating lower potency. Exarafenib and belvarafenib demonstrated efficacy against Class II and Class III cells to varying degrees, while vemurafenib is only effective in Class I. IC50 values were calculated from data of $n = 3$ biologically independent experiments. Cell viability curves are shown in Supplementary Fig. 2a. **b** MAPK signaling in BRAF-mutant cell lines treated with RAF inhibitors. Left: Various inhibitors at 300 nM for 4 h; exarafenib inhibited signaling across all BRAF classes, while vemurafenib only affected Class I. Middle: Dose-dependent inhibition by exarafenib (4-h treatment). Right: Time-dependent inhibition by exarafenib (300 nM). Exarafenib demonstrates dose- and time-dependent inhibition of MAPK signaling across all BRAF classes. Data are representative of three independent experiments with similar results. **c** In vivo efficacy of exarafenib in the NCI-H2405 (BRAF L485_P490delinsY) CDX model and PDX models CTG-1444 (BRAF G466V), LUN023 (BRAF D594N), and LUN037 (BRAF N581D). Mice were treated with exarafenib at various doses twice daily (BID). Tumor growth inhibition (TGI) was observed dose-dependently, with TGI values shown as percentages next to each dosage curve. Data represent mean ± s.e.m. ($n = 9$ mice per group; one animal in the CTG-1444 vehicle group was excluded from the analysis due to spontaneous tumor regression). *P*-values were calculated by one-way ANOVA followed by Bonferroni correction for multiple comparisons. Significant tumor growth inhibition compared to vehicle was observed in NCI-H2405 xenograft at 1.5 mg/kg (*$P = 0.04$), 3 mg/kg (**$P = 0.0025$), 10 mg/kg, and 15 mg/kg (****$P < 0.0001$); in CTG-1444 at 5 mg/kg (**$P = 0.0037$), 10 mg/kg, and 15 mg/kg (****$P < 0.0001$); in LUN023 at 10 mg/kg (***$P = 0.0004$); and in LUN037 at 10 mg/kg (**$P = 0.0022$). *$P < 0.05$, **$P < 0.01$, ***$P < 0.001$, ****$P < 0.0001$. Source data are provided as a Source data file.

mutated cancers. Using the same system, we conducted a screen of a series of oncogenic KRAS and NRAS mutations. We evaluated exarafenib's effects along with the selective KRAS G12C inhibitor sotorasib, which is an approved treatment for KRAS G12C mutant NSCLC[34], for benchmarking against KRAS G12 cells. Notably, we observed favorable exarafenib sensitivity across multiple oncogenic RAS mutations (Fig. 2e and Supplementary Fig. 1d), suggesting the potential utility of using emerging pan-RAF inhibitors in a wider range of cancer types driven by these oncogenic RAS mutations.

## Exarafenib inhibits BRAF-mutated and RAS-mutated lung cancer cells

Next, we evaluated the inhibitory effects of exarafenib on cell proliferation using NSCLC cell lines with endogenous oncogenic BRAF mutations. We used the non-cancerous bronchial epithelial cell line, AALE, to determine potential non-specific cellular effects of exarafenib[35]. Vemurafenib, a mutant-specific Class I BRAF inhibitor, showed excellent sensitivity in cell lines with BRAF V600E mutations but had little effect on cells with Class II or Class III mutations (Fig. 3a and Supplementary Fig. 2a). In contrast, exarafenib and belvarafenib demonstrated efficacy against Class II and Class III cancer cells, albeit to varying degrees (Fig. 3a and Supplementary Fig. 2a). It is worth noting that some cell lines exhibited lower sensitivity to these compounds, with IC50 values above 1 µM. Specifically, NCI-H1395, NCI-H2087, and Cal-12T showed IC50 values of 9441 nM, 2935 nM, and 3257 nM, respectively, for exarafenib. This variation in sensitivity within each class likely reflects the complexity of clinically relevant lung cancer cells and is an important preclinical model of the heterogeneity of real-world clinical cases, where genetic co-alterations and/or non-genetic mechanism(s) may modulate response to targeted therapies. Next, we investigated the MAPK pathway signaling status in cell lines from each class. The results revealed that exarafenib inhibited the components of this pathway in a dose-dependent and time-dependent manner (Fig. 3b). Western blot analyses under the same conditions in lower exarafenib sensitivity cells revealed more modest inhibitory effects on the MAPK pathway compared to the sensitive cells (Supplementary Fig. 2b), potentially due to compensatory or redundant signaling mechanisms, among other possible explanations.

Additionally, we extended our analysis to lung cancer cell lines harboring upstream KRAS mutations to assess whether the Ba/F3 findings translate to more clinically relevant cellular contexts. We found that exarafenib exhibited notable cell proliferation inhibitory effects in several KRAS mutant cell lines and highlight those in particular with non-KRAS G12C mutations that currently lack approved direct RAS inhibitors (Supplementary Fig. 2c). The sensitivity of exarafenib against non-KRAS G12C mutations suggests a broader spectrum of exarafenib activity that warrants further investigation for some of the RAS mutations.

## Exarafenib is potent against a BRAF V600E p61 splice variant that confers resistance to BRAF Class I inhibitors

Beyond the intrinsic BRAF molecular subtypes, we also investigated exarafenib's activity in a model of acquired resistance to the Class I mutant BRAF inhibitor vemurafenib in NSCLC cells. Vemurafenib-resistant subclones, such as HCC364-VR1, were previously generated by continuous vemurafenib exposure in BRAF V600E-containing HCC364 cells[36]. HCC364-VR1 derivatives express a BRAF V600E p61 splice variant that functions as a dimerized BRAF, resulting in insensitivity to agents such as vemurafenib[36], which can only bind to monomers[37]. In HCC364-VR1 cells, exarafenib showed potent cell growth inhibition compared to vemurafenib, with IC50 values of 721 nM and 7551 nM, respectively (Supplementary Fig. 2d). Further, both inhibitors suppressed p-MEK, p-ERK, and p-RSK levels in parental HCC364 cells, but only exarafenib demonstrated robust inhibition in the p61 splice-variant expressing cells (Supplementary Fig. 2e). These findings suggest that exarafenib may have potential utility in addressing acquired resistance to vemurafenib in patients harboring the BRAF V600E p61 splice variant.

## Exarafenib shows robust PK/PD and tumor suppression in vivo

Given its promising in vitro activity, exarafenib was advanced into in vivo studies focusing on lung cancer, using cell line-derived xenograft (CDX) and patient-derived xenograft (PDX) models of BRAF-mutated human cancers. These included the Class II cell line-derived xenograft (CDX) model NCI-H2405 and the Class III patient-derived xenograft (PDX) models CTG-1444, LUN023, and LUN037. Based on our previous study testing exarafenib in pancreatic ductal adenocarcinoma (PDAC) and melanoma xenograft models[20], we administered the drug twice daily (BID), replicating the prior effective and safe protocol. Compared to the vehicle-treated group, exarafenib treatment led to dose-dependent tumor growth inhibition in all models (Fig. 3c), and all dose levels were well-tolerated (Supplementary Fig. 3a). In vivo pharmacokinetics/pharmacodynamics (PK/PD) relationships were measured using exarafenib in a cohort of BALB/c nude mice bearing Class II BRAF-mutant NCI-H2405 tumors treated with 3 or 10 mg/kg BID and observed for up to 72 h after the last dose. By monitoring the kinetics of plasma concentrations, we detected a rapid increase in drug concentration after administration, followed by a gradual decrease over time (Supplementary Fig. 3b, blue line). Additionally, exarafenib demonstrated a dose-dependent suppression of p-ERK, with the 10 mg/kg dose achieving a more pronounced maximum suppression (-80%) compared to the 3 mg/kg dose (-60%) at the 1-h timepoint (Supplementary Fig. 3b, purple line). Although p-ERK levels gradually rebounded, reaching nearly baseline by 24 h, the twice-daily (BID) dosing regimen ensures sustained target engagement and optimal therapeutic efficacy, while maintaining tolerability (Supplementary Fig. 3a). Next, examination of the correlation between the plasma

concentration of unbound exarafenib and p-ERK inhibition revealed an in vivo $EC_{50}$ of 14.67 nM (Supplementary Fig. 3c). To further confirm the pharmacodynamic effects, western blot analysis was performed in NCI-H2405 tumor samples. The results showed dose-dependent inhibition of both p-MEK and p-ERK levels in tumors treated with 3 mg/kg and 10 mg/kg BID doses of exarafenib compared to the vehicle control (Supplementary Fig. 3d). These data demonstrate that exarafenib exhibits potent in vivo effects and pharmacodynamic modulation with achievable tolerability.

The efficacy observed across our PDX models may have broader clinical implications. Notably, the CTG-1444 PDX model used in our study harbors both BRAF G466V and NRAS Q61R mutations and demonstrated sensitivity to exarafenib treatment. Analysis of our patient cohort revealed that concurrent BRAF and RAS mutations occur in a substantial proportion of NSCLC patients: 9.0% in Class I, 20.7% in Class II, and 21.5% in Class III (Supplementary Fig. 3e; specific mutations and their frequencies are provided in Supplementary Data 2). The higher co-occurrence rates in Class II and III patients are particularly noteworthy, as BRAF-RAS co-mutations are traditionally considered rare or mutually exclusive in other cancer types, such as colorectal cancer, where they occur in <1% of cases[38]. In NSCLC, however, several genomic studies have shown that non-V600E BRAF mutations, particularly those in Class II and III, are often accompanied by KRAS alterations[12,39,40], in line with our findings. Combined with our in vitro findings showing exarafenib's activity against RAS-driven signaling, PDX results suggest that pan-RAF inhibition may offer therapeutic potential for the substantial subset of NSCLC patients harboring concurrent BRAF-RAS mutations, though clinical validation will be essential.

## Further identifying potential biomarkers of exarafenib efficacy and resistance in BRAF-mutant lung cancer

Exarafenib exhibited promising efficacy, PK/PD relationships, and tolerability in our preclinical studies. However, as a pan-RAF inhibitor, we reasoned that it is important to consider potential biomarkers and targets that may impact on therapeutic efficacy and/or mediate resistance. This line of investigation could enable identification of potential rational combination therapies to enhance efficacy and maintain or improve safety, similar to the current standard of care for BRAF Class I inhibitors with combination BRAF and MEK inhibitor treatment which shows heightened efficacy safely in patients[41,42]. Towards this end, we leveraged the established approach of generating and profiling exarafenib resistant cell line derivatives from parental treatment sensitive cells through long-term dose-escalation exposure to exarafenib for over three months (NCI-H2405-ER, NCI-H1755-ER, and NCI-H1666-ER; ER: Exarafenib Resistant derivative) (Fig. 4a). The resistant cell lines demonstrated significantly increased IC50 values, with more than 5-fold changes observed in all resistant cells compared to their parental cell line ancestors, and exhibited cross-resistance to another pan-RAF inhibitor, belvarafenib (Fig. 4b and Supplementary Figs. 2a and 4a). Furthermore, under continuous exposure to exarafenib (1 µM), the acquired resistant cells exhibited cell proliferation capabilities roughly equivalent to those of the parental cells (Supplementary Fig. 4b). Whole Exome Sequencing (WES) analysis of the resistant cells did not identify any genomic alterations previously reported to contribute to resistance, or that could be linked to resistance in an obvious manner[43,44].

To characterize the resistant cells further, we next investigated the activation status of MAPK pathway signaling in these resistant cell lines. The resistant cell lines, which were maintained under 1 µM exarafenib exposure, showed relatively sustained levels of p-MEK, p-ERK, and p-RSK compared to their parental counterparts treated with the same concentration of the drug (Supplementary Fig. 4c). To elucidate the mechanism of this sustained MAPK signaling activation, we analyzed pathway dynamics under varying doses and timepoints.

Given the known crosstalk between MAPK and PI3K/AKT pathways in cancer signaling and resistance mechanisms[45,46], we also extended our analysis to include p-AKT levels. Additionally, for all subsequent experiments using these resistant cell lines, unless otherwise noted, we implemented a 48-h drug-free period before each experiment to model consistent baseline conditions between the resistant and sensitive cell lines. At a relatively acute time point (4 h post-drug exposure following the drug-free period), all resistant cell lines showed dose-dependent inhibition of the MAPK pathway upon re-treatment with exarafenib (Supplementary Fig. 4d). However, under time-varying conditions, NCI-H2405-ER and NCI-H1666-ER cells exhibited a rebound in p-MEK, p-ERK, and p-RSK levels over time (Fig. 4c). In contrast, NCI-H1755-ER cells initially showed modest inhibition of MAPK signaling but maintained a relatively high activation state. Furthermore, all resistant cell lines demonstrated time-dependent increases in p-AKT levels (Fig. 4c).

To better understand these complex temporal dynamics and gain deeper mechanistic insights into MAPK and PI3K/AKT pathway behavior during drug withdrawal and re-challenge, we conducted comprehensive time-course analyses across both drug withdrawal and re-challenge periods. Following 72 h of exarafenib treatment, NCI-H2405-ER cells were subjected to drug-free periods of varying durations (−4 h, −24 h, −48 h) followed by drug re-exposure for up to 48 h (+4 h, +24 h, +48 h) (Supplementary Fig. 5a). During drug withdrawal, MAPK pathway components (p-MEK, p-ERK, p-RSK) showed progressive elevation, reaching levels substantially higher than those observed during the 72 h drug treatment. Conversely, p-AKT levels decreased during the drug-free period. Upon drug re-challenge, MAPK pathway components exhibited the previously observed pattern of initial suppression followed by reactivation, though the rebound levels remained moderate compared to the drug-free state. Similarly, p-AKT levels increased upon drug re-exposure. Additionally, we performed parallel cell proliferation analyses by cell counting under the same experimental conditions during drug withdrawal and re-challenge periods (Supplementary Fig. 5b). The results demonstrate that resistant cells maintain their proliferative capacity throughout the drug re-challenge period, confirming that the transient MAPK pathway suppression does not translate to meaningful growth inhibition.

Notably, this continuous proliferative capability throughout the drug withdrawal and rechallenge contrasts with the drug-dependent growth phenotype often observed with MAPK pathway inhibitor resistance, where cells become dependent on continued drug exposure to prevent excessive MAPK activation and subsequent cell death[47,48]. To investigate whether our resistant cells might exhibit such drug dependence, we examined cellular stress markers (Supplementary Fig. 5a). Western blot analysis across withdrawal timepoints (−4 h, −24 h, −72 h) showed no significant induction of DNA damage markers (p-γH2AX and p-CHK2), apoptosis marker (cleaved PARP), or pro-apoptotic protein (BIM). While some markers showed modest fluctuation (such as p-γH2AX elevation comparable to 72-h treatment levels), these changes did not indicate significant cellular stress or death, as indicated by our cell proliferation analyses (Supplementary Fig. 5b). The absence of such drug dependence in our exarafenib-resistant cells suggests a distinct resistance mechanism that enables cells to tolerate both drug withdrawal and re-exposure while maintaining viability and proliferative capacity.

These findings collectively demonstrate that exarafenib resistance involves adaptive mechanisms characterized by dynamic pathway remodeling rather than simple target mutation. These signaling phenotypes in both the MAPK and PI3K/AKT pathways suggested the potential involvement of upstream pathway activation, likely through the key immediate upstream signaling node RAS, which we hypothesized might re-activate either or both of these pathways to promote resistance.

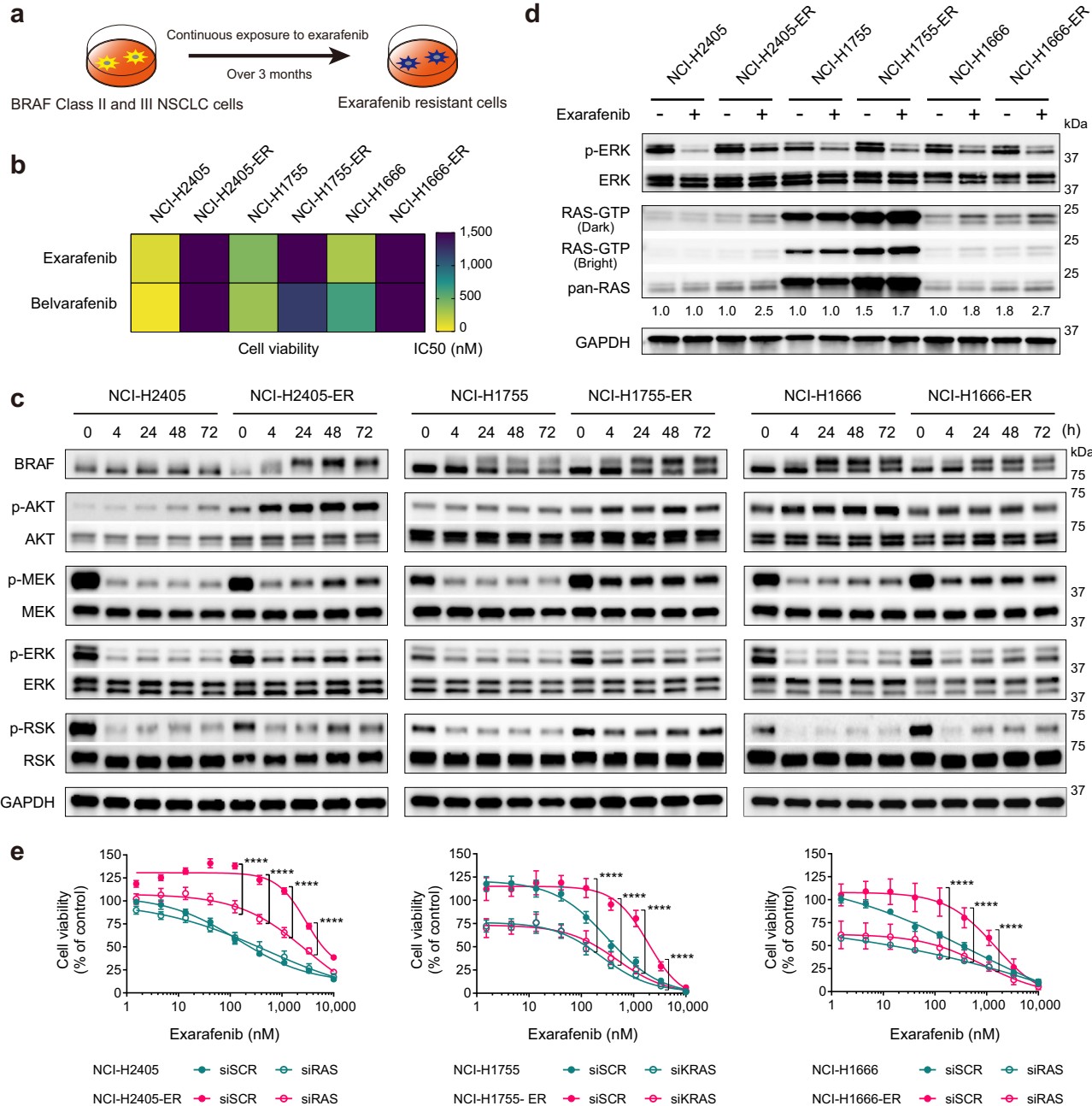

**Fig. 4 | RAS activation contributes to exarafenib resistance in BRAF-mutant lung cancer cell lines. a** Schematic of exarafenib-resistant cell lines (NCI-H2405-ER, NCI-H1755-ER, and NCI-H1666-ER) generation through long-term dose-escalation exposure. **b** Cell viability assay comparing exarafenib and belvarafenib sensitivity in parental and resistant cell lines. Heatmap presents IC50 values (nM), with lower values (yellow) indicating higher potency and higher values (purple) indicating lower potency. For exarafenib, IC50 values increased from 114 nM to 14,060 nM in NCI-H2405 cells, from 422 nM to 2139 nM in NCI-H1755 cells, and from 278 nM to 1513 nM in NCI-H1666 cells. The cells also exhibited cross-resistance to belvarafenib, with IC50 values increasing from 56 nM to 68,171 nM in NCI-H2405 cells, from 187 nM to 1181 nM in NCI-H1755 cells, and from 870 nM to 5100 nM in NCI-H1666 cells. IC50 values were calculated from data of $n = 3$ biologically independent experiments. Cell viability curves are shown in Supplementary Fig. 2a and Supplementary Fig. 4a. **c** Time course analysis of MAPK and AKT signaling in parental and resistant lines treated with 500 nM of exarafenib. NCI-

H2405-ER and NCI-H1666-ER cell lines exhibit a rebound in MAPK signaling, while NCI-H1755-ER cells maintain high pathway activation. Data are representative of three independent experiments with similar results. **d** RAS-RBD assay assessing RAS activation (RAS-GTP) in parental and resistant lines treated with exarafenib. Cells were treated with either DMSO or 500 nM exarafenib for 4 h. Data are representative of three independent experiments with similar results. **e** Cell viability assay evaluating RAS knockdown on exarafenib sensitivity in parental and resistant lines. Cells were transfected with control (siSCR) or RAS-targeting siRNA before exarafenib treatment. Data are mean ± s.d. of $n = 3$ biologically independent experiments. $P$-values were calculated at selected points of interest (two-sided Student's $t$-test). For NCI-H2405-ER, $P = 1.04 \times 10^{-9}$****, $4.02 \times 10^{-12}$****, $7.99 \times 10^{-12}$****, $3.41 \times 10^{-12}$**** (from left to right). For NCI-H1755-ER, $P = 5.57 \times 10^{-9}$****, $2.46 \times 10^{-12}$****, $1.75 \times 10^{-9}$****, $7.98 \times 10^{-7}$****. For NCI-H1666-ER, $P = 2.55 \times 10^{-6}$****, $2.10 \times 10^{-9}$****, $3.42 \times 10^{-7}$****. *$P < 0.05$, **$P < 0.01$, ***$P < 0.001$, ****$P < 0.0001$. Source data are provided as a Source Data file.

To test this hypothesis, we next performed cellular RAS activation state profiling (RAS-GTP detection via RAS-Ras-Binding-Domain [RBD] assays[49]). These studies revealed that RAS activation was induced in NCI-H2405-ER cells after 4 h of exarafenib treatment (Fig. 4d). In the case of NCI-H1666 cells, which are BRAF Class III mutant and generally sensitive to ERK-mediated feedback[5], the parental cells also showed an increase in RAS-GTP levels upon exarafenib treatment (Fig. 4d). However, the resistant cells exhibited stronger baseline activation and response to exarafenib (Fig. 4d). Notably, NCI-H1755-ER cells displayed a marked increase in RAS-GTP levels at baseline, consistent with the high activation of p-MEK, p-ERK, and p-RSK levels shown in Fig. 4c and Supplementary Fig. 4d. Upon exarafenib treatment, the NCI-H1755-ER cells showed a slight further increase in RAS-GTP, whereas parental cells showed no significant change (Fig. 4d). Moreover, the same experimental condition shown in Supplementary Fig. 4c, where the resistant cell lines were maintained under 1 µM exarafenib exposure, also demonstrated elevation of RAS-GTP, suggesting that this effect is sustained during prolonged and continuous drug administration (Supplementary Fig. 5c).

To investigate the biological ramifications of the robust RAS activation present in the resistant cells, we evaluated the effect of pan-RAS knockdown in cell viability assays. Parental NCI-H2405 cells did not show any increase in exarafenib sensitivity following RAS knockdown (Fig. 4e, left). In contrast, resistant NCI-H2405-ER cells exhibited an increase in sensitivity to exarafenib upon RAS silencing, compared to their non-silenced counterparts (Fig. 4e, left). RAS knockdown alone did not affect either the parental or resistant cells in this model (Fig. 4e, left). In NCI-H1755 cells, RAS knockdown demonstrated an additional inhibitory effect on cell viability in both parental and resistant cells, beyond exarafenib alone (Fig. 4e, middle). NCI-H1755 cells harbor the BRAF G469A mutation, which is classified as a Class II BRAF mutation and considered RAS-independent; yet, both parental and resistant cells exhibited a marked dependence on RAS for cell proliferation, even in the absence of drug treatment (Fig. 4e, middle). This observation may challenge the conventional understanding of Class II BRAF mutations and suggests that the relationship between BRAF mutation class and RAS dependence may be more complex than previously thought, particularly in the context of clinically relevant cancer cell lines. Notably, due to the strong RAS dependence in NCI-H1755 cells, our knockdown experiments in this cell line were limited to individual KRAS knockdown, as simultaneous silencing of multiple RAS isoforms was not feasible. In NCI-H1666 cells, RAS knockdown alone suppressed cell viability in both parental and resistant cells, while exarafenib treatment further enhanced this effect (Fig. 4e, right), aligning with the fact that these cells harbor a RAS-dependent BRAF Class III mutation. Taken together, these studies revealed that RAS silencing led to more pronounced inhibition of cell proliferation in each resistant cell line compared to their non-silenced counterparts (Fig. 4e). Furthermore, this effect was more substantial in resistant cells than in parental cells, suggesting that resistant cells have high dependency on RAS activation.

Next, protein expression analysis was performed to assess the changes in signaling upon RAS knockdown. In all resistant cells, RAS knockdown combined with exarafenib treatment resulted in more pronounced MAPK signaling inhibition compared to the scrambled control (SCR) (Supplementary Fig. 5d). In contrast, RAS knockdown showed minimal impact on p-AKT levels (Supplementary Fig. 5d), highlighting the complex nature of signaling pathway regulation and interaction in cancer cells. These results suggest that RAS activation may primarily promote the activation of the MAPK pathway, which appears to be a critical signaling cascade in the context of exarafenib resistance.

## The potential role of receptor tyrosine kinases (RTKs) in exarafenib resistance

To investigate the potential mechanism for elevated RAS-MAPK pathway signaling in the exarafenib resistant derivative cell lines, we performed phospho-RTK arrays comparing parental and exarafenib resistant cells. This approach allowed us to profile activation of upstream RTKs that are well-known to promote RAS and MAPK pathway signaling in various contexts[36,50,51]. While prolonged exarafenib treatment may lead to increased RAS activation through multiple mechanistic factors, such as relieving the negative feedback on the MAPK pathway[52], our primary goal was to identify druggable strategies to enhance exarafenib's efficacy. Thus, we reasoned that examining RTK activation status was a useful first step. The results revealed activation of multiple phospho-RTKs in NCI-H2405-ER and NCI-H1755-ER cells, including strong phosphorylation of AXL and MET, and less significant RTK increases in NCI-H1666-ER cells (Supplementary Fig. 6a).

Based on these findings, we first examined whether the RTKs contribute to the elevated RAS activation observed in resistant cells. To test this hypothesis, we performed knockdown experiments in NCI-H2405-ER cells, selecting AXL as the top hit from our phospho-RTK array, along with EGFR, based on previous reports suggesting EGFR's involvement in the development of resistance to RAF inhibitors[36,50,51]. Individual knockdown of AXL and EGFR showed modest reductions in RAS-GTP levels, while combined knockdown demonstrated a more pronounced reduction in RAS-GTP levels (Supplementary Fig. 6b). The greater efficacy of combined versus individual RTK knockdown may indicate functional redundancy among multiple RTKs in maintaining elevated RAS-GTP levels in resistant cells.

Given the established connection between RTK activation and RAS-GTP levels, we next conducted functional studies across all resistant cell lines to assess the biological consequences of RTK silencing. We performed knockdown experiments targeting AXL and MET, which exhibited the strongest upregulation in NCI-H2405-ER and NCI-H1755-ER cells, respectively, including EGFR as described above. The knockdown of individual RTKs did not substantially impact sensitivity to exarafenib, with the only exception of EGFR knockdown in NCI-H2405-ER cells, which modestly enhanced sensitivity (Supplementary Fig. 6c). Similarly, in western blot analysis, EGFR knockdown in NCI-H2405-ER cells only modestly inhibited p-ERK and p-RSK (Supplementary Fig. 6d). AXL knockdown in NCI-H2405-ER cells did not affect MAPK signaling but significantly reduced p-AKT levels. However, the contribution of AXL-induced AKT activation to drug sensitivity appears to be limited, as AXL knockdown had little effect on cell viability (Supplementary Fig. 6c).

Since individual RTK knockdowns had limited effects in sustaining the MAPK signaling, we next investigated whether resistant cells were dependent on extracellular growth factor stimulation, which could coordinate multiple RTK activities. Analysis under varying serum conditions revealed that MAPK signaling in resistant NCI-H2405-ER cells was strongly serum-dependent, with higher serum concentrations maintaining robust p-ERK and p-RSK activation even in the presence of exarafenib (Supplementary Fig. 6e). Notably, p-AKT levels showed minimal changes under these conditions. By comparison, parental NCI-H2405 cells showed no difference from low serum conditions in maintaining MAPK activity under exarafenib treatment (Supplementary Fig. 6e), indicating that the acquired ability to respond to extracellular growth factors is a specific characteristic of the resistant cells rather than an inherent property of the parental cell line.

These findings collectively demonstrate that exarafenib resistant cells exhibit strong dependence on the MAPK pathway for survival and proliferation, with multiple RTKs functioning cooperatively to promote RAS-MAPK pathway activation. The cooperative nature of RTK

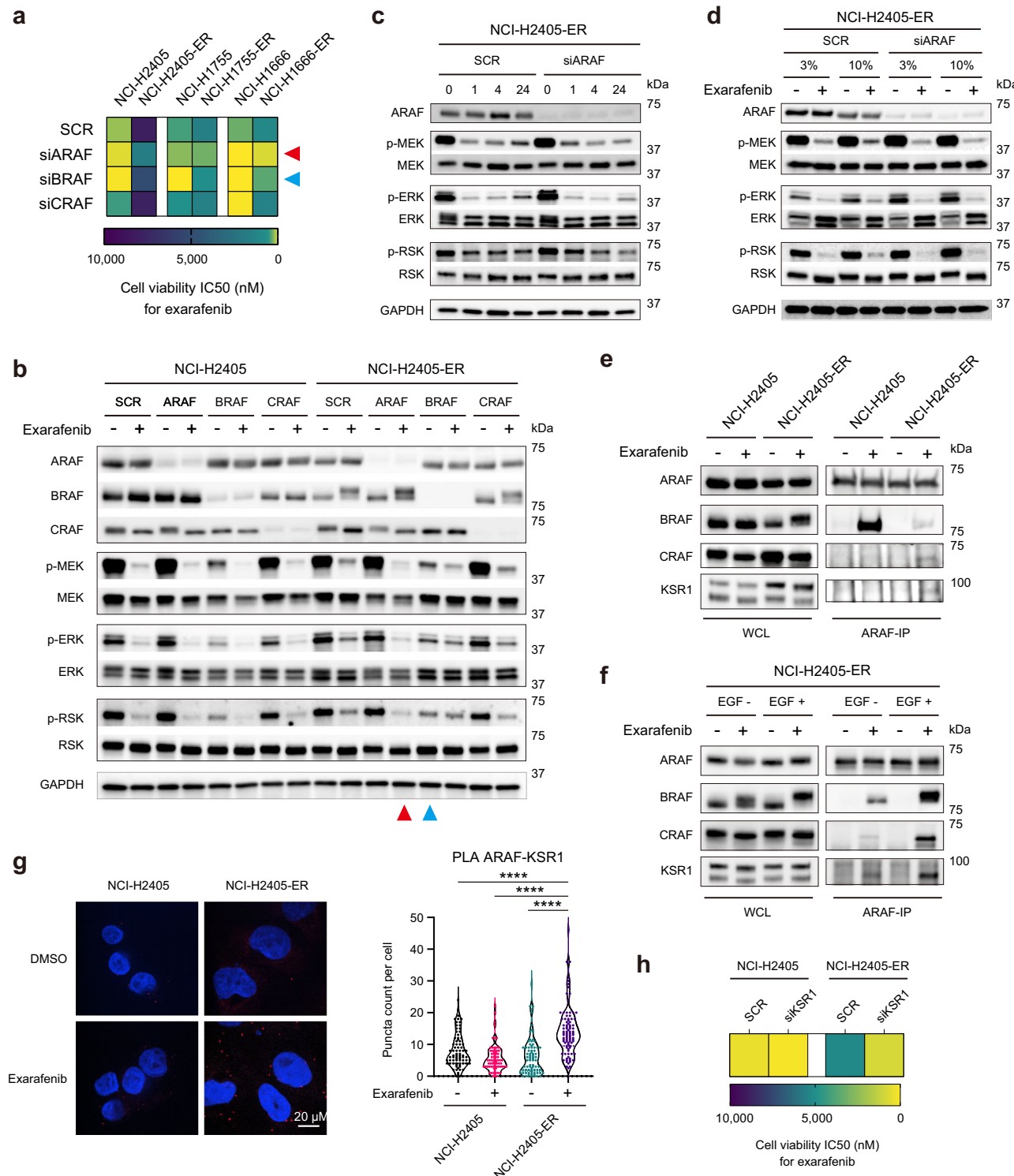

activation, combined with serum-dependent maintenance of pathway signaling, suggests that resistance involves adaptive rewiring of growth factor responsiveness rather than simple upregulation of individual RTKs. These observations are consistent with other reports in the context of targeted therapy resistance[52,53] and highlight the multi-factorial nature of acquired resistance mechanisms.

## Exarafenib resistance involves an adaptive switch to ARAF-dependent MAPK signaling

Our findings demonstrating RAS activation and cooperative RTK signaling in resistant cells prompted us to investigate whether resistant

cells might exhibit altered dependencies on specific RAF isoforms that enable bypass of drug inhibition.

To test this hypothesis, we performed individual RAF isoform knockdown experiments in both parental and resistant cell lines. Cell viability assays revealed that in parental cells harboring Class II BRAF mutants that are hyperactive (NCI-H2405 and NCI-H1755), BRAF knockdown resulted in the greatest reduction of exarafenib IC50 values, as expected (Fig. 5a and Supplementary Fig. 7a). In the Class III BRAF mutant NCI-H1666, both BRAF and CRAF knockdowns contributed to exarafenib IC50 reduction, reflecting the kinase-dead Class III BRAF mutation characteristics. Notably, ARAF knockdown also reduced IC50

**Fig. 5 | Exarafenib resistance involves an adaptive switch from BRAF to ARAF-KSR1 complex-dependent MAPK signaling. a** Cell viability IC50 heatmap showing RAF isoform dependency in parental and exarafenib-resistant cell lines calculated from Supplementary Fig. 7a. NCI-H2405, NCI-H1755, and NCI-H1666 parental and resistant (-ER) cells were transfected with scrambled control (SCR), siARAF, siBRAF, or siCRAF and treated with exarafenib. Lighter shades indicate lower IC50 values and greater sensitivity. Red and blue arrows highlight siARAF and siBRAF conditions, respectively. IC50 values were calculated from data of $n = 3$ biologically independent experiments. **b** Western blot analysis showing differential RAF isoform dependency for MAPK pathway maintenance under drug-free and drug treatment conditions. NCI-H2405 parental and resistant (NCI-H2405-ER) cells were transfected with scrambled control (SCR) or siRNAs targeting ARAF, BRAF, or CRAF, then treated with DMSO (−) or exarafenib (500 nM, +) for 48 h. Red and blue arrows highlight siARAF (under drug treatment) and siBRAF (under drug-free conditions) in resistant cells, respectively. Data are representative of three independent experiments with similar results. **c** Time-course analysis of MAPK pathway reactivation following exarafenib treatment in resistant cells. NCI-H2405-ER cells were transfected with scrambled control (SCR) or siARAF, then treated with exarafenib (500 nM) for the indicated time points (0, 1, 4, 24 h). Data are representative of two independent experiments with similar results. **d** Serum dependency of MAPK pathway reactivation in exarafenib-resistant cells. NCI-H2405-ER cells were transfected with scrambled control (SCR) or siARAF and cultured in low (3%) or high (10%) serum conditions, then treated with DMSO (−) or exarafenib (500 nM, +) for 24 h. Data are representative of two independent experiments with similar results. **e** Co-immunoprecipitation analysis of ARAF protein interactions in parental

and resistant cells. NCI-H2405 parental and NCI-H2405-ER cells were treated with DMSO (−) or exarafenib (500 nM, +) for 24 h. ARAF was immunoprecipitated, and co-precipitating proteins (BRAF, CRAF, KSR1) were detected by immunoblotting. Whole cell lysates (WCL) show input protein levels. Data are representative of two independent experiments with similar results. **f** Co-immunoprecipitation analysis demonstrating growth factor regulation of ARAF-KSR1 interaction. NCI-H2405-ER cells were treated with or without EGF (100 ng/ml) and DMSO (−) or exarafenib (500 nM, +) simultaneously for 24 h. EGF stimulation enhances ARAF-KSR1 interaction in resistant cells under drug treatment conditions. Data are representative of two independent experiments with similar results. **g** Proximity ligation assay (PLA) analysis of ARAF-KSR1 spatial interaction in parental versus resistant cells. NCI-H2405 parental and NCI-H2405-ER cells were treated with DMSO (−) or exarafenib (500 nM, +) for 24 h. PLA signals (red dots) indicate ARAF-KSR1 proximity. Nuclei are stained with DAPI (blue). Scale bar, 20 μm. Quantified data are presented as violin plots. Significance was determined using the Kruskal–Wallis test followed by two-sided Dunn's multiple comparisons test with adjustment for multiple comparisons. ****Adjusted $P < 0.0001$. Data represent at least 100 cells per condition across $n = 3$ independent experiments. **h** Heat map representation of IC50 values for exarafenib treatment in parental NCI-H2405 and resistant NCI-H2405-ER cells following scrambled control (SCR) or KSR1 siRNA (siKSR1) treatment. Lighter shades indicate lower IC50 values and greater sensitivity. IC50 values were calculated from data of $n = 3$ biologically independent experiments. Dose-response curves are shown in Supplementary Fig. 7f. Source data are provided as a Source data file.

values in Class III parental cells, suggesting a previously unreported ARAF dependency in Class III BRAF mutant cells. This unexpected ARAF dependency in Class III mutants, while not the primary focus of our study, represents an intriguing baseline characteristic that warrants further mechanistic investigation in drug-naïve contexts. Moreover, all Class II and III BRAF mutant, exarafenib resistant cell lines consistently showed the greatest IC50 reduction upon ARAF knockdown, demonstrating a clear convergent shift in RAF isoform dependency at acquired resistance (Fig. 5a and Supplementary Fig. 7a).

Protein analysis in NCI-H2405-ER cells provided mechanistic insight into this isoform switch. Under drug-free conditions, only BRAF knockdown reduced p-MEK, p-ERK, and p-RSK levels, indicating BRAF dependency similar to parental cells (Fig. 5b). However, under exarafenib treatment, ARAF knockdown resulted in the most profound suppression (Fig. 5b). These results demonstrate for the first time that resistant cells undergo a switch to an ARAF-dependent MAPK activation maintenance mechanism in the presence of pan-RAF inhibitor treatment.

To further characterize this mechanism in detail, we examined time-dependent signaling changes following pan-RAF inhibitor treatment under ARAF knockdown conditions. Consistent with our earlier observations (Fig. 4c), resistant cells exhibited time-dependent recovery of MAPK activity after drug treatment under control conditions (Fig. 5c). However, cells transfected with ARAF siRNA showed markedly suppressed signal recovery (Fig. 5c). This temporal analysis provides evidence that ARAF mediates the transition from conventional oncogenic-BRAF-dependent signaling (suppressed by exarafenib) to an alternative bypass pathway that enables sustained MAPK activation. Furthermore, we investigated whether this ARAF-mediated pathway reactivation depends on extracellular growth factor signaling by examining MAPK activity under different serum concentrations with and without exarafenib treatment. Resistant cells showed enhanced MAPK pathway reactivation under high serum (10% FBS) conditions compared to low serum (3% FBS) during drug treatment, while ARAF knockdown abolished this elevated pathway reactivation (Fig. 5d). These serum-dependency experiments demonstrate that the ARAF-mediated bypass mechanism requires upstream RTK-RAS activation specifically in the context of exarafenib treatment, establishing a functional link between growth factor signaling and the ARAF-dependent resistance pathway.

## ARAF-KSR1 complex formation mediates resistance-associated MAPK reactivation

The selective ARAF dependency observed in resistant cells prompted us to investigate the molecular mechanisms by which ARAF maintains MAPK signaling under pan-RAF inhibition. To elucidate ARAF's functional interactions, we examined its binding partners through co-immunoprecipitation analysis. We focused on other RAF isoforms (BRAF and CRAF) to assess potential heterodimerization changes, as differential RAF heterodimerization has been linked to variations in MAPK pathway activation and therapeutic response[27,54,55]. Additionally, we examined KSR1 (Kinase Suppressor of Ras 1), a critical scaffolding protein that organizes RAF-MEK-ERK signaling complexes and has been implicated in facilitating RAF isoform-specific MAPK activation[56]. Notably, KSR1 has been previously implicated in resistance mechanisms to RAF and MEK inhibitors, where it can facilitate alternative signaling pathways that bypass drug inhibition[57,58].

Our co-immunoprecipitation studies reveal striking differences in ARAF protein interaction patterns between parental and resistant cells (Fig. 5e). In resistant cells, ARAF-BRAF interactions were significantly reduced compared to parental cells, indicating a distinct interaction profile from the previously observed exarafenib-induced BRAF-ARAF heterodimerization in drug-sensitive contexts (Fig. 2b). Conversely, exarafenib treatment specifically induced both ARAF-CRAF and ARAF-KSR1 binding in resistant cells but not in parental cells (Fig. 5e). CRAF knockdown in resistant cells did not significantly impact exarafenib sensitivity (Fig. 5a). Therefore, we focused our subsequent mechanistic studies on this new ARAF-KSR1 interaction we found.

We wondered why exarafenib was unable to effectively inhibit ARAF (or CRAF)-MEK signaling in the resistant cells. One possibility is that exarafenib is not able to bind the ARAF-KSR1 or ARAF-CRAF complex due to allosteric or structural effects of increased ARAF-KSR1 protein interactions. Our finding that exarafenib treatment induced the ARAF-KSR1 and ARAF-CRAF binding suggested against this scenario (Fig. 5e). Nevertheless, to more directly address this question, we performed cellular thermal shift assays[59] (CETSA). In parental cells, exarafenib stabilized BRAF, indicating binding, but showed minimal engagement with ARAF, CRAF, and KSR1 (Supplementary Fig. 7b). In contrast, the resistant cells demonstrated enhanced exarafenib engagement with both ARAF and CRAF in addition to the maintained BRAF binding observed in parental cells,

while KSR1 showed minimal exarafenib-induced thermal stabilization (Supplementary Fig. 7b).

Consistent with the upstream growth factor/RTK/RAS dependence we show, EGF stimulation further amplifies this ARAF-KSR1 interaction in resistant cells (Fig. 5f). Conversely, pharmacological inhibition of RAS with RMC-6236, a pan-RAS (ON) inhibitor that targets the active GTP-bound form of both wild-type and mutant RAS isoforms[60], markedly reduced ARAF-KSR1 complex formation, as shown by co-immunoprecipitation (Supplementary Fig. 7c). These reciprocal effects demonstrate that growth factor/RTK/RAS activation controls the ARAF-KSR1 interaction in resistant cells. Supporting this RAS-dependent regulation, EGF-induced enhancement of complex formation coincided with enhanced exarafenib-induced thermal stabilization of all RAF proteins, while KSR1 showed minimal change, as demonstrated by CETSA (Supplementary Fig. 7d). Mechanistically, these results align with recent work showing that RAS activation primes RAF into a dimer-competent αC-IN state, which enhances the affinity of type II RAF inhibitors[55]. Inhibitor binding then further stabilizes this conformation and promotes RAF protein complex formation. Proximity ligation assays (PLA) provide additional validation, demonstrating increased ARAF-KSR1 proximity in resistant cells following exarafenib treatment compared to DMSO control, with less interaction observed in parental cells (Fig. 5g).

To gain structural insights into the exarafenib-binding mode within this complex, we performed AlphaFold-based molecular structural modeling. Docking analysis suggested that exarafenib primarily occupies the ATP-binding site of ARAF within the ARAF–KSR1 complex, contacting the gatekeeper residue (Thr382), residues in the hinge region (Gln383–Cys385), and the DFG motif (Asp447–Gly449), while also making limited contacts with a short KSR1 segment (Pro408–Asn413) that lies adjacent to the pocket edge (Supplementary Fig. 7e and Supplementary Table 7). These observations indicate that exarafenib is structurally compatible with binding at the kinase active site while the ARAF–KSR1 complex retains its scaffolding architecture and generally align with our experimental data discussed above.

To assess the functional significance of KSR1 in maintaining resistance, we performed KSR1 knockdown experiments. KSR1 silencing selectively restored exarafenib sensitivity in resistant cells, with IC50 shifting from 5276 nM to 884 nM while having minimal impact on parental cell viability (Fig. 5h and Supplementary Fig. 7f). Protein analysis revealed that KSR1 depletion further suppressed MAPK pathway activation (p-MEK, p-ERK, p-RSK) in resistant cells following exarafenib treatment, while showing no change in parental cells (Supplementary Fig. 7g). Collectively, these findings establish KSR1 as an essential scaffolding hub that enables ARAF-dependent MAPK reactivation, thereby sustaining exarafenib resistance.

Given that KSR1 functions as a scaffold protein that enhances RAF kinase efficiency[56], we hypothesized that the drug-bound ARAF-KSR1 complex maintains functional signaling capacity by recruiting MEK. Consistent with this hypothesis, MEK immunoprecipitation revealed a marked increase in MEK–KSR1 association in resistant cells under exarafenib treatment, whereas MEK–ARAF complexes were not detected, perhaps due to lack of substantial close physical proximity or the transient nature of the MEK-ARAF protein interaction (Supplementary Fig. 7h). Although weak associations with BRAF and CRAF were detected, these appeared unlikely to play a major role in resistance, as their genetic depletion produced minimal effects on exarafenib sensitivity (Fig. 5a). While exarafenib binding reduces ARAF intrinsic kinase activity, increased upstream RAS activation coupled with enhanced KSR1 scaffolding of the ARAF/KSR1/MEK protein complex in the resistant cells may compensate for this reduction by improving MEK recruitment and positioning, thereby enabling the residual ARAF catalytic activity to sustain MAPK pathway signaling more efficiently despite drug treatment. A prediction of our general model is that supra-physiologic doses of exarafenib should overcome

this mechanism and resistance. Our dose-response analysis supports this model: we show that the resistant cells retain sensitivity to supraclinical exarafenib concentrations and display a sigmoid response curve rather than complete resistance (Supplementary Fig. 7f, siSCR). This pattern indicates that at therapeutic doses, KSR1 scaffolding buffers the loss of significant ARAF activity despite drug binding, but at higher concentrations, the remaining kinase function becomes insufficient to sustain cell viability even with enhanced scaffolding efficiency.

These findings collectively establish a resistance mechanism wherein exarafenib-resistant cells undergo adaptive rewiring from conventional oncogenic-BRAF-dependent signaling to an ARAF-KSR1-dependent bypass pathway. The formation of a drug-bound yet partially active ARAF-KSR1 scaffold complex enables resistant cells to maintain MAPK signaling despite pan-RAF inhibition through enhanced scaffolding complex formation driven by upstream RAS activation. This mechanism underscores the continued RAF dependency in resistant cells, albeit through an alternative configuration that reduces drug efficacy within the therapeutic window.

### Potential effective exarafenib combination therapy strategies

Our mechanistic characterization of exarafenib resistance revealed a complex adaptive rewiring process involving upstream RTK activation, RAS-GTP accumulation, ARAF-KSR1 scaffolding complex formation, and sustained MAPK pathway signaling. While these findings identified specific resistance mechanisms, the cooperative nature of multiple RTKs and the multi-layered architecture of this resistance network suggested that targeting individual RTK components might be insufficient to overcome resistance comprehensively. Therefore, we reasoned that combination strategies targeting downstream convergence points—particularly at the level of RAS or within the MAPK cascade itself—would provide more effective therapeutic intervention. Based on these mechanistic insights, we systematically evaluated potential combination partners for exarafenib, focusing on agents that could disrupt the key signaling dependencies characterized in resistant cells.

As candidate drugs, we first took a direct approach targeting RAS, given its central role in driving RTK-mediated pathway activation and its direct regulation of ARAF-KSR1 complex formation in resistant cells. We selected RMC-6236, the pan-RAS (ON) inhibitor used earlier to demonstrate RAS-dependent regulation of ARAF-KSR1 complex formation (Supplementary Fig. 7c). This broad-spectrum noncovalent inhibitor, which targets the active GTP-bound state of both mutant and wild-type RAS variants[60], is particularly relevant for cell lines like ours lacking a RAS mutation but still exhibiting RAS-driven signaling and resistance through the ARAF-KSR1 scaffolding mechanism. Preclinical data show potent anticancer activity across multiple RAS-addicted cancer models, and early clinical studies (NCT05379985) have demonstrated objective responses in patients with KRASG12X-driven solid tumors[60]. We also included the SHP2 inhibitor RMC-4550, which targets a common hub (SHP2) operating downstream of multiple RTKs to promote RAS-MAPK pathway signaling[61]. Additionally, considering the sustained MAPK signaling maintained through the ARAF-KSR1 mechanism, we selected two MAPK pathway inhibitors to suppress this pathway more potently: the MEK inhibitor binimetinib and the ERK inhibitor ASTX029. These downstream MAPK pathway inhibitors were expected to be particularly effective as they target the final convergence points of both canonical oncogenic-BRAF-dependent signaling and the alternative ARAF-KSR1-mediated bypass pathway. Binimetinib, which is FDA-approved for use in combination with encorafenib for BRAF V600E or V600K mutant melanoma, was chosen for its established clinical efficacy and capability for clinical translation[62,63]. ASTX029, currently in phase I-II clinical trials (NCT03520075), was selected for its novel dual-mechanism ERK inhibition, which offers potent and complete MAPK pathway suppression compared to traditional single-mechanism inhibitors[64,65].

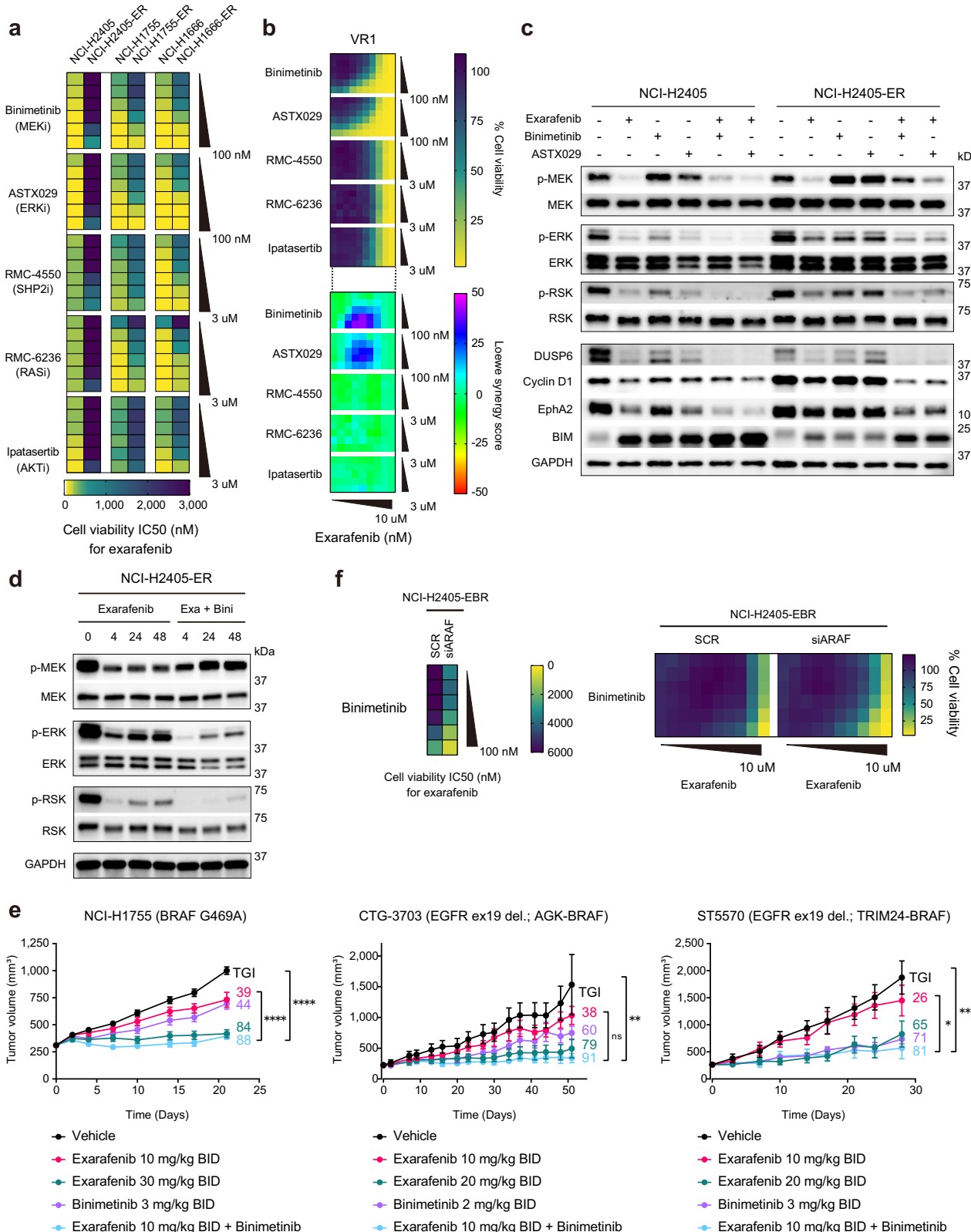

Furthermore, as a fourth candidate, we included the pan-AKT inhibitor ipatasertib[66]. Ipatasertib's mechanism of action and initial encouraging clinical results in multiple solid tumors, including lung cancer, provided additional rationale for inclusion in our studies here to further address the observed increase in p-AKT levels that might play a role in promoting exarafenib resistance apart from the MAPK reactivation we have demonstrated[67] (Fig. 4c and Supplementary Fig. 4d).

To establish a baseline reference for interpreting combination effects and to assess the individual potency of each agent, we first examined the monotherapy results from our drug screening analysis (leftmost column of each heatmap in Supplementary Fig. 8a). Binimetinib and ASTX029 demonstrated varying degrees of efficacy across the tested parental and exarafenib resistant models with the only exception being NCI-H2405-ER cells (Supplementary Fig. 8a, 1st and

**Fig. 6 | Exarafenib and MAPK pathway inhibitor combinations demonstrate synergistic anti-tumor effects in BRAF-mutant NSCLC models. a** Heatmap of IC$_{50}$ values for exarafenib combined with binimetinib, ASTX029, RMC-4550, RMC-6236 or ipatasertib across parental and resistant NSCLC cell lines. Lighter shades indicate lower IC$_{50}$ values and greater sensitivity. IC50 values were calculated from data of $n = 3$ biologically independent experiments. Full heatmap of cell viability is shown in Supplementary Fig. 8a. **b** Dose-response matrix (top) and synergy scores (bottom) for exarafenib combinations in the vemurafenib-resistant HCC364-VR1 cells. Shades indicates percent inhibition of cell viability. Data represent the mean of $n = 3$ biologically independent experiments. Synergy scores were calculated using the Loewe model; positive scores indicate synergy. **c** Western blot analysis of MAPK pathway and apoptotic markers in NCI-H2405 parental and resistant cells treated with exarafenib (500 nM), binimetinib (50 nM), ASTX029 (50 nM), or their combinations for 48 h. Data are representative of three independent experiments with similar results. **d** Time-course western blot analysis demonstrating MAPK pathway reactivation in NCI-H2405-ER resistant cells and the effect of combination therapy. Cells were treated with 500 nM exarafenib alone or in combination with 50 nM binimetinib for the indicated time points (0, 4, 24, 48 h). While exarafenib monotherapy leads to MAPK pathway reactivation over 48 h, combination treatment with binimetinib effectively suppresses this reactivation and maintains pathway inhibition throughout the treatment period. Data are representative of two independent experiments with similar results. Exa, exarafenib; Bini, binimetinib. **e** In vivo efficacy of exarafenib and binimetinib combination in BRAF-mutant NSCLC xenograft models NCI-H1755 (BRAF G469A), CTG-

3703 (AGK-BRAF), and ST5570 (TRIM24-BRAF). Mice were treated with vehicle, exarafenib (10, 20, or 30 mg/kg BID), binimetinib (2 or 3 mg/kg BID), or their combination. Tumor volumes were measured over time. Tumor growth inhibition (TGI) values are shown as percentages next to each dosage curve. Data represent mean ± s.e.m. ($n = 9$ mice per group for NCI-H1755; $n = 6$ mice per group for CTG-3703 and ST5570). P-values were calculated by one-way ANOVA followed by Bonferroni correction for multiple comparisons. In NCI-H1755 tumors, significant tumor growth inhibition was observed with the combination treatment compared to vehicle (****$P < 0.0001$) and exarafenib 10 mg/kg BID monotherapy (****$P < 0.0001$). In CTG-3703 tumors, the combination treatment significantly inhibited tumor growth compared to vehicle (**$P = 0.0043$) but not exarafenib 10 mg/kg BID monotherapy ($P = 0.1151$). In ST5570 tumors, the combination treatment significantly inhibited tumor growth compared to vehicle (**$P = 0.0012$) and exarafenib 10 mg/kg BID monotherapy (*$P = 0.0279$). *$P < 0.05$, **$P < 0.01$, ***$P < 0.001$, ****$P < 0.0001$. **f** Heatmaps showing drug sensitivity analysis in NCI-H2405-EBR cells under scrambled control (SCR) or ARAF siRNA knockdown (siARAF) conditions. Lighter shades indicate lower IC50 values and greater sensitivity. Left: Cell viability IC50 values (nM) for exarafenib in cells treated with varying concentrations of binimetinib (up to 100 nM). Right: Cell viability measurements showing the effects of exarafenib (up to 10 µM) in combination with binimetinib. The IC50 values shown in the left panel were calculated from the dose-response data depicted in the right panel. Cell viability data represent the mean of $n = 3$ biologically independent experiments. Source data are provided as a Source data file.

2nd row). Both RMC-4550 (a SHP2 inhibitor) and RMC-6236 (a RAS (ON) multi-selective inhibitor) monotherapy exhibited anti-tumor activity in Class III BRAF-mutant NCI-H1666 parental and resistant cell lines, consistent with a previous report[61]. In cell lines harboring BRAF Class II mutations, particularly in H1755 parental cells, RMC-6236 monotherapy showed efficacy, which was consistent with the previously demonstrated growth inhibitory effects observed with KRAS knockdown alone (Fig. 4e, middle). In contrast, RMC-4550 monotherapy showed limited efficacy in all tested BRAF Class II cell lines (both parental and exarafenib-resistant H1755 and H2405 cells). While these two agents converge at or near the level of RAS, they showed different results, possibly due to their distinct mechanisms of action: RMC-4550 indirectly affects RAS activation by inhibiting SHP2, whereas RMC-6236 directly inhibits the active, GTP-bound form of RAS. Lastly, ipatasertib alone showed minimal response in most cell lines (Supplementary Fig. 8a, 5th row).

Moving beyond the monotherapy results, the combinational screening data revealed that binimetinib and ASTX029 significantly reduced exarafenib IC50 values and demonstrated robust additional effects compared to exarafenib alone in both parental and exarafenib resistant cell lines, consistently across all BRAF mutation classes (Fig. 6a and Supplementary Fig. 8a, 1st and 2nd row). The broad efficacy of these MEK and ERK inhibitors likely reflects their ability to target the final common pathway outputs from both canonical oncogenic-BRAF-dependent signaling (as observed in treatment-sensitive parental cells) and the alternative ARAF-KSR1 bypass mechanism. Combining either RMC-4550 or RMC-6236 with exarafenib showed efficacy in Class III BRAF-mutant NCI-H1666 parental and exarafenib-resistant cell lines, aligning with the monotherapy findings (Fig. 6a and Supplementary Fig. 8a, 3rd and 4th row). In Class II BRAF-mutant NCI-H2405 cells, while neither RMC-4550 nor RMC-6236 monotherapy had any impact on the parental or exarafenib-resistant line, combining either inhibitor with exarafenib produced an additional anti-tumor cell effect exclusively in the exarafenib-resistant cells, consistent with the observed upstream RAS activation (Fig. 6a and Supplementary Fig. 8a, 3rd and 4th row). In NCI-H1755 cells (also Class II BRAF-mutant cells), combining exarafenib with RMC-4550 or RMC-6236 resulted in additional effect in both the parental and exarafenib-resistant lines, with RMC-6236 showing more pronounced combinatorial effects (Fig. 6a and Supplementary Fig. 8a, 3rd and 4th row). The combination of ipatasertib and

exarafenib showed additional inhibitory effects in Class II BRAF-mutant NCI-H2405 parental cells and Class III BRAF-mutated NCI-H1666 parental and exarafenib resistant cell lines, but the extent was limited compared to binimetinib and exarafenib (Fig. 6a and Supplementary Fig. 8a, 4th row).

Synergy calculations based on the same assay data confirmed the results of the cell proliferation assay, with binimetinib and ASTX029 showing large degrees of synergy with exarafenib in virtually all cell lines (Supplementary Fig. 8b). The distribution of synergy areas at moderate drug concentrations for these two agents suggested promising potential translatability to in vivo models and clinical settings in a feasible and tolerable manner. Furthermore, RMC-6236 exhibited robust synergy, particularly in resistant cells (Supplementary Fig. 8b, 4th row), reflecting the dependence on RAS activation in exarafenib-resistant cells. However, in parental cells, the synergistic effects were limited compared to those of the MEK and ERK inhibitors. Taken together, these differential response patterns, where inhibition of the MAPK pathway via MEK or ERK targeting elicited more pronounced antiproliferative effects compared to AKT inhibition in both parental and exarafenib resistant models, further suggest a predominant dependence on the MAPK pathway in these cellular models (Fig. 6a and Supplementary Fig. 8a, b).

To assess whether these findings could extend beyond exarafenib-resistant lines, we tested the same synergy approach in the previously studied HCC364-VR1 vemurafenib-resistant cell line. These cells harbor a BRAF V600E p61 splice variant that confers resistance to Class I BRAF inhibitors by a mechanism that is largely RAS independent but still promotes downstream MAPK pathway activation[36]. In these cells, significant additional effects and synergy were observed with the combination of binimetinib and ASTX029, confirming the broad spectrum of these two agents against diverse MAPK pathway activation mechanisms (Fig. 6b). In contrast, combining exarafenib with targeting upstream of RAF using RMC-4550 (SHP2) or RMC-6236 (RAS) did not yield significant additional effects or synergy in this BRAF Class I resistant model that achieves MAPK pathway activation in a largely RAS-independent manner (Fig. 6b). This distinct response pattern serves as an important mechanistic control, demonstrating that RAS-targeting agents are specifically effective against resistance mechanisms involving RAS-dependent processes like ARAF-KSR1 complex formation, while MEK/ERK inhibitors retain broad efficacy regardless of the upstream resistance architecture.

## Further validation of the exarafenib-MAPK pathway inhibitor combinations

We examined the changes in MAPK signaling upon binimetinib and ASTX029 combinations with exarafenib. As these drugs target MEK and ERK, respectively, we probed activity markers of the MAPK pathway[68,69], and included BIM, a pro-apoptotic protein that is suppressed by MAPK pathway signaling[70,71]. The concentrations of exarafenib (500 nM), binimetinib (50 nM), and ASTX029 (50 nM) used here were selected based on their high synergistic effects observed in all cell lines during the initial drug screening, and to use doses within clinically relevant levels to enable tolerability[20,62–65,72]. In all three sets of experiments, the drug combination more potently suppressed the pathway in both parental and resistant cells (Fig. 6c and Supplementary Fig. 9a). To further understand the temporal dynamics underlying these combination effects, we performed time-course western blot analysis in NCI-H2405-ER cells (Fig. 6d). Exarafenib monotherapy demonstrated progressive reactivation of MAPK pathway activity over 48 h, recapitulating the temporal rebound pattern described earlier (Fig. 4c). In contrast, combination treatment with binimetinib effectively prevented this reactivation, maintaining pathway inhibition throughout the treatment period (Fig. 6d).

Western blot analysis of the RMC-4550 combination with exarafenib also yielded results consistent with the cell viability assay, with the combination further inhibiting the MAPK pathway in all resistant cells as well as parental NCI-H1755 and NCI-H1666 cells (Supplementary Fig. 9b). Similarly, the combination of RMC-6236 with exarafenib showed enhanced MAPK pathway suppression in both NCI-H1755 and NCI-H1666 parental cells and enhanced inhibition in all resistant cells (Supplementary Fig. 9c). Immunoblotting analysis of the AKT inhibitor ipatasertib combination with exarafenib to measure expression of the MAPK activity markers DUSP6, Cyclin D1, EphA2, and BIM, did not show additional inhibitory effects, as expected (Supplementary Fig. 9d).

These findings suggest that two types of MAPK pathway inhibitors, targeting different nodes of the pathway (MEK, ERK), enhance the therapeutic efficacy of exarafenib in BRAF-mutant lung cancer cells by effectively disrupting both conventional oncogenic-BRAF signaling (predominant in parental cells and non-treated resistant cells) and ARAF-KSR1-mediated bypass signaling (activated in resistant cells under drug treatment). To determine whether this effect is specific to these drugs or can be achieved with other compounds with similar mechanisms of action, we expanded our investigation to additional inhibitors. We first conducted IC50 studies testing several MEK and ERK inhibitors as single agents using NCI-H2405, NCI-H1755, and NCI-H1666 cell lines (Supplementary Fig. 10a). Our aim was to identify compounds with potencies similar to binimetinib and ASTX029, allowing for a more direct comparison of combination effects by minimizing potential confounding factors related to significantly differing drug potencies. From these studies, we selected cobimetinib, a clinically established MEK inhibitor[42,73], and SCH772984, an experimental ERK inhibitor[74], based on their comparable potencies to our initially tested compounds. Subsequent experiments confirmed that cobimetinib, and to some extent SCH772984, enhanced the anti-tumor effects of exarafenib in the resistant cells (Supplementary Fig. 10b, c). These results support the robustness of our findings and suggest that the observed synergy is not strictly limited to specific MEK or ERK inhibitors but is likely to be a broader biological effect across these drug classes targeting the downstream MAPK pathway convergence points.

Among the agents that showed notable synergy with exarafenib, we considered binimetinib as highly promising given its established clinical use and favorable tolerability profile[62,63]. The consistent efficacy of MEK inhibition across all tested models further supports the rationale for targeting downstream convergence points where both conventional oncogenic- BRAF-dependent signaling and ARAF-KSR1-mediated bypass signaling ultimately converge. To further validate the broad applicability of the exarafenib and binimetinib combination, we tested its efficacy in BRAF-mutant Class II and III cell lines that were less responsive to exarafenib monotherapy. These models included the NCI-H1395, NCI-H2087, and Cal-12T cell lines, which we found to be sensitive to the combination of exarafenib and binimetinib, as evidenced by the reduction in cell viability and significant synergy scores (Supplementary Fig. 10d, e). The efficacy of MEK inhibitor combination in these intrinsically less sensitive cells, which may harbor baseline compensatory signaling, suggests that certain patients could benefit from initial combination therapy rather than sequential treatment.

## In vivo efficacy of exarafenib plus binimetinib in NSCLC

To validate the promising combination effects of exarafenib and binimetinib observed in vitro, we evaluated the efficacy of the combination in three patient-derived lung cancer models in vivo, including PDXs: NCI-H1755 (BRAF G469A), CTG-3703 (EGFR exon 19 deletion; AGK-BRAF PDX), and ST5570 (EGFR exon 19 deletion; TRIM24-BRAF PDX) (Fig. 6e). These models represent different BRAF alterations found in lung cancer, including point mutations and gene fusions. Notably, both the CTG-3703 and ST5570 models were derived from EGFR-mutant NSCLC patients who had progressed on EGFR inhibitors. These patients subsequently acquired clinically relevant BRAF fusion proteins, which are known to cause resistance to EGFR-tyrosine kinase inhibitors[75–78]; importantly, we found that exarafenib is effective against certain BRAF fusions in our preclinical profiling using the Ba/F3 system (Fig. 2d and Supplementary Fig. 1c). While we also attempted to establish CDX models using exarafenib-resistant cells, these efforts were unsuccessful due to altered tumorigenic capacity of the adapted cells.

In the NCI-H1755 and ST5570 models, the exarafenib plus binimetinib combination significantly suppressed tumor growth compared to both vehicle control and each monotherapy (Fig. 6e, right and left). Although the difference between the combination and exarafenib monotherapy did not reach statistical significance in the CTG-3703 model, a trend towards enhanced efficacy was also observed (Fig. 6e, middle). Notably, in all cases, the combination achieved tumor suppression that was at least comparable to, and in some cases better than, the higher dose of exarafenib monotherapy, while being well-tolerated, with no significant body weight loss or observed adverse effects (Supplementary Fig. 11a). To further elucidate the MAPK signaling changes induced by the exarafenib and binimetinib combination, we conducted western blot analysis in H1755 xenograft tumor samples at 1- and 7-h post-dosing (Supplementary Fig. 11b). The results revealed strong inhibition of p-ERK by the combination therapy at 1-h post-treatment, with this effect sustained at 7 h post-dosing compared to each monotherapy. Additionally, the combination treatment showed a more pronounced effect on reducing levels of EphA2 and DUSP6, two downstream targets of ERK signaling, further supporting the enhanced pathway inhibition achieved with dual RAF and MEK blockade (Supplementary Fig. 11b).

Although CDX models from exarafenib-resistant cells could not be established due to technical limitations, the activity demonstrated across diverse BRAF-altered models, including clinically relevant PDX models with acquired EGFR inhibitor resistance, validates the broad therapeutic potential of this combination approach. These findings demonstrate exarafenib and binimetinib combination as a promising therapeutic strategy for BRAF-mutant lung cancer, offering superior efficacy and the potential to decrease exarafenib doses to maintain or enhance therapeutic index while eliciting efficacy. Furthermore, evaluating this combination in these models provides valuable insights into possible strategies for overcoming resistance to EGFR inhibitors in EGFR-mutated NSCLC caused by certain BRAF variants.

## Combination therapy resistance reveals sustained ARAF dependency

Given the robust efficacy of exarafenib plus binimetinib combination therapy, we sought to understand potential resistance mechanisms

that might emerge under this dual RAF-MEK blockade. To model this clinically relevant scenario, we generated combination-resistant cells by treating established exarafenib-resistant NCI-H2405-ER cells with gradually escalating doses of binimetinib over 3 months, resulting in NCI-H2405-EBR (Exarafenib-Binimetinib Resistant) cells. Notably, attempts to generate resistance by treating parental cells with both drugs simultaneously were unsuccessful despite 6 months of continuous selection, underscoring the potent synergistic efficacy of upfront combination therapy.

To characterize the molecular mechanisms underlying combination resistance, we first examined RAS activation status, given its central role in monotherapy resistance. RAS-GTP pulldown assays revealed that NCI-H2405-EBR cells exhibited elevated RAS-GTP levels under combination drug treatment compared to parental cells (Supplementary Fig. 11c), mirroring the RAS activation pattern observed in exarafenib monotherapy-resistant cells. This finding indicated that combination-resistant cells retain the ability to elevate RAS-GTP levels under dual RAF-MEK inhibition.

Based on our concurrent findings that ARAF mediates resistance to exarafenib monotherapy through RAS-dependent mechanisms, we investigated whether this mechanism also contributes to combination therapy resistance. Remarkably, ARAF knockdown dramatically restored sensitivity to the drug combination in NCI-H2405-EBR cells, reducing IC50 values in the presence of binimetinib plus exarafenib (Fig. 6f). Western blot analysis further confirmed that ARAF knockdown enhanced MAPK pathway suppression under drug treatment (Supplementary Fig. 11d). These new data suggest that ARAF (or upstream RAS) activation could emerge as a conserved mechanism of resistance clinically. Thus, selective ARAF or RAS inhibition (for instance, with RMC-6236) could overcome multiple resistance layers and may warrant clinical investigation.

### Clinical activity I: exarafenib treatment of a NSCLC patient with TMEM106B-BRAF fusion

Lastly, to validate the clinical relevance of our preclinical findings regarding exarafenib's anti-tumor activity, we examined the clinical outcomes of patients with advanced BRAF-mutant lung cancer treated with exarafenib in the ongoing phase I/Ib clinical trial, KN-8701 (NCT04913285; interim monotherapy results reported)[79]. The first case presented is a patient with stage 4 lung adenocarcinoma harboring an oncogenic TMEM106B-BRAF fusion (considered a BRAF Class II variant)[80] (Fig. 7a). The patient received standard first-line therapy consisting of 4 cycles of cisplatin, pemetrexed, and pembrolizumab, followed by 2 cycles of maintenance therapy. Subsequently, the patient underwent radiotherapy to all observed lung and bone lesions and was closely monitored for over 2 years without recurrence or further treatment. Upon progression on prior therapies, the patient enrolled in the KN-8701 trial (NCT04913285). After starting on exarafenib at 800 mg/day (400 mg BID), the patient demonstrated a confirmed partial response, with a significant reduction in target lesions (2 liver metastases) (Fig. 7b). Additionally, while various cancer-associated genes were detected in ctDNA at baseline, a rapid decrease was observed after treatment (the original BRAF fusion was not detected, possibly due to technical limitations of liquid biopsy which can have varying degrees of fusion detection sensitivity) (Fig. 7c). During treatment, the patient experienced CTCAE v5 Grade 2 dermatitis acneiform, prompting a dose reduction of exarafenib to 600 mg/day (300 mg BID). This adjustment allowed for continued treatment with manageable side effects, with a prolonged therapeutic response. At the 32-week CT radiographic evaluation, despite sustained reduction and control of the target lesions, new lymph node metastases were detected, leading to a diagnosis of progressive disease. This case highlights both the potential clinical efficacy of exarafenib and the need to test rational combination therapies such as

combined RAF plus MEK (or RAS) inhibition, supported by our findings, to enhance the clinical impact.

### Clinical activity II: exarafenib treatment of a NSCLC patient with BRAF G469S mutation

The second patient presented is a patient with stage 4 lung adenocarcinoma harboring a BRAF Class II, G469S mutation (Fig. 7d). The patient received standard first-line therapy consisting of 4 cycles of carboplatin, pemetrexed, and pembrolizumab, followed by maintenance therapy. The adenocarcinoma demonstrated progressive disease, and the patient was subsequently enrolled in the exarafenib clinical trial. At the time of trial enrollment, the patient had suspected lymphangitic carcinomatosis (a condition where cancer spreads through the lymphatic system in the lungs) and required oxygen therapy. After initiating exarafenib at 400 mg/day (200 mg BID), the patient was weaned off oxygen within 2 weeks, and a significant improvement in bilateral lung infiltrates was observed after 2 cycles (28-day cycle) (Fig. 7e). The patient had two lymph node metastases as target lesions at baseline, and achieved a partial response approximately 60 weeks after treatment initiation. The disease remained controlled, and despite experiencing CTCAE v5 Grade 2 acneiform rash, which was manageable, the patient has continued exarafenib treatment with sustained disease control throughout the observation period (over 2 years).

These clinical cases provide initial evidence of the promising clinical activity with manageable toxicity of exarafenib in BRAF-mutant NSCLC patients with unmet clinical need, consistent with our supporting preclinical dataset.

## Discussion

Our study highlights the substantial unmet clinical need of Class II and Class III BRAF-mutations in lung cancer and provides a new biomarker-driven therapeutic opportunity, exarafenib and/or combination exarafenib plus MEK inhibitor treatment, to address this prevalent and aggressive cancer subtype. Our comprehensive analysis of the GuardantINFORM™ database, to our knowledge the largest cohort studied to date for evaluating BRAF frequency, reinforced the clinical prevalence of BRAF alterations in NSCLC, particularly Class II and III mutations with high unmet therapeutic need.

Real-world survival analysis revealed that patients with Class I mutations have superior outcomes compared to Class II patients, likely reflecting their access to FDA-approved targeted therapy. In contrast, Class II and III patients predominantly receive immunotherapy-based regimens, underscoring the absence of effective targeted options for these patients who represent the majority of BRAF-mutant NSCLC cases.

Key mechanistic insights and innovations present in our study include the elucidation of exarafenib's potency, selectivity, and mode of action across different BRAF mutation classes. Unlike traditional BRAF inhibitors that primarily target monomeric RAF, the strategic structure-based design features and innovations we described earlier, coupled with our experimental results, demonstrate exarafenib's ability to potently inhibit both monomeric and dimeric forms irrespective of RAF isoform. This unique property allows exarafenib to maintain efficacy across a broader spectrum of BRAF mutations, including those that are typically resistant to current therapies. In this regard, and intriguingly, while exarafenib induced RAF dimerization in some cellular contexts, the downstream pathway remained suppressed, indicating these are largely inactive dimers and providing compelling mechanism of action. Additionally, we found that compared to other pan-RAF inhibitors in development, such as naporafenib and belvarafenib, exarafenib demonstrated enhanced selectivity, potentially offering a more favorable therapeutic window in the context of therapeutic efficacy. Exarafenib has the potential to be a

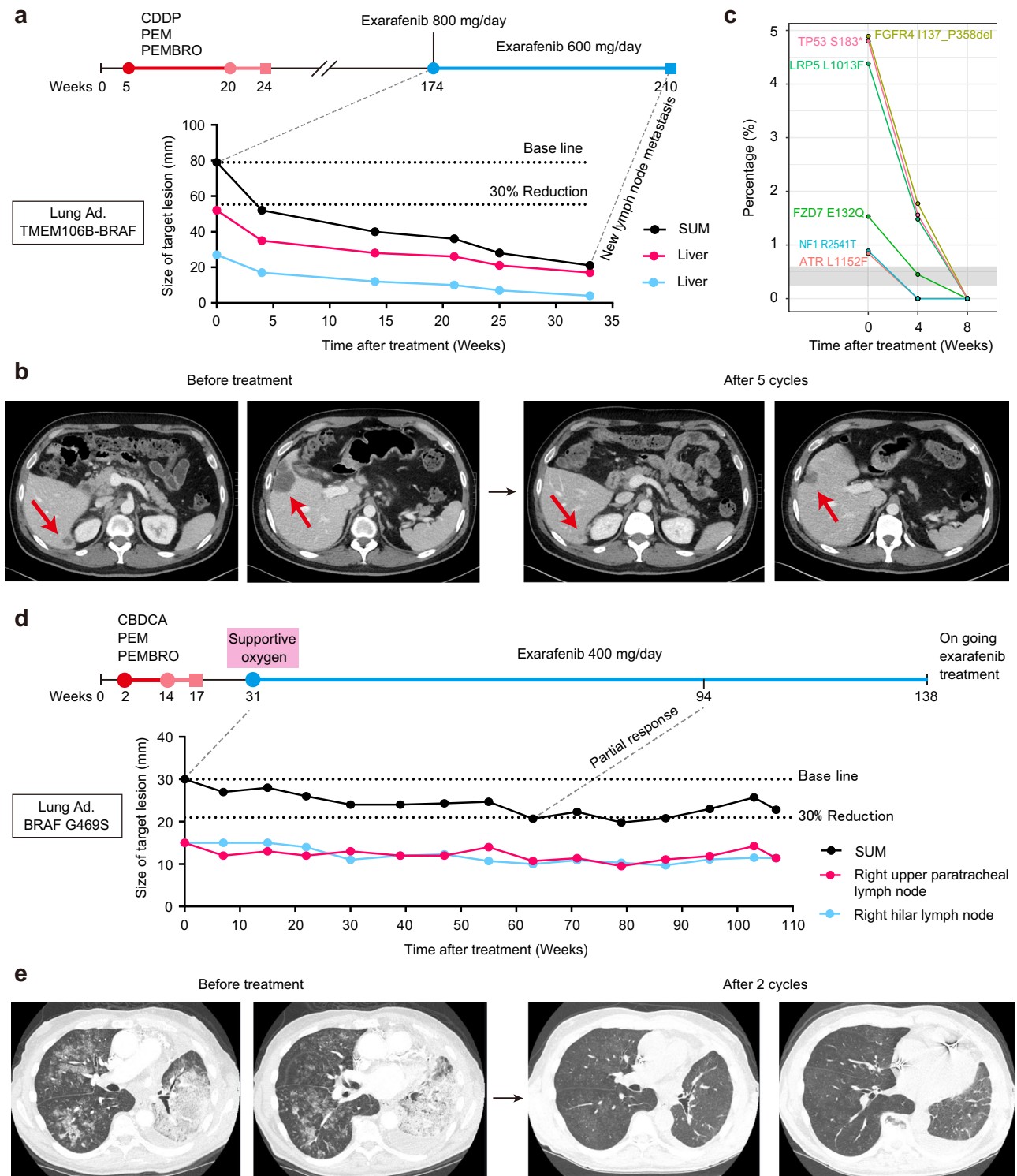

**Fig. 7 | Clinical case studies demonstrating the initial efficacy of exarafenib in patients with advanced BRAF-mutant lung cancer. a** A patient with stage 4 lung adenocarcinoma harboring a TMEM106B-BRAF fusion. Treatment course (top): red – 1st line therapy; pink - maintenance therapy; blue - exarafenib. The line graph (bottom) illustrates the sum of target lesion diameters (SUM) and individual target lesion diameters over time. The 30% reduction line is shown for the determination of partial response (PR). **b** CT scans of liver metastases in the patient from (**a**) before treatment (left) and after 5 cycles of exarafenib (right), showing a partial response. Arrows indicate the target lesions. **c** Longitudinal monitoring of various cancer-associated genes detected in ctDNA from the patient in panel (**a**). Each colored line represents a specific gene alteration, and the y-axis denotes the variant allele frequency (VAF) of each alteration, expressed as a percentage. **d** A patient with stage 4 lung adenocarcinoma harboring a BRAF G469S mutation. Treatment course (top): red – 1st line therapy; pink - maintenance therapy; blue - exarafenib. The line graph (bottom) illustrates the sum of target lesion diameters (SUM) and individual target lesion diameters over time. To evaluate partial response (PR), a threshold line at 30% reduction is presented. **e** Chest CT scans of the patient in (**d**) before treatment (left) and after 2 cycles of exarafenib (right), demonstrating significant improvement in bilateral lung infiltrates. CDDP, cisplatin; CBDCA, carboplatin; PEM, pemetrexed; PEMBRO, pembrolizumab. Source data are provided as a Source data file.

RAF Isoform Dependency Switch

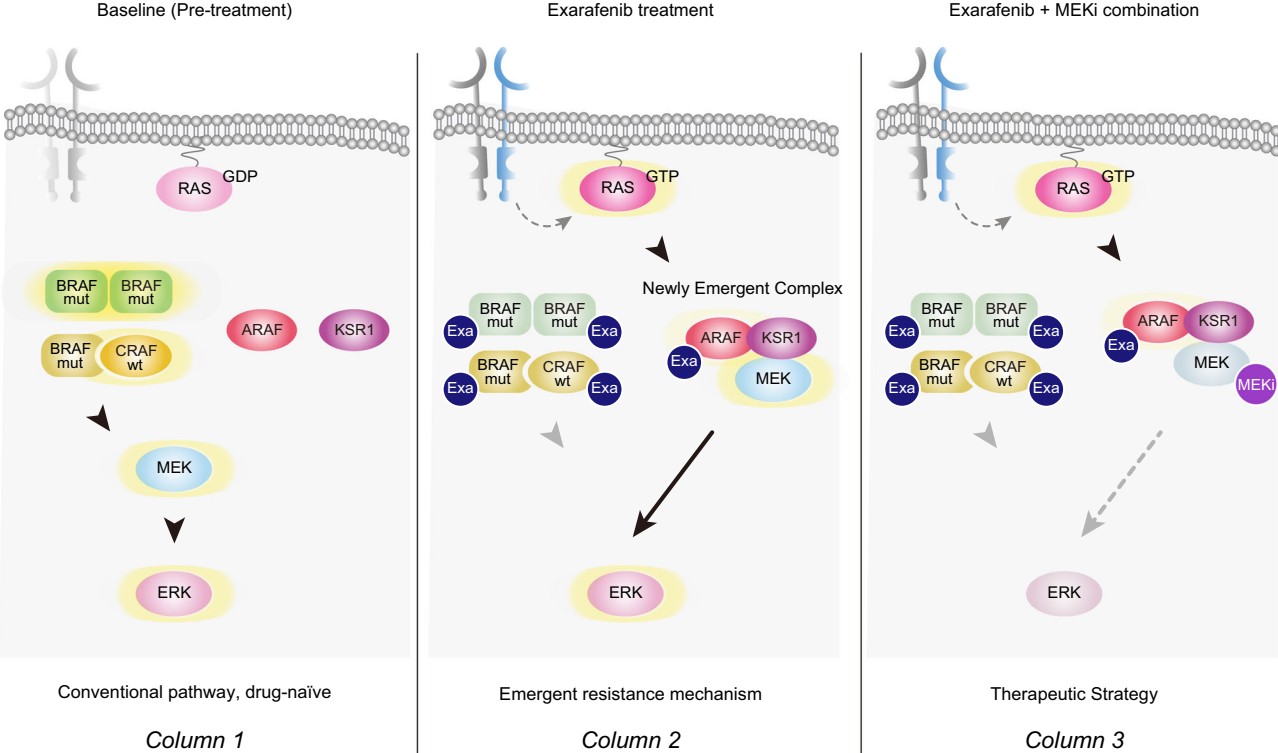

**Fig. 8 | Adaptive resistance mechanism involving RAS-ARAF-KSR1 signaling axis in BRAF-mutant NSCLC.** Schematic model illustrating MAPK signaling mechanisms in BRAF Class II & III mutant exarafenib-resistant cells under different treatment conditions. Column 1 (Pretreatment baseline/drug-free): Resistant cells exhibit high MAPK signaling through conventional oncogenic-BRAF-dependent pathway. Column 2 (Exarafenib treatment): Resistant cells maintain sustained MAPK activation through an alternative ARAF-dependent pathway. RTK activation enhances RAS-GTP loading, which facilitates formation of an ARAF–KSR1 complex stabilized by exarafenib. This emergent complex provides a bypass route for pathway reactivation under drug pressure. Column 3 (Exarafenib + MEKi combination): Combination therapy with MEK inhibitor (MEKi) effectively suppresses the reactivated MAPK pathway by targeting downstream signaling, resulting in improved pathway suppression. Black arrows indicate active signaling pathways; gray arrows indicate suppressed pathways. Exa exarafenib.

best-in-class pan-RAF inhibitor not only in NSCLC patients but also potentially other human cancers with unmet clinical need.

The translational impact of our preclinical findings is exemplified by encouraging published clinical outcomes from the KN-8701 trial[79] (NCT04913285). These include a patient with TMEM106B-BRAF fusion who maintained disease control for several months, and remarkably, a patient with BRAF G469S mutation who achieved sustained response for over two years with dramatic clinical improvement. Both cases demonstrate that exarafenib can provide substantial clinical benefit with manageable toxicity in patients lacking other targeted options.

The centerpiece of our mechanistic discoveries is an adaptive resistance mechanism involving the RAS-ARAF-KSR1 signaling axis (Fig. 8). We uncovered a remarkable convergence whereby resistant cells across all BRAF mutation classes shift from oncogenic BRAF dependency to an alternative ARAF-mediated pathway specifically under drug treatment (Fig. 8, Column 2). This isoform switch represents a fundamental rewiring of RAF signaling architecture that emerges through prolonged drug selection. Molecularly, this ARAF dependency involves formation of drug-induced ARAF-KSR1 complexes that are absent in parental cells. The complex formation is regulated by upstream RAS activation and enhanced by growth factor stimulation. Our cellular thermal shift assays demonstrate that exarafenib directly binds ARAF in resistant cells, with binding enhanced under growth factor stimulation, suggesting RAS activation primes ARAF into a drug-binding competent state. Functionally, the ARAF-KSR1 complex maintains MAPK signaling through a scaffolding mechanism that enhances the efficiency of residual ARAF kinase

activity by recruiting MEK in proximity into the complex. This model is supported by our findings that KSR1 depletion selectively restores drug sensitivity, and that resistant cells retain sensitivity to supra-clinical drug concentrations where scaffolding can no longer compensate for kinase inhibition.

Based on these mechanistic findings, we evaluated rational combination strategies targeting convergent signaling nodes. MEK inhibitors, particularly binimetinib, emerged as highly synergistic partners across all models. Time-course analyses revealed that while exarafenib monotherapy permits progressive MAPK reactivation through the ARAF-KSR1 mechanism, MEK inhibitor combination maintains sustained pathway suppression by targeting the final convergence point of both conventional and bypass signaling (Fig. 8, Column 3). Importantly, our discovery that ARAF mediates resistance to both monotherapy and combination therapy identifies a previously unrecognized vulnerability in RAF-targeted therapy. ARAF represents a conserved resistance node that could be therapeutically targeted through selective inhibitors or upstream RAS inhibition. The enhanced synergy observed with pan-RAS inhibitors specifically in resistant cells validates this approach for overcoming multiple resistance layers.

The sophisticated adaptive rewiring we describe, involving isoform switching and scaffolding complex assembly, differs fundamentally from previously described resistance patterns with selective BRAF inhibitors. While first-generation inhibitors cause rapid paradoxical activation through CRAF-containing dimers, exarafenib resistance involves a distinct mechanism that emerges after prolonged selection under clean pan-RAF inhibition. This provides new insights into how

cancer cells maintain MAPK signaling even when all RAF isoforms are pharmacologically engaged. Despite these advances, important questions remain regarding the structural basis of ARAF's unique role in resistance. Future studies examining specific ARAF domains, including RAS-binding, dimerization, and kinase domains, could provide insights for developing resistance-preventing strategies. Additionally, why cancer cells converge on ARAF rather than other RAF isoforms, and whether this reflects unique regulatory properties or protein interactions, warrants investigation.

Related to this question of RAF isoform selectivity, while our genetic depletion studies identified ARAF as the predominant mediator of resistance, we acknowledge that the role of CRAF in this context cannot be entirely dismissed. Although CRAF knockdown alone did not significantly alter exarafenib sensitivity in any of the resistant cell lines tested, our co-immunoprecipitation data revealed that exarafenib treatment induces both ARAF-KSR1 and ARAF-CRAF complexes in resistant cells (Fig. 5e). This raises the possibility that CRAF may play a supportive or redundant role within the ARAF-centered resistance network. Recent studies have demonstrated functional cooperation between RAF isoforms, where RAF heterodimers can exhibit distinct signaling properties[27,54,55]. The limited effect of CRAF knockdown alone in our resistant models may reflect several possibilities: CRAF could be functionally dispensable in the presence of intact ARAF-KSR1 scaffolding, it could play only a minor supportive role, or its contribution could be masked by compensatory mechanisms from the dominant ARAF-centered signaling network. Future studies employing simultaneous depletion of multiple RAF isoforms, particularly ARAF and CRAF together, would be valuable to dissect potential cooperative or redundant functions within the resistance complex. Such experiments could reveal whether CRAF contributes to maintaining MAPK signaling in the presence of ARAF or whether complete disruption of the RAF heterodimer network requires targeting both isoforms. Understanding these nuanced interactions could inform the development of next-generation isoform-directed RAF inhibitors or combination strategies that more comprehensively disable the adaptive RAF signaling architecture that emerges under therapeutic pressure.

In conclusion, our study provides compelling evidence for exarafenib's therapeutic potential in BRAF-mutated NSCLC and establishes the mechanistic foundation for rational combination strategies currently being evaluated clinically. Notably, preliminary results from the exarafenib plus binimetinib combination therapy arm of the KN-8701 clinical trial (NCT04913285) have recently demonstrated promising tolerability and activity of exarafenib in combination with binimetinib in BRAF-altered solid tumors and NRAS-mutant melanoma[81]. Our preclinical data provide strong rationale for this dual RAF-MEK inhibition strategy, which could represent a significant advance for patients with these challenging genetic driver mutations, including Class II and III BRAF mutations, and may extend to other contexts such as different forms of mutant RAS in lung cancer and potentially other tumor types. The discovery of the RAS-ARAF-KSR1 resistance axis advances our understanding of MAPK pathway plasticity and identifies new therapeutic vulnerabilities. These findings not only inform ongoing clinical development but also provide a roadmap for anticipating and circumventing resistance mechanisms, ultimately working toward improving outcomes for the substantial population of BRAF-mutant NSCLC patients who currently lack effective targeted therapies.

## Methods
### Cell lines and reagents
NSCLC cell lines harboring BRAF mutations (NCI-H2405, NCI-H1755, NCI-H1666, HCC364, NCI-H1395, NCI-H2087, Cal-12T) and various RAS mutations (NCI-H358, NCI-H23, NCI-H2030, SW-1573, A-427, A549, NCI-H441, NCI-H1944, Calu-6, NCI-H2347) were used in this study. HCC364-VR1, a vemurafenib-resistant subclone, was previously generated in our laboratory[36]. Cells were maintained in RPMI-1640 medium (Cytiva),

except for A-427, which was cultured in DMEM (Cytiva). All media were supplemented with 10% fetal bovine serum (FBS; Avantor) and 1% penicillin-streptomycin (Cytiva), and cells were cultured at 37 °C in a humidified atmosphere containing 5% $CO_2$. AALE, a normal epithelial cell line[35], was kindly provided by Eric Collison, University of Washington School of Medicine, and was maintained in SABM supplemented with SAGM SingleQuots (Lonza). HCC364 was kindly provided by David Solit, Memorial Sloan Kettering Cancer Center (MSKCC), New York. All other cell lines were obtained from the American Type Culture Collection (ATCC).

### Establishment of exarafenib-resistant cell lines
To generate exarafenib-resistant cell lines, parental cells were seeded at 50% confluence in 10-cm dishes and initially exposed to exarafenib at a concentration corresponding to the IC50 value. The drug-containing medium was replenished every 5 days. When cells reached confluence, they were passaged, and exarafenib was added the following day after cell attachment. The drug concentration was incrementally increased after every two passages until it reached 1 μM. Once this concentration was achieved, cells were continuously cultured with 1 μM of exarafenib. This process of continuous exarafenib exposure was maintained for over 3 months to establish stable drug-resistant cell lines, designated as NCI-H2405-ER, NCI-H1755-ER, and NCI-H1666-ER, derived from their respective parental cell lines. For combination-resistant cells, NCI-H2405-ER cells maintained in 1 μM exarafenib were exposed to escalating doses of binimetinib (escalated to 50 nM over 3 months), generating the NCI-H2405-EBR cell line. Attempts to generate EBR cells directly from parental cells using simultaneous drug exposure were unsuccessful.

### Compounds
The RAF inhibitors vemurafenib, exarafenib, naporafenib, and belvarafenib were provided by Kinnate Biopharma, while binimetinib (S7007), cobimetinib (S8041), ASTX029 (S0872), SCH772984 (S7101), RMC-4550 (S8718), RMC-6236 (E1597), ipatasertib (S2808), and sotorasib (S8830) were purchased from Selleck Chemicals (Houston, TX, USA).

### Antibodies
Complete list of antibodies can be seen at Supplementary Table 8.

### Western blotting and in vitro kinase assay
Following treatment, cells were washed with cold PBS and lysed in RIPA buffer supplemented with protease and phosphatase inhibitors (Thermo Scientific). Protein concentrations were determined using the BCA protein assay kit (Pierce). Equal amounts of protein (10 μg) were separated by SDS-PAGE and transferred to nitrocellulose membranes. Membranes were blocked with 5% non-fat dry milk in TBS-T for 30 min at room temperature and then incubated with primary antibodies overnight at 4 °C. After washing with TBS-T, membranes were incubated with the appropriate HRP-conjugated secondary antibodies for 30 min at room temperature. Signals were detected using the ECL western blotting substrate (Thermo Scientific) and visualized with the ImageQuant LAS 4000 (GE Healthcare). Quantified bands were normalized to GAPDH as a loading control.

For the in vitro kinase assay, cells were lysed in IP lysis buffer (Pierce) supplemented with protease and phosphatase inhibitors. Protein A/G magnetic beads (Pierce) were incubated with the appropriate antibodies (anti-ARAF, anti-BRAF, or anti-CRAF) for 1 h at 4 °C. The antibody-conjugated beads were then incubated with 150 μg of protein for 4 h to overnight at 4 °C. Beads were washed three times with cold PBS and twice with kinase buffer (Cell Signaling Technology) using a magnetic separator. Kinase reactions were performed by incubating the beads in kinase buffer containing 50 μM ATP and 0.4 μg of inactive MEK1 (Millipore) for 30 min at 30 °C. Reactions were

terminated by the addition of SDS sample buffer, and the samples were boiled for 2 min. Proteins were separated by SDS-PAGE, transferred to nitrocellulose membranes, and analyzed by western blotting using an anti-p-MEK antibody. The kinase assay results were normalized to total MEK levels.

### RAS activation assay

RAS activation was assessed using the RAS-RBD (Ras-binding domain) assay kit (Cytoskeleton), which measures the amount of active, GTP-bound RAS. Cells were lysed in ice-cold lysis buffer provided in the kit, supplemented with protease and phosphatase inhibitors. Lysates were clarified by centrifugation at $10,000 \times g$ for 2 min at 4 °C. Equal amounts of protein (150 μg) were incubated with 10 μL of Raf-RBD protein beads for 1 h at 4 °C with gentle rotation. The beads were washed with the kit's wash buffer, and bound proteins were eluted by boiling in 2× SDS sample buffer for 2 min. The eluted proteins were separated by SDS-PAGE, transferred to nitrocellulose membranes, and analyzed by western blotting using the anti-RAS antibody provided in the kit. Total RAS levels were determined by western blotting of the cell lysates used for the RAS-RBD pull-down assay. The levels of active RAS were normalized to GAPDH as a loading control to account for any variations in total protein content between samples.

### Phospho-RTK array analysis

Phospho-receptor tyrosine kinase (RTK) levels were assessed using the Human Phospho-RTK Array Kit (R&D Systems) according to the manufacturer's instructions. For each array, 300 μg of protein was used, and the arrays were incubated overnight at 4 °C. Pixel density was quantified using ImageJ software (National Institutes of Health), and the data were normalized to the positive control spots.

### Cell viability assays and drug screening

2D and 3D cell viability assays were performed to assess the efficacy of exarafenib, belvarafenib, vemurafenib, and sotorasib in various cell lines. For 2D assays, BRAF-mutant cells were seeded into 96-well plates at 2500 cells per well and incubated with either a vehicle control (DMSO) or serial 3-fold dilutions of compounds. In the preparation of 3D assays, KRAS-mutant lung cancer cells were seeded at a density of 2500 cells per well in 50 μL of growth medium into 96-well ultra-low attachment plates (Corning). In order to induce spheroid formation, centrifugation at $300 \times g$ for 5 min was carried out. After incubation for 72 h, the spheroids were treated in triplicate with serial 3-fold dilutions of compounds or vehicle control, DMSO, to a final volume of 100 μL per well. Unless otherwise indicated, plates were incubated for 120 h at 37 °C in a humidified atmosphere containing 5% $CO_2$. Cell viability was measured for 2D cultures using the CellTiter-Glo® 2.0 Assay (Promega) and for 3D cultures using the CellTiter-Glo 3D Cell Viability Assay (Promega), according to the manufacturer's instructions. In each of these assays, luminescence was measured with a SpectraMax M5 plate reader (Molecular Devices) and cell viability determined as percent relative to vehicle control. IC50 values were calculated by non-linear regression analysis using GraphPad Prism 7.0 by fitting the data to a four-parameter logistic curve. This form of normalization and calculation of IC50 was applied consistently to all cell lines and compounds tested in 2D and 3D assays. Statistical significance for cell viability data was determined through a two-tailed Student's t-test.

For the comprehensive drug screening, to ensure data quality and enable reliable synergy calculations, exarafenib was used at 9 dosages (3-fold change from 10 μM) and each drug partner candidate at 5 dosages (3-fold change), with the addition of DMSO controls. The range of dosages for each drug was determined by referring to previous in vitro studies[61,64–66,82], considering varying potencies across the different compounds. The initial screening was performed in triplicate on 6 cell lines, comprising 3 sets of parental and resistant cells, and then expanded to cells of interest. Loewe synergy scores were

calculated for each drug combination based on the drug screening assay data using the Combenefit software[83].

To assess the proliferation capabilities of exarafenib-resistant cell lines compared to their parental counterparts, cells were seeded in 96-well plates. Resistant cells were cultured in the presence of 1 μM exarafenib, while parental cells were grown without drug exposure. Cell viability was measured using the CellTiter-Glo® 2.0 Assay on day 1 and day 3. Relative cell viability was calculated as fold change compared to day 1 measurements. Each condition was tested in triplicate.

### Cell proliferation analysis by direct cell counting

NCI-H2405-ER cells were seeded at 30,000 cells/well in 6-well plates and treated with 1 μM exarafenib for 72 h. Following treatment, cells underwent drug withdrawal (complete medium replacement with drug-free RPMI-1640) with counts at −4 h, −24 h, and −48 h. For re-challenge, cells at −48 h were re-exposed to 1 μM exarafenib with counts at +4 h, +24 h, and +48 h. Viable cells were quantified using trypan blue exclusion with a Vi-CELL XR counter (Beckman Coulter). Data from three replicates per time point were normalized to the 72-h treatment baseline (fold change = 1.0) and expressed as mean ± SD.

### Ba/F3 cell viability assay

The effect of exarafenib, belvarafenib, vemurafenib, and sotorasib on cell viability was evaluated in Ba/F3 cells expressing various BRAF and KRAS mutations. The experiments were conducted at KYinno Biotechnology (Beijing) Co., Ltd., a contract research organization. Ba/F3 cells were seeded at 3000 cells per well in a 96-well plate in 90 μL growth media and incubated overnight at 37 °C with 5% $CO_2$. Compounds were serially diluted from a 10 μM top dose for a 9-point 3-fold dilution curve in DMSO. After 72 h of incubation at 37 °C with 5% $CO_2$, cell viability was assessed using the CellTiter-Glo® 2.0 Assay (Promega) according to the manufacturer's instructions.

### siRNA-mediated Gene Knockdown

To perform gene knockdown, siRNAs targeting specific genes were used. The following siRNAs were purchased from Dharmacon (Horizon Discovery): non-targeting control (D-001810-10-50), KRAS (L-005069-00-0020), NRAS (L-003919-00-0020), HRAS (L-004142-00-0020), AXL (L-003104-00-0020), MET (L-003156-00-0020), EGFR (L-003114-00-0020), ARAF (L-003563-00-0020), BRAF (L-003460-00-0010), CRAF (L-003601-00-0010), and KSR1 (L-003570-00-0005). Cells were seeded into appropriate plates, and transfection was performed the following day using Lipofectamine RNAiMAX (Invitrogen, Thermo Fisher Scientific) according to the manufacturer's protocol. Twenty-four hours after transfection, cells were plated and incubated for the appropriate time periods prior to the performance of different types of experiments. The efficiency of gene knockdown was confirmed by Western blot analysis.

### Proximity ligation assays (PLA)

Cells were seeded in 4-well Lab-Tek chambers and cultured overnight. Proximity ligation assays were performed using the Duolink In Situ Red Detection Kit (Millipore Sigma). Cells were fixed in 4% paraformaldehyde for 10 min at 37 °C, permeabilized with −20 °C methanol for 10 min, then treated with 0.25% Triton X-100 for 10 minutes before blocking for 1 h at 37 °C. Primary antibodies (1:200 dilution) were incubated overnight at 4 °C. PLA probes were applied at 1:5 dilution for 1 h at 37 °C, followed by ligation (30 min) and amplification (100 min) at 37 °C. Nuclei were stained with DAPI, and images were acquired using a 60× oil objective. PLA puncta per cell were quantified using Fiji ImageJ software.

### Three-dimensional (3D) structure prediction

For three-dimensional (3D) structure prediction, we used the protein sequences of ARAF (NP_001645.1, 606 amino acids) and KSR1 (Q8IVT5,

923 amino acids) as inputs to the AlphaFold3 server. The predicted structures were downloaded in.cif format and subsequently converted to PDB format using the PyMOL software package for downstream structural analysis.

Because no experimentally determined crystal structure of the pan-RAF inhibitor Exarafenib bound to either ARAF alone or to the ARAF−KSR1 complex is currently available, we adopted a homology-based modeling strategy. Specifically, we utilized the BRAF−Exarafenib co-crystal structure (PDB ID: 8QQG) as a reference. In PyMOL, the kinase domain of ARAF was aligned with the corresponding kinase domain of BRAF, and the coordinates of Exarafenib were transferred into the predicted ARAF binding pocket to generate an ARAF−Exarafenib binding model.

To evaluate the potential influence of KSR1 binding on inhibitor accommodation, we further superposed the ARAF−Exarafenib model with the ARAF−KSR1 predicted complex. By aligning the ARAF kinase domain in both structures, Exarafenib was transferred into the ARAF binding pocket within the ARAF−KSR1 complex. To identify molecular interactions, we extracted the contact residues from both ARAF and KSR1 that lie within 5 Å (angstrom) of Exarafenib.

### GuardantINFORM™ database analysis

Data for this real-world evidence analysis were obtained from the GuardantINFORM™ database, which contains targeted genomic data from circulating tumor DNA linked to a claims database for patients in the United States with advanced stage solid tumors who underwent testing with a Guardant360 assay between March 2014 and June 2021. At the time of data cut, the database included over 160,000 patients diagnosed with advanced or metastatic cancer. Within this population, the Guardant360 liquid biopsies were collected at various timepoints, with some obtained at initial diagnosis and others during disease progression. The database was queried for ctDNA tests of patients with a BRAF mutation. ctDNA was extracted from whole blood and isolated for digital targeted panel sequence. At the time of the study, Guardant360 included analysis of single-nucleotide variants in up to 83 genes, as well as small insertions or deletions, amplifications, and gene rearrangement/fusions in select genes. Nonsynonymous mutations and focal copy number alterations were processed using the SAS 9.4 statistical computing program. All nonsynonymous BRAF mutations and focal amplifications were included, while synonymous mutations and those associated with clonal hematopoiesis were excluded. The analysis focused on patients exhibiting Class I, II, or III BRAF alterations, with a specific emphasis on NSCLC ($n = 2398$). Sex and age were determined based on the linked claims database. Sex was considered as a biological variable in the study design to assess potential associations with BRAF mutation classes. No participant compensation was provided for this database analysis. BRAF alterations were classified according to Supplementary Table 1, which was summarized by referencing previous studies[3,12,15,84].

### Patient characteristics and statistical analysis

Patient characteristics of NSCLC cases with BRAF Class I, II, or III alterations were analyzed using the GuardantINFORM™ database. Cases with multiple BRAF alterations across functional classes were excluded from the analysis. Age, gender, and tobacco use history were identified as potential covariates. To compare the distribution of covariates across BRAF classes, we employed pairwise chi-squared tests with Holm-adjusted P-values.

### Survival analysis and statistical methods

Overall survival was measured from the time of the detection of BRAF, with patients censored at the date of their last known claim activity if no death date was available. Cohorts were compared pairwise using the log-rank test, and Kaplan−Meier curves were used to visualize survival probabilities over time. Descriptive statistics were reported for each BRAF class. Comparisons were made between the classes using chi-square tests when outcomes were binary and Student's $t$ tests when outcomes were continuous, with a *P-value < 0.05 considered statistically significant. Unadjusted Cox proportional hazard model was run to assess the relationship between BRAF classes. Outcomes from the model were reported as hazard ratios (HRs) with 95% confidence intervals. An HR of 1 indicates no difference in risk between groups, while an HR > 1 suggests increased risk, and an HR < 1 indicates decreased risk for the comparison group relative to the reference group.

### Kinome specificity profile

The kinome specificity profiles for 1 µM exarafenib, naparafenib, and belvarafenib were tested across a panel of 368 wild-type kinases using the HotSpot Technology[85] (Reaction Biology) at 10 µM ATP. Kinase inhibition was calculated as (100 − Enzyme activity), relative to DMSO controls.

### In vivo efficacy and pharmacokinetic (PK) and pharmacodynamic (PD) studies

The in vivo efficacy of exarafenib was evaluated, both as a single agent and in combination with binimetinib, using cell line-derived xenograft (CDX) and patient-derived xenograft (PDX) models. These studies were conducted at various institutions, including Pharmaron (Beijing, China), Champions Oncology (Hackensack, New Jersey, USA), XenoS-TART (San Antonio, Texas, USA), and GenenDesign (Shanghai, China). Animals were housed under controlled environmental conditions with a 12 h light/12 h dark cycle, ambient temperature of 20−26 °C, and relative humidity of 30−70%. The maximal tumor size permitted by the institutional ethics committees was 2500 mm³. All animals were immediately euthanized upon reaching this limit.

Pharmaron used NCI-H1755 CDX model for efficacy and NCI-H2405 CDX model for efficacy and PK/PD studies. Female BALB/c nude mice (6-8 weeks) were inoculated subcutaneously into the right flank with cells cultured in RPMI1640 medium with 5% HI FBS. For CDX studies, cells were inoculated into mice in 0.1 ml of a 1:1 mixture of media and Matrigel. Efficacy studies began at ~300 mm³ tumor volume, with mice ($n = 9$ per group) receiving vehicle (0.5 methylcellulose, 0.1% Tween 80 in water), exarafenib (1.5−30 mg/kg), binimetinib (2−3 mg/kg), or a combination twice daily. Animals were euthanized if in deteriorating condition or tumor size exceeded 2500 mm³. For PK and PD analyses, animals were treated for 3 days, then plasma samples were collected at 0, 0.25, 0.5, 1, 3, 7, 12, 18, and 24 h post-last dose, while tumor samples were collected at 1, 7, 24, 48, and 72 h post last dose ($n = 3$ per timepoint). Plasma concentrations were measured by LC/MS/MS, and tumor lysates were assessed for p-ERK1/2 and total ERK1/2 levels using the MSD Phospho/Total ERK1/2 Whole Cell Lysate Kit (Meso Scale Diagnostics, # K15107D).

Champions Oncology (CTG-1444, CTG-3703 PDX) and XenoS-TART (ST5570 PDX) implanted subcutaneous tumor fragments into the right flank of female athymic nude-Foxn1nu mice (6−8 weeks). Studies began at 175−300 mm³ mean tumor volume. Animals were euthanized if tumor volume was reached 2500 mm³. GenenDesign used LUN023 and LUN037 PDX models, following the same procedures but using Balb/c nude mice ($n = 9$ per group).

For all xenograft studies, tumors width and length were measured by calipers, and tumor volume was calculated by $width^2 \times length \times 0.5$. Tumor growth inhibition (TGI) was calculated as $[1 − (TVf − TV0)$ treated/$(TVf − TV0)$ control$] \times 100$, where TVf and TV0 are the mean tumor volumes on the final day and the initial day of treatment, respectively. Statistical analysis was performed using one-way ANOVA with Bonferroni correction for multiple comparisons. Significance levels were set at $*P < 0.05$, $**P < 0.01$, $***P < 0.001$, and $****P < 0.0001$.

## Patient case studies from the KN-8701 clinical trial

Two patients with advanced BRAF-mutant lung cancer treated with exarafenib in the phase I/Ib clinical trial, KN-8701 (NCT04913285), were selected for in-depth case studies. The first patient was treated at Stanford University School of Medicine, and the second patient was treated at the University of California, San Diego, Moores Cancer Center. Clinical data, including patient demographics, disease characteristics, treatment history, and response to exarafenib, were collected from the respective study sites. Tumor response was assessed using the Response Evaluation Criteria in Solid Tumors (RECIST) version 1.1. Adverse events were graded according to the Common Terminology Criteria for Adverse Events (CTCAE) version 5.0. In addition to clinical data, circulating tumor DNA (ctDNA) analysis was performed for the first patient to monitor treatment response. ctDNA was isolated from plasma samples collected at baseline and at various timepoints during treatment. The clinical information presented was current up to December 2023.

## Statistics and reproducibility

Quantitative results are reported as the mean ± standard deviation (S.D.) or the standard error of the mean (S.E.M.), as indicated in the figure legends. Statistical analyses were conducted using GraphPad Prism 7.0. Unless otherwise noted, comparisons between two groups were performed using two-tailed Student's $t$-tests. For analyses involving multiple independent groups, a one-way ANOVA was performed, followed by Bonferroni correction to account for multiple comparisons. When representative data are presented, they depict one of at least two separate experiments yielding comparable outcomes.

## Reporting summary

Further information on research design is available in the Nature Portfolio Reporting Summary linked to this article.

## Ethics statement

The clinical case studies from the KN-8701 trial (NCT04913285) were conducted in compliance with the Declaration of Helsinki. The study protocol was approved by the Stanford Institutional Review Board for the first patient and by the Advarra Institutional Review Board for the second patient. Both patients provided written informed consent for trial participation and for reporting of these case studies. No compensation was provided for participation.

The GuardantINFORM™ database analysis used deidentified research data approved by the Advarra Institutional Review Board with a waiver of consent, in compliance with sections 164.514(a)-(n)1ii of the U.S. Health Insurance Portability and Accountability Act.

All animal experiments followed the Association for Assessment and Accreditation of Laboratory Animal Care (AAALAC) guidelines and were approved by the Institutional Animal Care and Use Committees (IACUC) at Pharmaron (Beijing, China), Champions Oncology (Hackensack, New Jersey, USA), XenoSTART (San Antonio, Texas, USA), and GenenDesign (Shanghai, China).

## Data availability

The Whole Genome Sequencing (WGS) data generated in this study are publicly available as an NCBI Bioproject under accession number PRJNA1123823. The datasets generated and analyzed during the current study using the GuardantINFORM™ database are not publicly available due to the use of a third-party healthcare claims database with associated privacy and contractual restrictions. Access to the data is subject to approval by Guardant Health, Inc. Researchers interested in accessing the data for verification purposes may submit requests to Guardant Health directly. The remaining data are available within the Article, Supplementary Information, or Source Data file. Biological material (e.g., cell lines) generated in this study is available by request from the corresponding author. Additional reagents can be made available upon reasonable request. Source data are provided as a Source Data file. Source data are provided with this paper.

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

## Acknowledgements

This work was supported in part by a Uehara Memorial Foundation Postdoctoral Fellowship to T. Manabe and from the National Institutes of Health (NIH)/National Cancer Institute (NCI) to T. Bivona: U54CA224081, U01CA217882, R01CA204302, R01CA211052, as well as Kinnate Biopharma, and the Chan-Zuckerberg Biohub, both to T. Bivona. Study sponsorship changed from Kinnate Biopharma to Pierre Fabre on Feb 27, 2024.

## Author contributions

T.M., T.S.W., N.M., L.S., R.J.P., E.S.M., E.A.M., and R.W. conceived and designed the study. T.M. contributed to the design, conduct, and interpretation of all experiments and clinical aspects of the study. P.S., E.D., and N.Z. performed the bioinformatics analysis of the Guardan-tINFORM™ database. H.C.B., Q.L., Y.C., D.L.K., P.A., and W.W. contributed in vitro experimental design, conduct, and interpretation. K.B.G. contributed design, conduct, and interpretation of the kinome profiling. C.L. contributed design, conduct, and interpretation of in vivo experiments. V.K., J.W.N., C.T.C., and S.K. contributed to the interpretation of the clinical aspects of the study. T.G.B. supervised the project. T.M. and T.G.B. wrote the manuscript with input from all authors. All authors reviewed and approved the final version of the manuscript.

## Competing interests

T.G.B. is an advisor to Revolution Medicines, Relay, EcoR1, Engine, Novartis, Pfizer, Astrazeneca, Genentech, AbbVie, Daiichi Sankyo, and Granule Therapeutics, and receives research funding from Revolution Medicines, Verastem, and Nextpoint. J.W.N. reports stock ownership in SecondLook Health and receives royalties from UpToDate. He serves in a consulting or advisory role for AstraZeneca, Genentech/Roche, Takeda Pharmaceuticals, Eli Lilly and Company, Amgen, Iovance Biotherapeutics, Blueprint Pharmaceuticals, Regeneron Pharmaceuticals, Natera, Gilead Sciences, AbbVie, Summit Therapeutics, Novartis, Novocure, Janssen Oncology, Anheart Therapeutics, Bristol-Myers Squibb, Nuvation Bio, Boehringer Ingelheim, Daiichi Sankyo, GlaxoSmithKline, Oxford Bio-Therapeutics, Taiho Pharmaceutical, Pfizer, and Foresight Diagnostics. He receives research funding (to self) from Revolution Medicines and Nuvalent, and research funding (to institution) from Genentech/Roche, Merck, Novartis, Boehringer Ingelheim, Exelixis, Nektar Therapeutics, Takeda Pharmaceuticals, Adaptimmune, GSK, and Janssen. He has received honoraria from Research to Practice CME, Medscape CME, Projects in Knowledge CME, MJH Life Sciences CME, PlatformQ/Medlive CME, IDEOlogy Health, Navya Network, Curio Science, and PER. C.T.C. receives grants from ADC Therapeutics, Bolt Biotherapeutics, D3 Bio, Genentech-Roche, Gilead Sciences, Kinnate Biopharma, Mersana, ORIC Pharmaceuticals, Palleon Pharmaceuticals, Pionyr Therapeutics, Rain Oncology, Revolution Medicines, Seagen, Tango Therapeutics, GSK, Eli Lilly, and Takeda and is a consultant or on advisory boards for Johnson & Johnson, Mubadala Capital, Tango Therapeutics, and Radionetics Oncology. S.K. serves as a consultant for Aadi Bioscience, Medpace, Foundation Medicine, NeoGenomics, and CureMatch. S.K. receives speaker's fees from Chugai, Roche/Genentech, and Bayer, and serves on the advisory board for Pfizer. S.K. has research funding from ACT Genomics, Sysmex, Konica Minolta, OmniSeq, Personalis, and Function Oncology. T.S.W., P.S., N.M., C.L., K.B.G., L.S., R.J.P., E.S.M., and R.W. were full-time employees of Kinnate Biopharma at the time of the research. P.S. is an employee of Tupos Therapeutics and serves as a paid consultant for Pierre Fabre Laboratories. T.S.W. is an employee of Alterome Therapeutics. N.M. is an employee of Actio Biosciences. C.L. is an employee of Zentalis Pharmaceuticals. K.B.G. is a co-founder and employee of KymaThera Inc. L.S. is an employee of Fortvita Biologics. R.J.P. is an employee of Innovent Biologics. E.S.M. is an employee of Arpeggio Biosciences, Oncology New Co, and Storm Therapeutics.

E.A.M. receives compensation, owns equity, has stock options, serves on the BOD, and is a co-founder of Alterome Therapeutics and Sidewinder Therapeutics. He serves as an advisor and has stock options in Architect Therapeutics, and previously received compensation, owned equity, had stock options, served on the BOD, and is a co-founder of Kinnate Bio-pharma. R.W. is an employee of Khora Therapeutics and serves as a paid consultant for Pierre Fabre Laboratories. E.D. and N.Z. are full-time employees and shareholders of Guardant Health. The remaining authors declare no competing interests.
