## [Transparent Peer Review file · Nature Communications]

Pan-RAF Inhibitor Exarafenib Targets BRAF Class II/III NSCLC and Reveals ARAF-KSR1 Resistance and Combination Strategies

Corresponding Author: Dr Trever Bivona

Version 0:

Reviewer comments:

Reviewer #1

(Remarks to the Author)

The authors have convincingly addressed all my requests providing appropriate reasoning and discussion. The only exception was the inability to carry out my suggestion to perform in vivo experiments with patient derived material. Yet, I understand that, as explained by the authors, that failure to establish CDX models from resistant cells is a frequent experimental outcome.

Author Response:

We sincerely thank all three Reviewers for their thoughtful and constructive feedback on our manuscript. We deeply appreciate the time and expertise they have invested in evaluating our work. Their insightful comments have helped us strengthen our manuscript and better articulate the significant novel contributions of our study.

In response to the Reviewers' valuable suggestions, we have conducted additional experiments, expanded our analyses, and clarified our presentation of the data. These revisions include comprehensive mechanistic studies revealing an unanticipated new mechanistic twist: an interesting ARAF-KSR1 signaling axis that emerged as a novel axis regulating pan-RAF inhibitor efficacy/resistance; serum dependency analyses that elucidate the mechanistic role of mitogenic signals in resistance pathway reactivation; an expanded clinical analysis from the GuardantINFORM™ database; Each of these new experiments and findings was prompted by the Reviewers' insightful comments.

Given the substantial nature of these new mechanistic discoveries, particularly the identification of the RAS-ARAF-KSR1 signaling axis as a previously unrecognized resistance mechanism, we have revised the manuscript title to "Targeting BRAF Class II and III Mutations in NSCLC with the pan-RAF inhibitor Exarafenib Reveals ARAF-KSR1-Mediated Resistance and Rational Combination Strategies" to more accurately reflect these major findings.

We believe these revisions have substantially improved the manuscript and address the comments raised by the Reviewers. *We hope that the improved, revised manuscript is now suitable for publication in Nature Communications.*

Below, we provide detailed responses to each Reviewer's comments.

Reviewer #1:

The authors describe characterization of a recently developed pan-RAF inhibitor for lung cancers with class 1, II or III BRAF mutations. They then establish drug resistant derivatives and test a number of combinations of MAPK drugs to overcome resistance. Not surprisingly (based on prior literature), they find that combination with MEK or ERK inhibitor overcomes resistance to exarafenib. The major criticism of this work is that there is really not a lot of novelty in this work. Activation of RAS through RTK activation as a mechanism of resistance to RAF or MEK inhibition has been seen by many and has formed the bases for combination trials with SHP2 inhibitors (and found to be overly toxic). This work deserves to be published, but in a more specialized journal.

We sincerely thank the Reviewer for their thoughtful consideration of our manuscript and for acknowledging that this work deserves to be published. We deeply appreciate the Reviewer's expertise in this field and their perspective on novelty, which has prompted us to better articulate the significant innovations in our study and to conduct additional experiments to increase novelty. We believe that our new findings provide substantial

mechanistic and translational advances that significantly move the field forward beyond published work.

Major Novel Mechanistic Discoveries:

Our revision experiments have revealed a previously unknown resistance mechanism involving a RAS-ARAF-KSR1 signaling axis that uniquely emerges to promote pan-RAF inhibitor resistance:

1. ARAF-Dependent Resistance: As shown in our initial submission, resistant cells exhibited upstream RAS activation under drug treatment, which was involved in sustaining MAPK signaling. To further elucidate this mechanism, we investigated whether activated RAS maintains MAPK activation through specific RAF isoforms by performing knockdown experiments of each RAF isoform (ARAF, BRAF, CRAF) in both parental and resistant cells.

Cell viability assays revealed that in parental cells harboring Class II BRAF mutants that are hyperactive (NCI-H2405 and NCI-H1755), BRAF knockdown resulted in the greatest reduction of exarafenib IC50 values, as expected (Fig. 5a, blue arrow). In the Class III BRAF mutant NCI-H1666 cells, reflecting the kinase-dead mutation characteristics, both BRAF and CRAF knockdowns contributed to exarafenib IC50 reduction. Notably, ARAF knockdown also reduced IC50 values in Class III parental cells, suggesting a previously unreported ARAF dependency in Class III BRAF mutant cells. Moreover, all Class II and III BRAF mutant, exarafenib resistant cell lines consistently showed the greatest IC50 reduction upon ARAF knockdown, demonstrating a clear convergent shift in RAF isoform dependency at acquired resistance (Fig. 5a, red arrow). The whole cell viability assay curves are shown in Supplementary Fig. 8a.

Fig. 5a

Sup Fig. 8a

(Fig. 5a) Cell viability IC50 heatmap showing RAF isoform dependency in parental and exarafenib-resistant cell lines calculated from Sup Fig.8a. NCI-H2405, NCI-H1755 (Class II BRAF mutants), and NCI-H1666 (Class III BRAF mutant) parental and resistant (-ER) cells were transfected with scrambled control (SCR), siARAF, siBRAF, or siCRAF and treated with exarafenib. IC50 values (nM) are displayed as a heatmap. Blue arrow indicates predominant BRAF dependency in parental cells; red arrow indicates shift to ARAF dependency in resistant cells across all mutation classes. **(Sup Fig. 8a)** Complete dose-response curves for RAF isoform knockdown experiments. Cell viability was measured in parental (top) and exarafenib-resistant (bottom) cell lines following transfection with scrambled control (siSCR, black), siARAF (red), siBRAF (green), or siCRAF (purple) and treatment with increasing concentrations of exarafenib. Data represent mean \pm SD. ****P < 0.0001.

Protein analysis using Class II BRAF mutant H2405 exarafenib resistant cells showed that under drug-free conditions, only BRAF knockdown reduced pMEK, pERK, and pRSK levels, indicating BRAF dependency similar to parental cells (Fig. 5b, blue arrow). However, under drug treatment, ARAF knockdown resulted in the most profound signal suppression (Fig. 5b, red arrow). These results demonstrate for the first time that resistant cells undergo a switch to an ARAF-dependent MAPK activation maintenance mechanism in the presence of drug treatment.

Fig. 5b

(Fig. 5b) Western blot analysis showing differential RAF isoform dependency for MAPK pathway maintenance under drug-free and drug treatment conditions. NCI-H2405 parental and resistant (NCI-H2405-ER) cells were transfected with scrambled control (SCR) or siRNAs targeting ARAF, BRAF, or CRAF, then treated with DMSO (-) or exarafenib (500 nM, +) for 48 hours. Blue arrow indicates BRAF dependency in resistant cells under drug-free conditions; red arrow indicates ARAF dependency in resistant cells under drug treatment.

To further characterize this mechanism in detail, we examined time-dependent signaling changes following pan-RAF inhibitor treatment under ARAF knockdown conditions. In our initial submission, we demonstrated that resistant cells exhibit time-dependent recovery of MAPK activity after drug treatment, whereas cells with ARAF knockdown showed markedly suppressed signal recovery (Fig. 5c). This temporal analysis provides evidence that ARAF mediates the transition from conventional oncogenic-BRAF-dependent signaling (suppressed by exarafenib) to an alternative bypass pathway that enables sustained MAPK activation. Furthermore, we investigated whether this ARAF-mediated pathway reactivation depends on extracellular growth factor signaling by examining MAPK activity under different serum concentrations with and without exarafenib treatment. Resistant cells showed enhanced MAPK pathway reactivation under high serum (10% FBS) conditions compared to low serum (3% FBS) during drug treatment (Fig. 5d, blue arrow), while ARAF knockdown abolished this elevated pathway reactivation (Fig. 5d, red arrow). These serum-dependency experiments demonstrate that the ARAF-mediated bypass mechanism requires upstream RTK-RAS activation specifically in the context of exarafenib treatment, establishing a functional link between growth factor signaling and the ARAF-dependent resistance pathway.

Fig. 5c

Fig. 5d

(Fig. 5c) Time-course analysis of MAPK pathway reactivation following exarafenib treatment in resistant cells. NCI-H2405-ER cells were transfected with scrambled control (SCR) or siARAF, then treated with exarafenib (500 nM) for the indicated time points (0, 1, 4, 24 hours). **(Fig. 5d)** Serum dependency of MAPK pathway reactivation in exarafenib-resistant cells. NCI-H2405-ER cells were transfected with scrambled control (SCR) or siARAF and cultured in low (3%) or high (10%) serum conditions, then treated with DMSO (-) or exarafenib (500 nM, +) for 24 hours. Blue arrows indicate enhanced MAPK pathway reactivation under high serum conditions; red arrows indicate ARAF knockdown abolishes serum-dependent pathway reactivation.

These results strongly suggest that drug-induced activation of upstream receptor tyrosine kinase (RTK)-RAS signaling primarily induces MAPK pathway reactivation through ARAF. Thus, resistant cells undergo an adaptive switch from conventional oncogenic-BRAF-dependent signaling to an alternative ARAF-dependent bypass mechanism under drug treatment conditions.

- 2. Novel RAS-ARAF-KSR1 Signaling Cascade:** The selective ARAF dependency observed in resistant cells prompted us to investigate the molecular mechanisms by which ARAF maintains MAPK signaling under pan-RAF inhibition. To elucidate ARAF's functional interactions, we examined its binding partners through co-immunoprecipitation analysis, focusing on other RAF isoforms and KSR1, a critical scaffolding protein that organizes RAF-MEK-ERK signaling complexes and has been implicated in resistance mechanisms to RAF and MEK inhibitors (Mckay MM et al., *Curr Biol.* 2011 and Daley RB et al., *Proc Natl Acad Sci USA*, 2023), although not in the setting of ARAF dependence.

Our co-immunoprecipitation studies reveal striking differences in ARAF protein interaction patterns between parental and resistant cells (Fig. 5e). In resistant cells, ARAF-BRAF interactions were significantly reduced compared to parental cells, while exarafenib treatment specifically induces both ARAF-CRAF and ARAF-KSR1 binding in resistant NCI-H2405-ER cells but not in parental cells (Fig. 5e). CRAF knockdown in resistant cells did not significantly impact exarafenib sensitivity. Therefore, we focused our subsequent mechanistic studies on this new ARAF-KSR1 interaction we found.

We wondered why exarafenib was unable to effectively inhibit ARAF (or CRAF)-MEK signaling in the resistant cells. One possibility is that exarafenib is not able to bind the ARAF-KSR1 or ARAF-CRAF complex due to allosteric or structural effects of increased ARAF-KSR1 protein interactions. Our finding that exarefenib treatment induced the ARAF-KSR1 and ARAF-CRAF binding suggested against this scenario. Nevertheless, to more directly address this question, we performed cellular thermal shift assays (CETSA). In parental cells, exarafenib stabilized BRAF, indicating binding, but showed minimal engagement with ARAF, CRAF, and KSR1 (Supplementary Fig. 8b). In contrast, the resistant cells demonstrated enhanced exarafenib engagement with both ARAF (red box) and CRAF in addition to the maintained BRAF binding observed in parental cells, while KSR1 (blue box) showed minimal exarafenib-induced thermal stabilization (Supplementary Fig. 8b).

Fig. 5e**Sup Fig. 8b**
(Fig. 5e) Co-immunoprecipitation analysis of ARAF protein interactions in parental and resistant cells. NCI-H2405 parental and NCI-H2405-ER cells were treated with DMSO (-) or exarafenib (500 nM, +) for 24 hours. ARAF was immunoprecipitated and co-precipitating proteins (BRAF, CRAF, KSR1) were detected by immunoblotting. Whole cell lysates (WCL) show input protein levels. **(Sup Fig. 8b)** CETSA analysis of RAF isoforms and KSR1 in parental and resistant cells treated with DMSO or exarafenib (500 nM) for 24 hours. Exarafenib enhanced thermal stabilization of ARAF (red box) and CRAF in resistant cells, while BRAF stabilization was largely maintained. KSR1 showed minimal stabilization (blue box).

Consistent with the upstream growth factor/RTK/RAS dependence we show, EGF stimulation further amplifies this ARAF-KSR1 interaction in resistant cells (Fig. 5f, red arrow). Conversely, pharmacological inhibition of RAS with the pan-RAS inhibitor RMC-6236 markedly reduced ARAF-KSR1 complex formation, as shown by co-immunoprecipitation (Supplementary Fig. 8d, blue arrow). These reciprocal effects demonstrate that growth factor/RTK/RAS activation controls the ARAF-KSR1 interaction in resistant cells.

Supporting this RAS-dependent regulation, EGF-induced enhancement of complex formation coincided with enhanced exarafenib-induced thermal stabilization of all RAF proteins (ARAF; red boxes), while KSR1 showed minimal change (blue boxes), as demonstrated by CETSA (Supplementary Fig. 8c). Mechanistically, these results align with recent work showing that RAS activation primes RAF into a dimer-competent α C-IN state, which enhances the affinity of type II RAF inhibitors. Inhibitor binding then further stabilizes this conformation and promotes RAF protein complex formation (Vasta JD et al., Cell Chem Biol, 2023).

Fig. 5f**Sup Fig. 8d****Sup Fig. 8c**
(Fig. 5f) Co-immunoprecipitation analysis demonstrating growth factor regulation of ARAF-KSR1 interaction. NCI-H2405-ER cells were treated with or without EGF (100 ng/ml) and DMSO (-) or exarafenib (500 nM, +) simultaneously for 24 hours. EGF stimulation enhances ARAF-KSR1 interaction in resistant cells under drug treatment conditions. **(Sup Fig. 8c)** CETSA analysis of RAF isoforms and KSR1 in exarafenib-resistant NCI-H2405-ER cells treated with DMSO or exarafenib (500 nM) for 24 hours, with or without EGF stimulation. EGF enhanced exarafenib-induced thermal stabilization of all RAF isoforms (ARAF; red boxes). KSR1 showed minimal stabilization regardless of EGF (blue boxes). **(Sup Fig. 8d)** Co-immunoprecipitation analysis investigating RAS-dependent regulation of ARAF-KSR1 complex formation. NCI-H2405-ER cells were treated with DMSO, exarafenib (500 nM), RMC-6236 (1 μ M), or combination of both for 24 hours. ARAF immunoprecipitation demonstrates that the multi-RAS inhibitor RMC-6236 suppresses ARAF-KSR1 interaction, confirming RAS-dependent regulation of this protein complex.

Proximity ligation assays (PLA) provide additional validation, demonstrating increased ARAF-KSR1 proximity in resistant cells following exarafenib treatment compared to DMSO control, with less interaction observed in parental cells (Fig. 5g).

To gain structural insights into the exarafenib-binding mode within this complex, we performed AlphaFold-based molecular structural modeling. Docking analysis suggested that exarafenib primarily occupies the ATP-binding site of ARAF within the ARAF-KSR1 complex, contacting the gatekeeper residue (Thr382), residues in the hinge region (Gln383-Cys385), and the DFG motif (Asp447-Gly449), while also making

limited contacts with a short KSR1 segment (Pro408–Asn413) that lies adjacent to the pocket edge (Supplementary Fig. 8e and Supplementary Table S2). These observations indicate that exarafenib is structurally compatible with binding at the kinase active site while the ARAF–KSR1 complex retains its scaffolding architecture and generally align with our experimental data discussed above.

Fig. 5g

Sup Fig. 8e

(Fig. 5g) Proximity ligation assay (PLA) analysis of ARAF-KSR1 spatial interaction in parental versus resistant cells. NCI-H2405 parental and NCI-H2405-ER cells were treated with DMSO (-) or exarafenib (500 nM, +) for 24 hours. PLA signals (red dots) indicate ARAF-KSR1 proximity. Nuclei are stained with DAPI (blue). Scale bar, 20 μ m. Quantification confirms significantly increased ARAF-KSR1 proximity in resistant cells following drug treatment (** $p < 0.001$, unpaired t-test). **(Sup Fig. 8e)** Docking analysis of exarafenib in the AlphaFold-predicted ARAF–KSR1 complex. Exarafenib binds the ATP pocket of ARAF, contacting the gatekeeper residue, residues in the hinge region and DFG motif, with additional contacts to a short KSR1 segment. ARAF chain: gray, KSR1 chain: yellow, Exarafenib: red, ARAF gatekeeper and hinge region (Thr382–Cys385): Cyan, ARAF DFG motif (Asp447–Gly449): orange, KSR1 segment (Pro408–Asn413): blue, Chemical bonds: magenta.

During our co-immunoprecipitation studies, we also attempted to probe for RAS interactions but could not obtain consistent, reproducible results for these proteins in RAF immunoprecipitates, consistent with previous reports indicating that standard co-IP conditions may not stably capture these more transient or low-affinity interactions, likely due to the dynamic nature of kinase-substrate interactions, antibody epitope masking in protein complexes, potential disruption during cell lysis, and the typically lower endogenous protein expression levels in patient-derived cell line models compared to isogenic overexpression systems (Monaco KA et al., Clin Cancer Res. 20221).

To assess the functional significance of KSR1 in maintaining resistance, we performed KSR1 knockdown experiments. KSR1 silencing selectively restored exarafenib sensitivity in resistant cells, with IC₅₀ shifting from 5276 nM to 884 nM while having minimal impact on parental cell viability (Fig. 5h and Supplementary

Fig. 8f). Protein analysis shows that KSR1 depletion further suppresses MAPK pathway activation (p-MEK, p-ERK, p-RSK) in resistant cells following exarafenib treatment, while showing no change in parental cells (Supplementary Fig. 8g). Collectively, these findings establish KSR1 as an essential scaffolding hub that enables ARAF-dependent MAPK reactivation, thereby sustaining exarafenib resistance.

Given that KSR1 functions as a scaffold protein that enhances RAF kinase efficiency (Mckay MM et al., *Curr Biol.* 2011 and Daley RB et al., *Proc Natl Acad Sci USA*, 2023), we hypothesized that the drug-bound ARAF-KSR1 complex maintains functional signaling capacity by recruiting MEK. Consistent with this hypothesis, MEK immunoprecipitation revealed a marked increase in MEK-KSR1 association in resistant cells under exarafenib treatment (red arrow), whereas MEK-ARAF complexes were not detected (blue arrow), perhaps due to lack of substantial physical proximity (Supplementary Fig. 8h). Although associations with BRAF and CRAF were also detected, these appeared unlikely to play a major role in resistance, as their genetic depletion produced minimal effects on exarafenib sensitivity (Fig. 5a). While exarafenib binding reduces ARAF intrinsic kinase activity, increased upstream RAS activation coupled with enhanced KSR1 scaffolding of the ARAF/KSR1/MEK protein complex in the resistant cells may compensate for this reduction by improving MEK recruitment and positioning, thereby enabling the residual ARAF catalytic activity to sustain MAPK pathway signaling more efficiently despite drug treatment. A prediction of our general model is that supra-physiologic doses of exarafenib should overcome this mechanism and resistance. Our dose-response analysis supports this model: we show that the resistant cells retain sensitivity to supra-clinical exarafenib concentrations and display a sigmoid response curve rather than complete resistance (Supplementary Fig. 8f, siSCR). This pattern indicates that at therapeutic doses, KSR1 scaffolding buffers the loss of significant ARAF activity despite drug binding, but at higher concentrations, the remaining kinase function becomes insufficient to sustain cell viability even with enhanced scaffolding efficiency.

Fig. 5h

Sup Fig. 8f

Sup Fig. 8g

Sup Fig. 8h

(Fig. 5h) Heat map representation of IC50 values for exarafenib treatment in parental NCI-H2405 and resistant NCI-H2405-ER cells following scrambled control (SCR) or KSR1 siRNA (siKSR1) treatment. IC50 values were calculated from dose-response curves shown in Supplementary Fig. 8f. The color scale represents IC50 values in nM, demonstrating selective sensitization of resistant cells upon KSR1 depletion.

(Sup Fig. 8f) Cell viability dose-response curves showing the effect of KSR1 knockdown on exarafenib sensitivity in NCI-H2405 parental and resistant cells. Cells were transfected with scrambled control (SCR) or KSR1 siRNA (siKSR1) and treated with increasing concentrations of exarafenib. Data represent mean \pm SD. ****P < 0.0001.

(Sup Fig 8g) Western blot analysis of MAPK pathway components in NCI-H2405 and NCI-H2405-ER cells following KSR1 knockdown and exarafenib treatment. Cells were transfected with scrambled control (SCR) or KSR1 siRNA and treated with exarafenib (+) or vehicle (-). KSR1 depletion enhances suppression of p-MEK, p-ERK, and p-RSK in resistant cells under drug treatment while showing no effect in parental cells.

(Sup Fig 8h) Co-immunoprecipitation analysis of MEK complexes in parental (NCI-H2405) and resistant (NCI-H2405-ER) cells treated with DMSO or exarafenib (500 nM) for 24 hours. In resistant cells, exarafenib treatment enhanced MEK–KSR1 association (red arrow) without detectable MEK–ARAF binding (blue arrow).

We also attempted several additional mechanistic studies to further characterize ARAF's role in resistance. To determine which specific ARAF domains are essential for resistance, we attempted to rescue ARAF knockout in resistant cells with various ARAF mutants (RAS-binding deficient, dimer-deficient, kinase-dead variants, and gate keeper mutation to prevent drug binding). However, these experiments were hindered by poor exogenous ARAF expression following endogenous ARAF knockout, potentially due to off-target effects of the CRISPR-Cas9 system despite designing sgRNAs to avoid targeting exogenous constructs. Additionally, given recent reports that ARAF forms plasma membrane condensates that compete with (and occlude) the RAS GAP NF1 for RAS binding and inactivation to thereby activate RAS (Li W et al., Nat Chem Biol. 2025), we investigated whether similar membrane compartmentalization occurs in our resistant cells through plasma membrane fractionation studies. However, we could not obtain reproducible results demonstrating such ARAF membrane condensate formation in our resistance models, suggesting that the resistance mechanisms we identified may be distinct from the condensate-mediated pathway described in those interesting studies.

Taken together, our identification of the RAS-ARAF-KSR1 signaling axis as a novel resistance mechanism, combined with the functional validation of ARAF's critical role across multiple BRAF mutation classes, establishes ARAF as a key mediator of pan-RAF inhibitor resistance (Fig. 8). Mechanistically, our findings reveal that resistant cells undergo an adaptive switch from conventional BRAF-dependent signaling to an ARAF-dependent bypass mechanism specifically under drug treatment conditions. This switch is characterized by enhanced ARAF-KSR1 complex formation and requires the increased upstream growth factor/RTK/RAS activation present in the resistant cells or induced by growth factor treatment. Notably, while exarafenib directly binds to ARAF in resistant cells, it likely induces conformational changes that facilitate KSR1 recruitment, suggesting that the drug may function as a molecular glue enabling this resistance complex assembly. Functionally, KSR1 scaffolding enhances MEK recruitment to the drug-bound ARAF complex, thereby maintaining MAPK pathway activation

through improved ARAF kinase efficiency and/or effector (MEK) positioning to thereby buffer the therapeutic efficacy of exarafenib.

While our comprehensive genetic and biochemical studies consistently identified ARAF as the dominant RAF isoform mediating resistance, the functional contribution of CRAF within this resistance architecture warrants additional consideration. Although CRAF depletion alone failed to significantly impact exarafenib sensitivity in any resistant cell line tested (Fig. 5a and Supplementary Fig. 8a), our co-immunoprecipitation analyses revealed that exarafenib treatment induces both ARAF-KSR1 and ARAF-CRAF complex formation in resistant cells (Fig. 5e). This raises the possibility that CRAF may play a supportive or redundant role within the ARAF-centered resistance network. The limited effect of CRAF knockdown alone in our resistant models may reflect several possibilities: CRAF could be functionally dispensable in the presence of intact ARAF-KSR1 scaffolding, it could play only a minor supportive role, or its contribution could be masked by compensatory mechanisms from the dominant ARAF-centered signaling network. Future studies employing simultaneous depletion of multiple RAF isoforms, particularly ARAF and CRAF together, would be valuable to dissect potential cooperative or redundant functions within the resistance complex. Such experiments could reveal whether CRAF contributes to maintaining MAPK signaling in the presence of ARAF or whether complete disruption of the RAF heterodimer network requires targeting both isoforms. Understanding these nuanced interactions could inform the development of next-generation isoform-directed RAF inhibitors or combination strategies that more comprehensively disable the adaptive RAF signaling architecture that emerges under therapeutic pressure.

These mechanistic insights are entirely distinct from resistance patterns observed with first-generation selective BRAF inhibitors and represent a novel understanding of the dynamic intricacies of MAPK pathway wiring and of pan-RAF inhibitor resistance. Furthermore, our work specifically addresses Class II and III BRAF mutations, which represent 65% of BRAF-mutant NSCLC cases and possess distinct biology from the well-studied Class I mutants that have dominated previous resistance studies. Importantly, the resistance mechanisms we describe occur in the context of a highly selective pan-RAF inhibitor that lacks the rapid paradoxical activation that occurs with selective V600X inhibitors, providing insights into resistance development under highly clean RAF inhibition conditions that were not previously achievable. These findings are comprehensively described in the Results section under 'Exarafenib Resistance Involves an Adaptive Switch to ARAF-Dependent MAPK Signaling' and 'ARAF-KSR1 Complex Formation Mediates Resistance-Associated MAPK Reactivation' (pages 15-18, lines 513-646), with supporting data presented in Fig. 5a-h, Supplementary Fig. 8a-h, and the mechanistic model summarized in Fig. 8.

Fig. 8 RAF Isoform Dependency Switch

(Fig. 8) Schematic model illustrating MAPK signaling mechanisms in BRAF Class II & III mutant exarafenib-resistant cells under different treatment conditions. **Column 1 (Pretreatment baseline/drug-free):** Resistant cells at baseline exhibit high MAPK signaling through conventional oncogenic-BRAF-dependent pathway. **Column 2 (Exarafenib treatment):** Under exarafenib treatment, resistant cells maintain sustained MAPK activation through an alternative ARAF-dependent pathway. RTK activation enhances RAS-GTP loading, which facilitates formation of an ARAF–KSR1 complex stabilized by exarafenib. This emergent complex provides a bypass route for pathway reactivation under drug pressure. **Column 3 (Exarafenib + MEKi combination):** Combination therapy with MEK inhibitor (MEKi) effectively suppresses the reactivated MAPK pathway by targeting downstream signaling, resulting in improved pathway suppression. Black arrows indicate active signaling pathways; gray arrows indicate suppressed pathways. Exa, exarafenib.

Clinical and Translational Significance:

- Mechanistic Rationale for Current Clinical Trial Strategy:** The time-course western blot analysis (Fig. 6d) demonstrates that exarafenib monotherapy leads to progressive reactivation of MAPK pathway activity over 48 hours, as evidenced by recovery of p-MEK, p-ERK, and p-RSK levels. Therefore, we tested the hypothesis that downstream pathway inhibition would be an effective approach against resistance. Importantly, combination treatment with binimetinib effectively suppresses this reactivation, maintaining pathway inhibition of p-ERK and p-RSK throughout the treatment period (p-MEK is accumulated in the presence of binimetinib, in line with known effects (Hatzivassiliou G et al., Nature, 2013)). This finding provides direct mechanistic justification for the MEK combination arm in the ongoing KN-8701 clinical trial (NCT04913285), demonstrating that MEK inhibition specifically targets the downstream signaling of the RAS-ARAF-KSR1

pathway reactivation we identified. Encouragingly, preliminary results from the combination therapy arm of the KN-8701 clinical trial (NCT04913285) have recently demonstrated promising tolerability and activity of exarafenib in combination with binimetinib in BRAF-altered solid tumors and NRAS-mutant melanoma, as predicted by our preclinical mechanistic work (Cassier P et al., *Annals of Oncology* 2024;35:S491-S2). The time-course analysis demonstrating MAPK pathway reactivation under exarafenib monotherapy and its suppression by binimetinib combination is presented in Fig. 6d (Results section, page 22, lines 756-761).

Fig. 6d

(Fig. 6d) Time-course western blot analysis demonstrating MAPK pathway reactivation in NCI-H2405-ER resistant cells and the effect of combination therapy. Cells were treated with 500 nM exarafenib alone or in combination with 50 nM binimetinib for the indicated time points (0, 4, 24, 48 hours). While exarafenib monotherapy leads to MAPK pathway reactivation over 48 hours, combination treatment with binimetinib effectively suppresses this reactivation and maintains pathway inhibition throughout the treatment period.

The RAS-ARAF-KSR1 mechanism represents a novel therapeutic target for RAS, ARAF and/or KSR-selective inhibitors. Remarkably, follow-up studies addressing other Reviewer questions demonstrated that even RAFi/MEKi combination-resistant cells showed restored sensitivity upon ARAF knockdown (detailed below in Reviewer #2 response), suggesting that ARAF (or upstream RAS) could emerge as conserved mechanism of resistance clinically and thus selective ARAF or RAS inhibition (for instance with RMC-6236) could overcome multiple resistance layers and may warrant clinical investigation.

2. **Comprehensive Real-World Genomic Landscape Analysis:** In response to other Reviewers' valuable suggestions, our expanded GuardantINFORM™ analysis now provides comprehensive real-world evidence including detailed treatment patterns and co-occurring RAS mutation frequencies across BRAF classes, representing the largest such clinical dataset analyzed to date (detailed below in Reviewer #2 and #3 response).

3. **Clinical Validation of Exarafenib Efficacy in BRAF-Mutant NSCLC Patients:** We have now included clinical outcome data from the ongoing phase I/Ib exarafenib trial KN-8701 (NCT04913285) to demonstrate direct translational impact of our findings focusing on clinical activity of exarafenib monotherapy.

Case 1: A 38-year-old male with TMEM106B-BRAF fusion (Class II variant) lung cancer achieved a confirmed partial response on exarafenib 800 mg/day, with significant target lesion reduction and rapid ctDNA clearance, maintaining response until progression at 32 weeks with new metastases (Fig. 7a-c).

Fig. 7a

Fig. 7c

Fig. 7b

(Fig. 7a) A 38-year-old male with stage 4 lung adenocarcinoma harboring a TMEM106B-BRAF fusion. Treatment course (top): red - 1st line therapy; pink - maintenance therapy; green - second line therapy; blue - exarafenib. The line graph (bottom) illustrates the sum of target lesion diameters (SUM) and individual target lesion diameters over time. The 30% reduction line is shown for the determination of partial response (PR). **(Fig. 7b)** CT scans of liver metastases in the patient from **(Fig. 7a)** before treatment (left) and after 5 cycles of exarafenib (right), showing a partial response. Arrows indicate the target lesions. **(Fig. 7c)** Longitudinal monitoring of various cancer-associated genes detected in ctDNA from the patient in panel **(Fig. 7a)**. Each colored line represents a specific gene alteration, and the y-axis denotes the variant allele frequency (VAF) of each alteration, expressed as a percentage. CDDP, cisplatin; PEM, pemetrexed; PEMBRO, pembrolizumab.

Case 2: An 84-year-old male with BRAF G469S mutation (Class II) lung cancer demonstrated remarkable

clinical improvement, weaning off oxygen therapy within 2 weeks and achieving sustained partial response over 2 years on exarafenib 400 mg/day (Fig. 7d).

Fig. 7d

Fig. 7e

(Fig. 7d) A 84-year-old male with stage 4 lung adenocarcinoma harboring a BRAF G469S mutation. Treatment course (top): red - 1st line therapy; pink - maintenance therapy; blue - exarafenib. The line graph (bottom) illustrates the sum of target lesion diameters (SUM) and individual target lesion diameters over time. To evaluate partial response (PR), a threshold line at 30% reduction is presented. **(Fig. 7e)** Chest CT scans of the patient in **(Fig. 7d)** before treatment (left) and after 2 cycles of exarafenib (right), demonstrating significant improvement in bilateral lung infiltrates. CBDCA, carboplatin; PEM, pemetrexed; PEMBRO, pembrolizumab.

Both patients experienced manageable Grade 2 dermatitis requiring dose modifications but maintaining therapeutic benefit. These clinical cases provide compelling evidence of exarafenib's promising clinical activity with manageable toxicity in BRAF-mutant lung cancer patients with unmet clinical need, directly supporting our preclinical mechanistic findings. These clinical cases are presented in the Results section (pages 25-26, lines 866-906) and illustrated in Fig. 7a-e.

In conclusion, we are grateful for the Reviewer's expertise, which has helped us better articulate our contributions. While RTK-RAS activation has been described previously, our study provides entirely novel mechanistic insights

into specific RAF isoform dependencies, switches, and protein complex formations that maintain MAPK signaling under pan-RAF inhibition. These discoveries, combined with comprehensive clinical characterization and direct translational relevance to ongoing clinical trials, represent substantial advances that we believe merit high-impact publication.

Reviewer #2:

We appreciate the thoughtful consideration of, and feedback on our manuscript as well as the enthusiasm for our study. We address the points raised below.

-Re: Guardant database used - were these liquid biopsies all at initial diagnosis, or some during/after treatment?

We thank the reviewer for this important clarification regarding the timing of liquid biopsy collection in our study. All patients included in our GuardantINFORM™ analysis had advanced or metastatic disease, as this is the prime clinical use of Guardant360 assay testing. Within this advanced/metastatic population, the Guardant 360 liquid biopsies were collected at various timepoints that reflect real-world clinical practice - some were performed at initial diagnosis while others were conducted during disease progression. This timing diversity provides a comprehensive view of BRAF mutation prevalence across different phases of disease management in the metastatic and real-world clinical setting, offering a valuable contribution to the field. This clarification has been added to the Methods section (Page 46, Line 1562-1582).

-Re: rwOS from Guardant database - any data available re: Stage/Treatment? What was timeframe looked at? (i.e. were these all in the IO era, as differences in types of treatments offered over time could have an impact on survival outcomes?)

We thank the reviewer for this important question regarding the timing of liquid biopsies and potential bias from treatment evolution during the study period.

The GuardantINFORM™ database used in our analysis spans from March 2014 to June 2021, predominantly encompassing the immunotherapy era when immune checkpoint inhibitors were established as a standard of care for NSCLC. The most commonly used first-line treatment regimens (top 3) for each BRAF class are displayed as Fig. 1d. The entire treatment information for 1st, 2nd, and 3rd-line therapies by BRAF mutation class has been provided in Supplementary Data 1-3. As demonstrated in these datasets, immunotherapy-containing regimens represent a substantial proportion of treatments across all BRAF mutation classes, further confirming that our survival analyses reflect outcomes in the modern immunotherapy era. These treatment data and the expanded clinical analysis have been incorporated into the Results section (pages 5, lines 158-163).

Fig. 1d

BRAF Class	First Line Regimen (top 3)	Frequency	Percent
Class I (n=307)	trametinib + dabrafenib	73	23.8
	pembrolizumab	45	14.7
	carboplatin + pemetrexed + pembrolizumab	33	10.8
Class II (n=271)	carboplatin + pemetrexed + pembrolizumab	53	19.6
	pembrolizumab	41	15.1
	carboplatin + pemetrexed	27	10.0
Class III (n=271)	carboplatin + pemetrexed + pembrolizumab	45	17.9
	pembrolizumab	44	17.5
	carboplatin + pemetrexed	27	10.8

(Fig. 1d) First-line treatment regimens for BRAF-mutant NSCLC patients by mutation class from the GuardantINFORM™ database. The table shows the top 3 most frequently used first-line therapies for each BRAF mutation class (Class I, II, and III) with corresponding patient numbers and percentages.

Regarding disease stage, all patients in our analysis had advanced/metastatic disease at the time of testing, providing a consistent baseline for survival comparisons across different BRAF mutation classes. This clarification regarding the study timeframe and treatment details has been added to the Methods section (Page 46, Line 1562-1582).

-Re: activity seen against V60E p61 splice variant conferring resistance - what proportion of resistance is comprised of this alteration? Do they have data from the Guardant database that can inform prevalence of this?

Thank you for raising this important point. While resistance mechanisms, including V600E p61 splice variants, have been well characterized in melanoma, their frequency in NSCLC remains an area of ongoing study due to the challenges in obtaining rebiopsy specimens after progression (Poulikakos PI, Nature, 2011; Tsamis L et al., Clin Transl Oncol., 2023). We did detect this variant in human BRAF mutant lung cancer with BRAF inhibitor resistance in our prior report describing our discovery of this resistance variant in human BRAF class I mutant lung cancers treated with first generation BRAF inhibitors (Lin L et al., PNAS, 2014). Currently, the GuardantINFORM™ database's liquid biopsy assay does not include RNA sequencing capabilities, which would be important for readily detecting splice variants. This technical limitation, which is common across many liquid biopsy platforms, highlights the need for continued development of comprehensive molecular profiling approaches. Future iterations of the assay are planned to include RNA analysis capabilities, which will help address this knowledge gap.

-Re: RAS activation seen in exarafenib resistance - the experiments investigating RTKs as a potential mediator of RAS activation did not seem to yield a clear mechanism. did the authors perform any other mechanistic studies to try to get at how this is occurring?

We thank the Reviewer for this insightful question regarding the mechanistic basis of RAS activation in exarafenib resistant cells. While our initial RTK profiling identified activation of multiple RTKs (e.g. AXL, MET, EGFR) in resistant cells, individual RTK knockdowns showed limited effects on exarafenib sensitivity, suggesting functional redundancy and that no single RTK was solely responsible for the RAS activation observed upon exarafenib treatment. To address the Reviewer's question about the underlying mechanism, we performed additional mechanistic studies:

- 1. Combined RTK Silencing:** We conducted combined combined AXL and EGFR knockdown experiments in NCI-H2405-ER cells, which demonstrated a more pronounced reduction in RAS-GTP levels compared to individual knockdowns (Supplementary Fig. 7b). This finding supports our hypothesis that multiple RTKs function cooperatively to promote RAS activation in response to exarafenib exposure in resistant cells.

Sup Fig. 7b

(Sup Fig. 7b) Western blot analysis of RAS activation in NCI-H2405-ER resistant cells following individual and combined RTK knockdown. Cells were transfected with scrambled control (siSCR), siEGFR, siAXL, or combination of both and treated with exarafenib (+) or vehicle (-). Combined AXL and EGFR knockdown demonstrates more pronounced reduction in RAS-GTP levels compared to individual knockdowns.

- 2. Serum Dependency Analysis:** We investigated MAPK signaling under varying serum conditions (low vs. high serum) in the presence of exarafenib and observed that MAPK activation in resistant cells was serum-dependent (Supplementary Fig. 7e, red arrows), further supporting the role of extracellular growth factor signaling through RTKs in maintaining RAS-MAPK pathway activation. Notably, in parental NCI-H2405 cells, increased serum concentration showed no difference from low serum conditions in maintaining MAPK activity under exarafenib treatment (Supplementary Fig. 7e, blue arrows), indicating that the acquired ability to respond to extracellular growth factors is a specific characteristic of the resistant cells rather than an inherent property of the parental cell line.

Sup Fig. 7e

(Sup Fig. 7e) Western blot analysis of serum-dependent MAPK signaling in parental NCI-H2405 and resistant NCI-H2405-ER cells. Cells were cultured under low (3%) or high (10%) FBS conditions and treated with exarafenib (+) or vehicle (-). Resistant cells show enhanced MAPK pathway activation (p-ERK, p-RSK) specifically under high serum conditions during exarafenib treatment (red arrows), while parental cells show consistent drug sensitivity regardless of serum conditions (blue arrows).

SHP2 Inhibitor Synergy: We previously demonstrated synergistic effects between SHP2 inhibitor and exarafenib in resistant cells (Supplementary Fig. 9b). Since SHP2 is a critical mediator of RTK-to-RAS signaling, we believe this finding also provides additional mechanistic support for the RTK-RAS connection in the context of exarafenib-induced resistance mechanisms.

Sup Fig. 9b (excerpt)

(Sup Fig. 9b) Heatmap of synergy scores for exarafenib combined with RMC-4550 in parental and exarafenib-resistant NCI-H2405, NCI-H1755, and NCI-H1666 cell lines. Synergy scores were calculated using the Loewe model. Positive scores indicate synergy.

These findings collectively suggest that exarafenib resistance involves redundant and cooperative RTK signaling networks that promote RAS activation upon drug exposure, rather than dependence on a single dominant RTK. The additional mechanistic data have been incorporated into the Results section (page 14, lines 473-480 and 494-503) .

-preclinical synergy with MEK and ERK inhibitors is a currently active investigational strategy. resistance that develops in this setting would be of interest to understand.

We thank the Reviewer for this important comment regarding combination therapy resistance mechanisms. We generated exarafenib plus binimetinib resistant cells by sequentially adding binimetinib to established exarafenib-resistant NCI-H2405-ER cells with gradually escalating doses of binimetinib over three months, resulting in NCI-H2405-EBR (Exarafenib-Binimetinib Resistant) cells. While we also attempted to generate resistance by treating parental cells with both drugs simultaneously, this approach did not yield viable resistant cells for additional experiments despite six months of effort, highlighting the potent efficacy of this combination therapy *in vitro*.

To characterize the molecular mechanisms underlying combination resistance, we first examined RAS activation status, given its central role in monotherapy resistance. RAS-GTP pulldown assays revealed that NCI-H2405-EBR cells exhibited elevated RAS-GTP levels under combination drug treatment compared to parental cells (Supplementary Fig. 12c), mirroring the RAS activation pattern observed in exarafenib monotherapy-resistant cells. This finding indicated that combination-resistant cells retain the ability to elevate RAS-GTP levels under dual RAF-MEK inhibition.

Based on our concurrent findings that ARAF mediates resistance to exarafenib monotherapy through RAS-dependent mechanisms (detailed in Reviewer #1 response), we investigated whether this mechanism also contributes to combination therapy resistance. Remarkably, ARAF knockdown dramatically restored sensitivity to the drug combination in NCI-H2405-EBR cells, reducing IC50 values in the presence of binimetinib plus exarafenib (Fig. 6f). Western blot analysis further confirmed that ARAF knockdown enhanced MAPK pathway suppression under drug treatment (Supplementary Fig. 12d). These new data suggest that ARAF (or upstream RAS) could emerge as conserved mechanism of resistance clinically and thus selective ARAF or RAS inhibition (for instance with RMC-6236) could overcome multiple resistance layers and may warrant clinical investigation.

Sup Fig. 12c

Fig. 6f

Sup Fig. 12d

(Sup Fig. 12c) RAS-GTP pulldown assays in NCI-H2405 parental and combination-resistant (EBR) cells. Cells were treated with exarafenib (500 nM) in combination with binimetinib (50 nM) for 24 hours. EBR cells exhibit elevated RAS-GTP levels under combination drug treatment compared to parental cells. **(Fig. 6f)** Heatmap showing cell viability IC₅₀ values (nM) for exarafenib in NCI-H2405-EBR cells treated with varying concentrations of binimetinib (up to 100 nM) in combination with scrambled control (SCR) or ARAF siRNA knockdown. ARAF knockdown restores sensitivity to the drug combination, demonstrating ARAF's critical role in mediating resistance to RAF-MEK combination therapy. **(Sup Fig. 12d)** Western blot analysis of MAPK pathway components in NCI-H2405-EBR cells following ARAF knockdown. Cells were transfected with scrambled control (SCR) or ARAF siRNA and treated with exarafenib (500 nM) and binimetinib (50 nM) combination for 24 hours. ARAF knockdown enhances suppression of p-MEK, p-ERK, and p-RSK levels under drug treatment.

These findings are particularly significant because the fact that ARAF knockdown effectively restored sensitivity in both monotherapy-resistant and combination-resistant models suggests that ARAF represents a common and critical convergent node for resistance across different therapeutic contexts. This identifies ARAF (and upstream RAS) as a potential therapeutic target for overcoming resistance to RAF-based combination therapies and opens new avenues for developing next-generation therapeutic strategies. These findings have been incorporated into the Results section (Page 24, Line 840-864).

Reviewer #3:

In the context of exarafenib resistance, the lack of evident genetic alterations as potential mediators together with the observed exarafenib re-sensitization following a drug holiday period are all indicative of adaptive resistance mechanisms. Yet, the re-challenge experiments described in Figure 4c and Ext Fig 5 are not entirely clear. The authors implemented a 48 hr drug-free period during which they claim that “the resistant cell lines remain resistant -see text line 331-“ The meaning of this sentence is unclear. Furthermore, 4 hours post-drug exposure following the drug-free period all resistant cell lines showed

a dose-dependent inhibition of the MAPK pathway. If the cells respond to the drug re-challenge, it is unclear why they were considered “resistant” during the drug holiday period. In order to clarify the underlying mechanisms it would be informative to perform a more detailed time-course analysis of the ER cells when kept in the absence of drug as well as during the re-addition of exarafenib. Furthermore, it would be informative to carry out this exercise with concomitant measurements of cell proliferation.

We appreciate the Reviewer's thoughtful consideration and constructive feedback regarding exarafenib resistance mechanisms and experimental design. We acknowledge that our initial presentation lacked clarity regarding the resistance mechanism and the rationale for our experimental approach. We address these important points below.

The 48-hour drug-free period was implemented as a standardized experimental protocol to establish consistent baseline conditions between parental and resistant cell lines while ensuring that the resistant phenotype was maintained. During this period, the resistant cells retain their acquired resistance characteristics, as evidenced by their continued ability to proliferate at drug concentrations that would normally be inhibitory to parental cells.

The apparent contradiction noted by the Reviewer—where resistant cells show initial MAPK pathway inhibition upon drug re-challenge yet maintain resistance—reflects the complex dynamics of the resistance mechanism. Our time course experiments described below demonstrate that while exarafenib can still engage its target (RAF kinases) and transiently suppress MAPK signaling, the resistant cells have developed compensatory mechanisms that we have now elucidated (RAS-ARAF-KSR1 activation and dependence) that rapidly restore pathway activation and maintain cell survival.

To address the Reviewer's request for more detailed mechanistic insights, we have conducted comprehensive time-course analyses examining the temporal dynamics during both drug withdrawal and re-challenge periods. During drug withdrawal, MAPK pathway components (p-MEK, p-ERK, p-RSK) showed progressive elevation, reaching levels substantially higher than those observed during 72 hours of drug treatment (Supplementary Fig. 6a). Upon drug re-challenge, MAPK pathway components exhibited the previously observed pattern of initial suppression followed by reactivation, though the rebound levels remained moderate compared to the drug-free state.

Additionally, as suggested by the Reviewer, we performed parallel cell proliferation analyses by cell counting under the same experimental conditions during drug withdrawal and re-challenge periods (Supplementary Fig. 6b). The results demonstrate that resistant cells maintain their proliferative capacity throughout the drug re-challenge period, confirming that the transient MAPK pathway suppression does not translate to meaningful growth inhibition. This continuous proliferative capability contrasts with the drug-dependent growth phenotype often observed with MAPK pathway inhibitor resistance, where cells become dependent on continued drug exposure to prevent excessive MAPK activation and subsequent cell death. The absence of such drug dependence in our exarafenib-resistant cells suggests a distinct resistance mechanism that we now have elucidated (RAS-ARAF-KSR1) that enables cells to tolerate both drug withdrawal and re-exposure while

maintaining viability and proliferative capacity. The comprehensive time-course analyses during drug withdrawal and re-challenge, including cell proliferation data, have been incorporated into the Results section (page 11, lines 382-396) and are presented in Supplementary Fig. 6a-b.

Sup Fig. 6a

Sup Fig. 6b

(Sup Fig. 6a) Time-course analysis of MAPK pathway components, DNA damage markers, and apoptosis markers in NCI-H2405-ER cells during drug withdrawal and re-challenge. Cells were initially maintained under exarafenib treatment for 72 hours, then switched to drug-free medium (negative timepoints) followed by exarafenib re-treatment (positive timepoints). During withdrawal, MAPK pathway components show progressive elevation without significant induction of DNA damage or apoptotic markers. **(Sup Fig. 6b)** Cell number fold change over time for exarafenib-resistant NCI-H2405-ER cells. Cells were initially grown in the presence of exarafenib, then subjected to drug withdrawal (white area), followed by drug re-challenge (purple shaded area). Data points represent relative cell number normalized to the 72h time point (set as 1-fold). Results demonstrate that resistant cells maintain proliferative capacity throughout both drug withdrawal and re-challenge periods. Data shown as mean \pm SD.

Along these lines, does the drug holiday result in oncogenic toxicity stress/DNA damage due to MAPK hypersignaling? This has been observed in other MAPK driven tumours when inhibitors are eliminated.

The Reviewer raises an important point about potential oncogenic stress and DNA damage due to MAPK hypersignaling during drug withdrawal, as observed in other MAPK-driven tumors. To address this question, we examined markers of DNA damage, apoptosis, and cellular stress during our comprehensive time-course analysis described above (Supplementary Fig. 6a).

Despite enhanced MAPK pathway activity during drug withdrawal, western blot analysis across withdrawal

timepoints (-4h, -24h, -72h) showed no significant induction of DNA damage markers (p- γ H2AX and p-CHK2), apoptosis marker (cleaved PARP), or pro-apoptotic protein (BIM). While some markers showed modest fluctuation (such as γ -H2AX elevation comparable to 72-hour treatment levels), these changes did not indicate significant cellular stress or death, as indicated by our cell proliferation analyses (Supplementary Fig. 6b).

These findings suggest that the exarafenib-resistant cells have adapted mechanisms that allow them to tolerate sustained and even enhanced MAPK pathway activity without experiencing the oncogenic stress typically associated with MAPK hyperactivation in other contexts. These data have been added to the Results section (page 12, lines 398-409).

Adaptive resistance mediated by upstream RAS/RTK signalling was demonstrated by performing either panRAS or KRAS knockdown. Yet, the underlying mechanisms that promote RAS activation likely differ across the different ER models. SHP2, MEK, ERK or AKT inhibitors were assessed in this resistance setting eventually selecting the MEK inhibitor binimetinib. The biological rationale to select this combination for further characterization is sound. Furthermore, there is also clinical interest given the ongoing clinical trial KN-8701 comparing monotherapy versus combination therapy. Yet, taking into account the recent developments in the RAS inhibitor field, there is an obvious combination with panRAS inhibitors that should have been tested. This combination is potentially applicable across the different ER cell models and could be clinically informative.

We appreciate this valuable suggestion. While we have tested several combinations including SHP2, MEK, ERK, and AKT inhibitors, we agree that evaluating multi-RAS inhibitors would be highly relevant given recent developments in the field. We expanded our combination studies to include an available multi-RAS inhibitor (RMC-6236; Jiang J et al., Cancer Discov, 2024) across our exarafenib-resistant models using full dose-response matrices similar to our other combination studies (Fig. 6a and Supplementary Fig. 9a, b).

The multi-RAS inhibitor combination demonstrated differential anti-proliferative effects across the cell line panel. While NCI-H2405 parental cells showed limited additional benefit from the combination, both NCI-H1755 and NCI-H1666 parental cells, as well as all three resistant cell lines (NCI-H2405-ER, NCI-H1755-ER, and NCI-H1666-ER), showed effective suppression of cell proliferation when treated with the exarafenib plus RMC-6236 combination (Fig. 6a and Supplementary Fig. 9a). More importantly, synergy analysis revealed that the combination showed enhanced synergistic interactions, particularly in the resistant cell lines compared to their parental counterparts (Supplementary Fig. 9b).

Fig. 6a (excerpt)**Sup Fig. 9a (excerpt)****Sup Fig. 9b (excerpt)**
(Fig. 6a) Cell viability heatmaps showing IC₅₀ values (nM) for cell proliferation in parental and resistant cell lines treated with exarafenib in combination with RMC-6236. NCI-H2405, NCI-H1755, and NCI-H1666 parental and resistant cells were treated with varying concentrations of RMC-6236 (up to 3 μ M). All resistant cell lines show enhanced sensitivity to the combination compared to monotherapy treatment. **(Sup Fig. 9a)** Cell viability heatmaps showing percentage cell viability for the same experimental conditions as Fig. 6a. **(Sup Fig. 9b)** Loewe synergy score heatmaps for the exarafenib and RMC-6236 combination dataset. Synergy scores range from -50 (antagonism) to +50 (synergy). The combination shows enhanced synergistic interactions particularly in resistant cell lines compared to their parental counterparts, supporting the role of RAS activation in resistance mechanisms.

Western blot analysis confirmed that the exarafenib plus RMC-6236 combination resulted in stronger suppression of MAPK pathway signaling in the resistant cells (Supplementary Fig. 10c), pharmacologically recapitulating and extending our previously shown RAS knockdown experiments. This provides additional validation that RAS activation represents a key resistance mechanism across the different resistant models, with the synergistic interaction observed across all three resistant cell lines (NCI-H2405-ER, NCI-H1755-ER, and NCI-H1666-ER). The RMC-6236 combination studies have been added to the Results section (page 20, lines 708-717; page 21, lines 726-729; page 22, lines 765-767).

Sup Fig. 10c

(Sup Fig. 10c) Western blot analysis of MAPK pathway components and downstream targets in parental and resistant cell lines treated with exarafenib and RMC-6236 combination. NCI-H2405, NCI-H1755, and NCI-H1666 parental and resistant (-ER) cells were treated with exarafenib (500 nM), RMC-6236 (1 μ M), or combination of both for 48 hours. The combination treatment results in stronger suppression of MAPK pathway signaling in resistant cells compared to monotherapy treatments.

While I have no reason to doubt the outcome of the *in vitro* results, it is somehow surprising that RAS reactivation is a key resistance mediator but, yet, a naïve PDX harbouring a concomitant NRASQ61H mutation is clearly sensitive to exarafenib (CTG-1444 PDX in Figure 3). I wonder to what extent the resistance mechanisms leading to RAS reactivation identified in the various ER cells would be operative under low mitogenic stimulation. The authors may want to repeat some of the *in vitro* drug-holiday + re-challenge experiments in low serum conditions.

We appreciate this insightful observation regarding the relationship between RAS activation states in different contexts. To address this question, we conducted additional experiments examining the role of mitogenic stimulation in the resistance mechanisms we identified. Following a 48-hour drug holiday, we re-treated both parental and resistant NCI-H2405 cells with exarafenib under different serum concentrations (3% and 10% FBS) (Supplementary Fig. 7e). Western blot analysis revealed that resistant cells showed enhanced MAPK pathway reactivation (p-ERK, p-RSK) specifically under 10% FBS conditions compared to 3% FBS (Supplementary Fig. 7e, red arrows), while parental cells showed consistent drug sensitivity regardless of serum conditions (Supplementary Fig. 7e, blue arrows). Given that ARAF emerged as a critical mediator of MAPK pathway reactivation in our additional studies, we next examined whether ARAF mediates the serum-dependent effects described above (Fig. 5d). Knockdown of ARAF abolished this serum-dependent MAPK pathway reactivation in resistant cells, with p-ERK, p-RSK, and p-MEK levels remaining equally suppressed under both serum conditions

(Fig. 5d, red arrows).

Sup Fig. 7e

Fig. 5d

(Sup Fig. 7e) MAPK pathway analysis in NCI-H2405 parental and resistant cells treated with exarafenib (500 nM) under low (3%) or high (10%) FBS conditions. Resistant cells show enhanced pathway reactivation under high serum conditions. **(Fig. 5d)** ARAF's role in growth factor-dependent pathway reactivation. NCI-H2405-ER cells transfected with SCR or ARAF siRNA were treated with exarafenib (500 nM) for 24 hours under low (3%) or high (10%) FBS conditions. ARAF knockdown abolishes serum-dependent MAPK reactivation.

In our broader investigation of ARAF-mediated mechanisms, we identified ARAF-KSR1 protein interactions as potentially important. Building on this finding and in the context of the mitogenic signal-dependent resistance observed above, we examined ARAF-KSR1 binding under drug treatment with EGF stimulation (Fig. 5f). We observed that EGF stimulation further enhanced ARAF-KSR1 binding specifically in resistant cells under drug treatment, providing additional evidence that elevated growth factor signaling drives the observed mechanism of resistance.

Fig. 5f

(Fig. 5f) Co-immunoprecipitation analysis of ARAF protein interactions in NCI-H2405-ER cells. Cells were treated with or without EGF stimulation and exarafenib (500 nM) for 24 hours. Whole cell lysate (WCL) and ARAF immunoprecipitates (ARAF-IP) were analyzed for ARAF, BRAF, CRAF, and KSR1. EGF stimulation enhances ARAF-KSR1 interaction specifically under drug treatment conditions.

These mechanistic insights explain the apparent paradox raised by the Reviewer. While RAS activation is necessary for resistance, it is not sufficient on its own in the treatment naïve setting which is what the PDX reflects. The key distinction lies in the adaptive rewiring that occurs during acquired resistance development. The CTG-1444 PDX model harbors an endogenous NRAS Q61H mutation that provides constitutive RAS-GTP levels but critically lacks the acquired adaptive changes provided by ARAF-KSR1 co-optation and bypass signaling under drug pressure that are critical for resistance. In other words, this naïve system likely lacks the acquired ability to reactivate the MAPK pathway through mitogenic signal-induced ARAF-KSR1 complex formation that drives resistance in our models. This would explain why, despite elevated baseline RAS activity, the CTG-1444 PDX remains exarafenib-sensitive because it lacks the sophisticated bypass machinery that emerges through prolonged drug exposure and selection pressure. Regarding the Reviewer's question about low mitogenic stimulation, our serum titration experiments demonstrate that the resistance mechanisms we identified are indeed operative and critically dependent on mitogenic signals, with ARAF serving as the key mediator of this serum-dependent resistance. These data have been incorporated into the Results section, including serum titration experiments (page 14, lines 494-503; page 16, lines 545-553) and growth factor stimulation studies (page 17, lines 587-588).

Even more informative would be to assess the exarafenib sensitivity of a subset of these ER cells in vivo upon subcutaneous implantation in comparison to their parental counterparts.

We appreciate the Reviewer's valuable suggestion regarding the *in vivo* assessment of exarafenib-resistant cells. We attempted to establish CDX models using the resistant cell lines on multiple occasions over a 4-month period. Despite systematic optimization of engraftment conditions—including varying cell injection numbers, testing different immunocompromised mouse strains, and adjusting EGF concentrations within Matrigel—we were unable to successfully establish tumors from the resistant cells across four independent attempts. In contrast, the parental cells consistently formed robust tumors under identical conditions. We (and other investigators) have noted variable engraftment of tumors with acquired resistance to many targeted therapies in our broader and fairly substantial experience in studying acquired resistance in general (Okimoto RA et al., Proc Natl Acad Sci USA, 2016).

As previously demonstrated, the exarafenib-resistant cells exhibit increased dependency on multiple receptor tyrosine kinase (RTK) signaling pathways. Furthermore, our additional experiments conducted during this revision revealed that serum concentration is critical for the survival and proliferation of these resistant cells. Although this dependency is particularly evident under drug treatment conditions, the altered survival signaling

networks that confer drug resistance may simultaneously compromise the cells' ability to adapt to the *in vivo* milieu. The complex growth factor dependencies that have evolved in these resistant cells could render them particularly vulnerable to the transient nutrient-limited microenvironment encountered during the initial phases of tumor establishment *in vivo*, despite their robust growth in serum-rich culture conditions.

While we acknowledge that *in vivo* validation would strengthen our findings, the consistent failure to establish CDX models from resistant cells across multiple experimental approaches suggests fundamental alterations in their tumorigenic capacity that warrant further investigation in future studies.

Following this rationale, it would be interesting to know how many of the 2,398 BRAF mutant NSCLC patients identified upon scrutiny of the GuardantINFORM repository display concomitant RAS mutations. While this combination is unlikely to exist in Class I mutants, it could be informative in the case of Class II and Class III patients. An extended Table reporting this information could be valuable.

We thank the Reviewer for this insightful suggestion. We conducted additional analysis of RAS mutations in the GuardantINFORM repository for BRAF-mutant NSCLC patients. For this analysis, we specifically focused on established functionally significant RAS mutations (G12, G13, and Q61 variants) to exclude variants of uncertain significance, including silent mutations and rare variants. Interestingly, contrary to our initial expectations, we observed concomitant RAS mutations even in Class I patients, with higher frequencies in Class II and the highest rates in the upstream RAS-dependent Class III patients. Specifically, 76/845 (9.0%) Class I patients, 168/810 (20.7%) Class II patients, and 160/743 (21.5%) Class III patients harbored RAS mutations (Supplementary Fig. 4e). The higher co-occurrence rates in Class II and III patients are particularly noteworthy, as BRAF-RAS co-mutations are traditionally considered rare or mutually exclusive in other cancer types, such as colorectal cancer where they occur in <1% of cases (Midthun L et al., J Gastrointest Oncol, 2019). In NSCLC, however, several genomic studies have shown that non-V600E BRAF mutations, particularly those in Class II and III, are often accompanied by KRAS alterations (Dagogo-Jack I et al., Clin Cancer Res, 2019; Bourhis A et al., Clin Lung Cancer, 2020; Lin Q et al., J Transl Med, 2019), in line with our findings.

The substantial proportion of BRAF-mutant patients harboring concurrent RAS mutations (9-21.5% depending on BRAF class) demonstrates that RAS co-mutations occur frequently in clinical practice. While exarafenib demonstrated efficacy against individual RAS mutations in our initial studies and showed activity in the CTG-1444 PDX model harboring both BRAF G466V and NRAS Q61R mutations, the clinical efficacy of exarafenib in patients with concurrent BRAF-RAS mutations remains to be established. Future clinical studies examining treatment outcomes in this subpopulation could provide valuable insights into how concurrent RAS mutations influence therapeutic response in BRAF-mutant patients. The analysis of RAS co-mutation frequencies across BRAF mutation classes has been added to the Results section (page 10, lines 332-343) and is presented in Supplementary Figure 4e.

Sup Fig. 4e

	BRAF Class I (n=845)		Class II (n=810)		Class III (n=743)	
	# Patients	Percent (%)	# Patients	Percent (%)	# Patients	Percent (%)
KRAS	61	7.2	129	15.9	123	16.6
HRAS	3	0.4	7	0.9	3	0.4
NRAS	12	1.4	32	4.0	34	4.6
Total	76	9.0	168	20.7	160	21.5

(Sup Fig. 4e) Frequency of concomitant RAS mutations in BRAF-mutant NSCLC patients from the GuardantINFORM™ database. Table shows the number and percentage of patients harboring KRAS, HRAS, or NRAS mutations across BRAF Class I (n=845), Class II (n=810), and Class III (n=743) mutations.

In summary, we sincerely thank all Reviewers for the thoughtful comments and hope that the expanded, improved revised manuscript is now suitable for publication in *Nature Communications*.